# How do we see fractures? Quantifying subjective bias in fracture data collection.

Billy J. Andrews[1*], Jennifer J. Roberts[1], Zoe K. Shipton[1], Sabina Bigi[2], Maria C. Tartarello[2], Gareth O. Johnson[1,3]

1 Department of Civil and Environmental Engineering, University of Strathclyde, Glasgow, G11XJ, Scotland
2 Department of Earth Science, Sapienza – University of Rome, P.le Aldo Moro, 5, 00185 Rome, Italy
3 School of GeoSciences, University of Edinburgh, Edinburgh, EH93FE, Scotland

*Correspondence to:* Billy J. Andrews (billy.andrews@strath.ac.uk)

**Abstract.** The characterisation of natural fracture networks using outcrop analogues is important in understanding sub-surface fluid flow and rock mass characteristics in fractured lithologies. It is well known from decision sciences that subjective bias can significantly impact the way data is gathered and interpreted, introducing scientific uncertainty. This study investigates the scale of and nature of subjective bias on fracture data collected using four commonly used approaches (linear scanlines, circular scanlines, topology sampling and window sampling) both in the field and in workshops using field photographs. We demonstrate that geologists' own subjective biases influences the data they collect, and, as a result, different participants collect different fracture data from the same scanline or sample area. As a result, the fracture statistics that are derived from field data can vary considerably for the same scanline, depending on which geologist collected the data. Additionally, the personal bias of geologists collecting the data affects the scanline size (minimum length of linear scanlines, radius of circular scanlines or area of a window sample) needed to collect a statistically representative amount of data. Fracture statistics derived from field data often inputs into geological models that are used for a range of applications, from understanding fluid flow to characterising rock strength. We suggest protocols to recognise, understand and limit the effect of subjective bias on fracture data biases during data collection. Our work shows the capacity for cognitive biases to introduce uncertainty into observation-based data, and has implications well beyond the geosciences.

## 1 Introduction

Natural fracture networks exert a strong control on the hydrogeological and mechanical properties of a rock mass, and are useful indicators of palaeo-stress directions. Geological models that depict the spatial distribution and nature of a fracture network rely on input data (either distributions or mean values) of fracture statistics to provide a geologically reasonable model of the subsurface. Models such as discrete fracture networks (DFNs) may be used for estimating up-scaled permeability (e.g. (Bigi et al., 2013; Min et al., 2004)) or for rock mechanics analysis (Harthong et al., 2012; Jing and Hudson, 2002), with applications, including understanding fluid flow in tight oil and gas reservoirs (Aydin, 2000) and hydrogeology (Comerford et al., 2018), and assessing rock strength for mine engineering (Mas Ivars et al., 2011). Four methods for characterizing natural fractures in outcrops: linear scanlines (Priest, 1993; Priest and Hudson, 1981); circular scanlines (Mauldon et al., 2001; Rohrbaugh et al., 2002); topology sampling (characterising node types; Manzocchi, 2002; Sanderson and Nixon, 2015, 2018); and tracing out the fracture network (window sampling;(Wu and D. Pollard, 1995)). These methods handle orientation, censoring or truncation biases (Mauldon et al., 2001; Zeeb et al., 2013), and heterogeneity in the fracture network (Watkins et al., 2015) with different degrees of success. Here, we explore how each of these methods are susceptible to subjective uncertainties related to observer biases. Furthermore, we characterise how much the degree of variability introduced by subjective uncertainties is dependent on the method of data collection.

Uncertainties in geological data can be broadly split into objective and subjective uncertainty (Tannert et al., 2007). Objective uncertainty (also called external, aleatory inherent, structural, random, or stochastic uncertainty) refers to more traditional concepts of uncertainty, such as precision or processing error in a technique or a dataset, and can be represented through error bounds. Subjective uncertainty (also called epistemic, knowledge, functional, or internal uncertainty) arises from the mind, that is, stems from biases that affect how individuals perceive, gather and interpret geological data (Bond et al., 2015). Subjective uncertainty is common in geosciences where developing geological models typically relies on extrapolation of sparse data (Wood and Curtis, 2004), but its magnitude and impact is difficult to quantify (Bond et al., 2015).

The collection of fracture attributes will be affected by subjective biases. Depending on the aims of a study (e.g., determining the connectivity and permeability of the fracture network; determining strength of a fractured rock mass; understanding paleo-stress conditions), these attributes could include the number of fracture sets; orientations, trace lengths, degree of clustering, and aperture of the fracture population in a set; and the topology and intensity of the network (Jolly and Cosgrove, 2003; Lei et al., 2017; Watkins et al., 2015). The presence and amplitude of these biases may also be affected by the study medium. For example, previous work has investigated the operator, used here to describe the person undertaking the interpretation, variability extracting lineament or landform data from remote sensing (e.g. LANDSAT imagery or aerial photographs) (Burns et al., 1976; Burns and Brown, 1978; Huntington and Raiche, 1978); DEMs (Hillier et al., 2015) and LiDAR datasets (Scheiber et al., 2015). Differences in operator interpretations can occur due to: (a) technical factors in data acquisition, for example, band width for Landsat data, image quality for aerial photographs or illumination direction for LiDAR; (b) the scale of observation, for example, 1: 20,000 compared to 1: 5,000; and (c) inter-operator differences (i.e. human factors). Scheiber et al. (Scheiber et al., 2015) found inter-operator replicability to be poor for bedrock lineaments interpreted from airborne LiDAR by six operators'. Significant variability was observed in the number, trace-length and orientation of the reported lineaments. Burns et al. (1976) attributes a difference of 8% in interpretations to 'human factors' for lineaments identified using aerial photography. While differences in inter-operator interpretation has been previously identified, the underlying human factors causing these differences remain unclear. It is also unclear how such factors affect the collection of fracture data either in the field or from field photographs.

In this study, we investigate the magnitude and source of subjective uncertainty in fracture data collected by linear scanlines, circular scanlines, fracture topology and window sampling. Fracture data were collected from Carboniferous rocks cropping out near Whitley Bay, Northumberland (UK) in two phases: (1) in the field where 7 participants collected fracture data directly from outcrop; and (2) two classroom workshops during which 29 participants with different levels of geological

training and expertise collected fracture data from field photographs. In both the field and classroom, the participants collected fracture data individually and in small groups. We compare the values collected by individual participants for the same sample (scanline, circle, window sample etc). It is the values as reported by the participants rather than the underlying statistics of the measured fracture networks that is the focus of this work. We quantify and compare the scale of subjective uncertainty for each method, and identify "problem areas" or factors that amplify the subjective uncertainty. We consider the effect of the variations due to subjective uncertainty on fracture statistics derived from the data, and propose a number of protocols to limit operator bias in collaborative work.

## 2. Fracture data collection and analysis

Linear scanlines are a quick and relatively simple way of systematically collecting fracture data (Agosta et al., 2010; Bigi et al., 2015; Chesnaux et al., 2009; Guerriero et al., 2011; Ortega et al., 2006; Tóth, 2010). This method was developed in rock engineering for a quantitative description of discontinuities in rock masses (Priest, 1993), and then adopted to describe natural fracture networks (Becker and Gross, 1996; Van Dijk et al., 2000; Newman, 2005; Peacock and Sanderson, 2018). The method involves laying out a tape measure on the outcrop and measuring both the number (N) and the attributes of fractures which intersect the scanline (e.g. orientation, spacing, length above and below the scanline, aperture, type of terminations, filling or mineralization) (Priest, 1993; Priest and Hudson, 1981). To fully sample a fracture network, multiple linear scanlines should be completed with different orientations, and the Terzaghi correction should be applied to reduce orientation bias (Mauldon and Mauldon, 1997; Terzaghi, 1965). The goal is to collect enough data to obtain a statistical distribution for each of the main fracture parameters rather than a mean value. It has been recommended that over 225 fractures should be sampled by the population of linear scanlines for the method to estimate accurately the characteristic of a fracture network (Zeeb et al., 2013).

Circular scanlines provide estimates of fracture attributes based on the number of fractures intersecting a circular scanline, n, and the number of fracture trace endpoints, m, within a circular window (Mauldon et al., 2001; Rohrbaugh et al., 2002). The fracture density, intensity, and an estimate of mean trace length for the scanline can be calculated from the n and m values (Mauldon et al., 2001). To be statistically valid, the number of fracture end points (m) should exceed 30 (Rohrbaugh et al., 2002), however, values between 20 and 30 can also be considered reliable (Procter and Sanderson, 2017). This rule defines the radius of the scanline as a function of fracture density and limits the use of the technique in areas of poor exposure and low-density fracture networks. A circular scanline is a maximum likelihood estimator (Lyman, 2003) and does not suffer from the same orientation biases observed in linear scanlines (Mauldon et al., 2001). Circular scanlines are ideal for rock masses with evenly distributed fracture attributes, but may need to be combined with other methods to give a true representation of the heterogeneity of the fracture network (Watkins et al., 2015).

Fracture topology describes a fault or fracture network as a series of branches and nodes (Manzocchi, 2002; Sanderson et al., 2018; Procter and Sanderson, 2017; Sanderson and Nixon, 2015; Laubach et al., 2018). A branch is a fracture trace with a node at each end that can be classified as terminating into rock at i-nodes (unconnected terminations), abutting against another fracture at a y-node, or crossing another branch at an x-node. Topology may be combined with circular scanlines by assessing the nodes present within the circular window and using the sum of i- and y- nodes as the number of trace end points (m-value) in the circle (Procter and Sanderson, 2017). The relative frequencies of different node types (i, y and x) can be plotted on a triangular diagram for the purposes of characterizing and quantifying the connectivity of a fracture network (Manzocchi, 2002; Sanderson and Nixon, 2015).

Finally, window sampling is a technique where all fractures within a given sample area (window) are traced out either by hand, or on a computer, and the resulting traces used to calculate the fracture statistics (Pahl, 1981; Priest, 1993; Wu and D. Pollard, 1995). This technique is often utilised to analyse remote-sampling data such as aerial photographs (Healy et al.,

2017), Unmanned Arial Vehicle (UAV) images (Salvini et al., 2017), bathymetry (Nixon et al., 2012), or satellite imagery (Koike et al., 1998), as well as in outcrop studies (Belayneh et al., 2009). It has been suggested that a minimum of 110 fractures need to be sampled to statistically describe the fracture network using window sampling (Zeeb et al., 2013).

Using these four methods, fracture parameters can be collected to calculate key fracture statistics, for example, trace *length*
(mean and distributions), fracture abundance (*Intensity* and *Density*), and connectivity (Summarised in Table 1).

       Trace length, and trace length distribution are key fracture parameters for DFN simulations (e.g., in simulating fracture-hosted fluid flow. Trace lengths may be measured directly with the linear scanlines and widow sampling, or estimated using the circular scanline method). Challenges to determining the trace lengths of individual fractures include: the scale of observation used to collect the data (Zeeb et al., 2013); classification of fracture intersections (Ortega and Marrett, 2000); and
the fracture fill properties (Olson et al., 2009). Mean trace length is a commonly used fracture statistic and is useful where the fractures in a network are evenly distributed (Mauldon et al., 2001). However fracture modelling typically uses a statistical distribution representative of the fracture length population rather than the mean (Neuman, 1993). Trace length distribution, obtained from measuring individual fractures, should be used when investigating sub-surface fluid flow or characterising spatial variations in fracture trace length (Watkins et al., 2015). We investigate the impact of subjective bias on mean trace
length for all four methods, including the range of reported trace lengths for linear scanlines and window sampling and trace length distribution for window sampling.

       The characterisation of fracture networks and comparison of techniques is greatly confounded by inconsistencies in terminology. Because fractures may be sampled using techniques which are either 1-dimensional (scanlines, boreholes), 2-dimensional (maps, surface exposure), or 3-dimensional (rock volumes), numerous different methodologies and terminology
have arisen to characterise the abundance of fractures in a network. One of the most widely used methods to characterise a network is to define the number of fractures (N) normalised to line length (L), sample area (A) or sample volume (V) depending on the dimension of sampling. In the literature, this statistic is either termed *fracture intensity (I)* or *fracture frequency (f)* (Sanderson and Nixon, 2015). For linear scanlines, *fracture spacing* can be regarded as the inverse of *fracture intensity* for a single set of sub-parallel fractures (Sanderson and Nixon, 2015). Fracture abundance within a network may also be expressed
as the total trace length per unit area (Dershowitz and Einstein, 1988; Rohrbaugh et al., 2002). This statistic is either termed *fracture intensity* (Sanderson and Nixon, 2015) or *fracture density* (Nixon et al., 2012; Zeeb et al., 2013). One attempt to simplify the use of terms is to use the $P_{xy}$ terminology as defined by (Dershowitz and Einstein, 1988) where x denotes the dimension of the sampling region (1 = line, 2 = area, 3 = volume) and y donates the dimension of the feature (0 = number, 1 = length, 2 = area, 3 = volume). For the purposes of our study, we use the term *fracture intensity* (I) to refer to number of fractures
30  per line length (P10, for linear scanlines) or fracture length per unit area (P21, for circular scanlines), and we use *fracture density* for number of fractures per unit area (P20) (Table 1).

       It is also important to understand how individual fractures relate to each other; particularly how the individual fractures connect, and hence contribute to the strength or fluid flow through the rock mass. The number of connections on a fracture trace ($C_L$) is a commonly used measure of connectivity (e.g. Manzocchi, 2002). However, a fracture network consisting
of only y and x nodes could have different $C_L$ values depending on the fracture intensity (Sanderson and Nixon, 2015). It has been suggested that it is better to either consider the average number of connections per branch ($C_B$) (Ortega and Marrett, 2000) or the proportion of connected nodes (Pc) (Sanderson and Nixon, 2015). In our study, we use the proportion of connected nodes for circular scanline and window sampling. To measure connectivity in linear scanlines, the percentage of connected fracture trace ends is reported (Table 1).

## 3. Study methods

### 3.1. Study area

The field site is located in the Northumberland Basin, just north of Whitley Bay, NE England (Fig. 1). The Northumberland Basin is a 50 km wide, ENE-WSW trending half-graben formed during mid-late Carboniferous extensional reactivation of the underlying Iapetus Suture (Chadwick et al., 1995; Johnson, 1984). The stratigraphy consists of thinly (cm - dm) bedded sandstones, siltstones, shales, seat earth, and coals of the Middle Coal Measures (Westphalian B). At the field site the easily accessible and well exposed wave-cut platform clearly exhibits two sets of faults and sub-vertical joints (>75°) which trend E-W to NE-SW and N-S respectively.

### 3.2 Fracture data collection procedure

Six linear scanlines were set up by laying out a tape measure on sandstone beds, both in map and cliff section (Fig. 1C). Participants were asked to identify for each fracture: a) the intersection distance along the tape and b) the length and termination (into rock, abutting against another fracture or not seen/obscured) of the fracture either side of the tape. Eight circular scanlines were drawn with chalk directly onto the sub-horizontal bedding planes of three separate, decimetre thick, medium grained sandstone beds (Fig. 1D). The location and radius for all circular scanlines, apart from C6, were selected by the lead author (Participant G/11) in order to represent what they believed to include a statistically significant number of fracture terminations (i.e. m <30; Table 2). C6 was selected by Participant F.

A N-arrow and NS/EW lines were drawn onto the circle to aid observation. Participants counted the number of intersections with the circumference (n). Following the methodology of Procter and Sanderson (2017), participants were asked to identify the number of i-, y- and x- nodes within the circles. Finally, window sampling was conducted by tracing out the fracture networks on photographs of the circular scanlines in the workshops. Our study did not aim to collect sufficient fractures to represent the fracture network at the field site, and the tested scanlines were not designed to be statistically representative.

Fieldwork was undertaken by 7 participants (labelled A-G) in July 2018 with fracture data collected using field notebooks from 7 circular and 4 linear scanlines (Table 2). There was no particular guidance as to how the participants collected the scanline data, but no more than one person or one group collected fracture data from a scanline at any one time, so as to avoid influencing the data collected by other participants. For the same reason, participants did not annotate or disturb the rock or scanline. Orientation and aperture data were also measured in the field, but they are not included in this study because they generally are not included in circular scanline methods and cannot be measured from field photographs in the workshops. Three of the fieldwork participants also completed the workshop tasks (Participant C = Participant 8; Participant D = Participant 10; Participant G = Participant 11).

Workshop 1 (WS1) was held in September 2018 in Glasgow, with 11 participants (labelled P1-11). Workshop 2 (WS2) was held in October 2018 in Rome with 18 participants (P12-29). Participants were recruited from the authors' research groups (the Faults and Fluid Flow research group within the Centre for Ground Engineering & Energy Geosciences at the University of Strathclyde and the Tectonics and Fluid Chemistry Lab of Earth Science Dept. at Sapienza) as well as colleagues from their departments: participation was voluntary and all data were anonymised for analysis. Each 2-part workshop lasted 3 hours. In the first part, participants worked individually to complete 3 circular and 1 linear scanline, and in the second part, worked in small groups to complete 2 circular and 1 linear scanline (Table 2). Participants were provided with A3 (29.7 x 42.0cm) colour photographs of the scanlines. WS1 participants were encouraged to annotate these with the observed fracture intersections and interpreted termination type, whereas WS2 participants were specifically asked to trace out the interpreted fracture network (i.e. to undertake window sampling). Both workshops enabled us to investigate the impact of subjective bias, however, the fracture maps from WS2 enabled us to examine the impact on window sampling along with investigating the root cause of differences for participant classification of nodes.

To examine the effect of geological experience on subjective uncertainty, participants were asked to indicate their level of geological training, familiarity with geological fieldwork, and their level of experience collecting fracture data (summarised in Table 3, questionnaire provided in Supplementary Information, S1). In the workshops, a small number of participants (Participants 2, 5, 24 and 28) consistently reported anomalously high n-values compared to the node counts. Three of these participants (Participants 2, 5 and 28) had no formal geological training or experience in geological fieldwork and fracture data collection. It is possible that these participants only considered fractures that intersected the edge of the circle in their interpretation, neglecting fractures within the circle that do not intersect the circumference, and introducing a different source of subjective error.

### 3.3 Post-workshop analysis

For the workshop data, we digitised the interpreted fracture traces and node classification for all participants who traced the networks (see Table 2) using ArcGIS. Individual fracture trace lengths for all scanlines, and the distance along the scanline that each fracture intersected linear scanlines were exported as 'Arcmap unit' lengths. These lengths were then scaled to the field to enable comparison of the fracture statistics. In some cases, the counts of n or node types reported by participants differed from the count indicated on the worksheet (see S7). In these cases, to be consistent with field-data collection, we take the value reported by the participant. Digitised networks from Circle 8 were used as a case example to (a) construct heat maps of point density for n, i, y-, x- nodes, and line density for fracture traces, and (b) identify areas within the circular scanline with the greatest variability in the identification and quantification of fracture characteristics such as trace, node type, termination etc.

Fracture statistics were calculated for the data populations from the different fracture characteristics that were measured or counted, and then were then investigated as a function of the field and workshop participants. We report on the impact of subjective bias for the following fracture statistics: *fracture intensity (I), fracture density (d),* the *connectivity* of the network *(Pc & Pf), mean trace length (Tl),* and *trace length distributions (tl).* Statistics are calculated using the equations outlined in Table 1.

In theory, each of the scanlines have a 'true' value for each of the fracture parameters (number and type of fracture intersections and terminations, i.e. n, $N_i$, $N_y$ and $N_x$). In this paper, we are not interested in defining that 'true' value, rather we wish to explore the ranges in reported values from different participants, showing the scale of subjective bias for the collected data, and the factors that affect this range. Therefore, we define the uncertainty, or level of variability, present in fracture data collection and the related statistics as a function of the observers/operators.

### 3.4 Analytical framework

We describe the quantitative fracture data that the participants collected using the following approaches:

**Spatial distribution and node triangle space:** Several fracture attributes are determined by the spatial distribution of features, e.g. fracture traces, within a sample area. For linear scanlines, we visually determine the relative location of interpreted fracture traces from the digitised data. For circular scanlines, the spatial distribution of nodes is represented via point density heat maps, generated from digitised data in ArcGIS, and used to identify areas of uncertainty. We also visually compare the participants' interpretation using node triangle plots. For example, for all circular scanlines, we compare the relative position of node data interpreted for each participant.

**Range/variability:** The spread of data is described using the range between the minimum and maximum value for a given parameter or statistic (e.g. fracture count), and the quartile-based coefficient of variance (QCV, Equation 1).

$$QCV = \frac{Q3-Q1}{Q2}$$ Eq. 1

QCV is interpreted in a similar manner to the standard coefficient of variation (CV) and provides a dimensionless measure of variability which can be used to compare between scanlines and attributes. QCV is more appropriate than the standard CV for this study because much of the data do not display a normal distribution. Further, the median and IQR are less susceptible to being skewed by outliers. We describe variability using the following descriptors: *very low* (QCV = 0.00 to 0.10), *low* (QCV 0.11 to 0.25), *moderate* (0.26 to 0.50), *large/high* (QCV = 0.51 to 0.70), *very large/high* (QVC = 0.71 to 1.00) and *extreme* (QCV >1.01).

**Co-variance:** We describe the strength of the relationship between quantitative data (e.g. fracture count and time taken) using the linear coefficient of correlation ($R^2$). Trends are described using the following descriptors: *no* ($R^2$ <0.35), *very weak* ($R^2$ 0.35 to 0.50), *weak* ($R^2$ 0.51 to 0.70), *moderate* ($R^2$ 0.70 to 0.9) and *strong* ($R^2$ >0.90).

**Consistency:** Consistency can be used to describe two different aspects of the data. First, it can describe the rank position of participants for a specific reported (e.g. n-point count) or calculated (e.g. fracture intensity) value across all scanlines. In this case, high consistency would describe a participant that remains within 3 rank positions for a reported or calculated value for all circles. In contrast, low consistency would describe a participant who ranks highly in once scanline and low on another. Consistency uses descriptors depending on the range in rank position across scanlines as follows: *no* (> 16 rank positions for individual and > 6 for group exercises), *low* (15 to 11 rank positions for individual and 4 to 6 for group exercises), *moderate* (7 to 10 rank positions for individual and 2 to 4 for group exercises) and *high* (< 7 rank positions for individual and < 2 for group exercises). Consistency is also used to describe the range/variability, quantitative data or visual assessments across all scanlines within a method.

For **qualitative data**, such as the degree of experience of collecting fracture data, statistical interrogation is not appropriate, given the potential for ambiguity in the response categories; the categories are not necessarily linear, and participants may judge "high", "moderate" and "low" differently. Instead, we visually interpret trends in qualitative data, and use numerical indicators, such as the range or median, to interpret trends across participant responses and their interpretation.

## 4. Results

### 4.1 Linear Scanlines

The results of statistical analysis of fracture data collected from linear scanlines are shown in Table 4. The range in the number of fractures interpreted to intersect the scanline varied between participants and between scanlines both in the workshop and the field. For example, in the field, QCV ranges from 0.03 for Line 4 to 0.71 for Line 1 (Table 4). The variability in the trace length data depended on the scanline being sampled, more so than which participant was sampling, and could be as low as 0.15 (L1) or as high as 0.82 (L5, WS1). We find that there is greater variability in the minimum recorded trace length (high to extreme), than the maximum recorded trace length (moderate to high). For example, for Line 6, participants reported minimum trace length ranging from 0.02 to 0.23, and maximum trace lengths ranging from 0.25 to 0.72 m (See S5). It is clear that the interpretations by participants differed about individual fracture terminations. For example, for one fracture intersecting Line 3, Participants G + F interpreted that after 8.0 meters the fracture terminated against another fracture, whereas Participants C + D felt that it terminated in an area of no exposure after 22.0 m (S5). The correlation between the number of fractures intersecting a linear scanline and the range of reported fracture trace lengths by participants for that scanline shows weak to no trend in the field (e.g. $R^2 = 0.59$ for Line 1) and no trend in the workshop (e.g. $R^2 = 0.24$ for Line 1). That is, our results indicate that trace length is not correlated to the number of interpreted fractures.

The fracture traces drawn onto photographs in the workshops helped us to understand the underlying controls on differences in interpretation. We examined the fracture traces of Line 6 in detail and the interpreted fracture networks can be considerably different (Fig. 2). All participants identified two large fractures located roughly 1/3 and 2/3 of the way along Line 6, however participants differed greatly in their interpretations of the first third of the scanline: Participant 28 does not identify

any fractures, whereas Participants 10 and 14 identified 3 and 10, fractures respectively. Such differences between participants' observations could be a function of the site; the fractures are partly obscured by water and have thin fracture traces. These 'hairline' fractures are also present in other parts of the scanline and in all cases increase the observation variation between participants. Also in Line 6, a feature trending at a low angle to the scanline half way along was only identified by 14 of 29 (48%) participants. Where this feature is identified, it is also the longest visible fracture trace that transects the scanline, and so identifying this fracture affects the trace length statistics. Our analysis suggests that the main source of uncertainty for characterizing fractures along photographs of linear scanline is the decision of how a fracture terminates, and hence how long the fracture is interpreted to be.

## 4.2 Circular Scanlines: Topological sampling and fracture mapping

We present the results of circular scanlines and topological scanlines together because participants defined nodes within sample circles for both sets of measurements. For the circular scanlines, the number of fracture terminations (m), although not explicitly discussed in this section, is equivalent to the total number of i- and y-nodes.

The reported values for n-points and topological characterisation for circles undertaken in the field are presented in Figure 3. The number of fracture intersections with the edge of a circle (n) displayed very low to low variability as recorded by the field participants (QCV ranged from 0.05 to 0.19; S7). However, there is greater spread in the number of reported nodes identified within a circle. The scale of variance depends on the properties of the circle that is being sampled; variance ranged from very low for Circle 1 (QCV = 0.03) to high for Circle 6 (QCV = 0.62). All node types (i-, y- and x-nodes) displayed a wide spread in variability, ranging from low to extreme across different circles.

Similar reporting behaviour is observed for data collected in workshops, however, the workshop data is even more variable than field data (Fig. 4; Table 5). However, when particularly large variability was observed for a topological parameter (e.g. y-nodes), it was not necessarily replicated for the counts of other parameters (e.g. n-points) for the same circle. For example, the number of y-nodes interpreted in the field varied greatly for Circle 6 (7 to 27; QCV = 0.66), even though this circle had the smallest range in values for n-points (6 to 9; QCV = 0.19). In this case, clearly the participants saw almost the same fracture *intersections with* Circle 6 (i.e. subjective bias for n-points is small). At the same time, the participants differed in their observations and classifications of fracture characteristics *within* the circle, leading to a greater range in the number of fracture intersections there. The consistent observation is that subjective bias affects node counts more than n-point counts, but that the degree of variability is dependent on the sample site – i.e. the characteristics of the circle being sampled.

No single circular scanline was particularly prone to subjective bias for all of the studied fracture parameters. For example, compared to other circular scanlines, the variability in data collected from Circle 3 is small for n-points and y-nodes, but is one of the most variable for i-nodes and shows moderate variability for x-nodes. In contrast, the variability in data collected from Circle 7 is small for n-points, but displays high variability in y-nodes, very high in i-nodes and extreme in x-nodes (Table 5). The trends are seen in both field and workshop data.

Although individual circles displayed considerable variability between participants, many participants remained consistant in their observations between different circles (Fig. 3 and 4). For example, Participants A and C, or Participant 2, tended to report lesser counts for all circles than Participant G, or Participant 13. That said, when Participants C and D repeated the data collection exercise for the same scanline in the field, there were differences within the repeat data (Fig. 3), although it was far less than the discrepancy between participants. The level of consistency depends both on the participant and attribute being measured. For example, for circles undertaken in the workshops by individual participants, node count displays a high degree of consistency (6.6), whereas n-point count displays moderate consistency (9.7). When individual participants are inspected, the level of consistency between scanlines ranged from 1 (Participant's 2, 3 and 13) to 19 (Participant 9). It is clear that some participants displayed a greater level of consistency (e.g. Participant 28), while other participants' interpretations varied from one circle to another (e.g. Participant 9). The relative proportion of specific node classification (e.g. y-nodes)

remained consistent between circles (Fig. 5). For example, Participant 11 consistently recorded more y-nodes when compared to other participants, while Participants 5 and 21 tended to record more i- and x- nodes. The same trends are seen both in field data and workshop data collected as groups.

In general, the scale of uncertainty (the range in reported values) in the workshop data is greater than field data as indicated by a wider range in reported values and higher QCV. Overall, the number of fractures reported was larger in the field data than the workshop data. For example, the reported number of fracture intersections in Circle 3 in the field (Fig. 3) ranged from 19 (Participant C) to 30 (Participant B), whereas from the workshops ranged from 14 (Group 8) to 23 (Group 6) (Fig. 4). Similarly, the number of y-nodes is generally greater in the field and the range in values for each circle is less extreme − e.g. in the number of y-nodes for Circle 5 ranged from 28 (Participant C) to 47 (Participant D) in the field (QCV = 0.38; Fig. 3C), and from 4 (Participant, P2) to 41 (P13) in the workshops (QCV = 0.81; Fig 4). It is possible that in the field participants could observe fractures in more detail (e.g. the hairline fractures in Fig. 2) resulting in more consistency in their reported values.

In our data there was a clear discrepancy between the number of nodes or n-points *reported* by participants during the workshops and the number *recorded* in the paper copies of interpreted circular or linear scanlines. Participants tended to report a smaller number of nodes or n-points than they had drawn on their worksheets. While the magnitude of this error varied both between participants and between scanlines, the differences were consistently higher for data collected within an area (i.e. node counting) compared to that collected along a sample line (i.e. n-points). This counting error was much more pronounced within the circle than around the edge, suggesting that as data gatherers we are relatively good at counting when we follow a sample line (e.g. edge of a circle or linear scanline). However, when counting within a sample area the accuracy of results is reduced.

## 4.3 Window sampling

For window sampling, the number of recorded fractures displayed moderate to high variability (Table 6), with the largest variation occurring for Circle 4 (11 to 29; QCV = 0.76). The maximum trace length reported by all participants remained fairly consistent (QCV ranging from 0.01 for Circle 8 to 0.29 for Circle 1). However, considerable variability in trace length distributions was observed between participants (Fig. 6), with the number of small fractures recorded across all scanlines displaying the most variability. For example, the number of fractures below 0.2 m recorded for Circle 8 ranged from 7 to 41, which represents 36.8% and 75.9% of the reported fractures for both participants. This is also seen in the minimum reported trace length data, which displayed very high to extreme variability (e.g. 0.02 to 0.11 m for Circle 4; QCV = 0.94). While the number of small fractures recorded by participants varies between circles, whether a participant records a high or low relative percentage of small fractures remains consistent. For Circles 8, 5 and 1, Participant 3 consistently recorded a high percentage of small fractures, whereas Participant 24 consistently recorded a low percentage of small fractures (Fig. 6a). In short, participants either consistently record the presence of small fractures in a network, or consistently do not record the existence of small fractures in a network. For trace lengths longer than about 15-20% of the diameter of the circle, the shape of the distributions remains consistent across all participants, indicating that the larger traces in the fracture network are consistently identified independent of participant (Fig. 6).

## 4.5 Areas of increased uncertainty: A case study using Circle 8

To highlight potential causes of differences in interpretation, Fig. 7d compares the interpretations of fracture traces and nodes in three particular 'problem areas' (so called owing to how differently these parts were interpreted), from end-member Participants 11, 18 and 21, who reported high, medium, and low node counts respectively. Area 1 is well exposed and contains several intersecting fractures. The nature of the connections was interpreted differently by each participant. Participant 21 interpreted only the major fractures coming into the junction, and depicted the fractures interesting in a star-like formation. Participant 18 interpreted a standard x-node, with a second larger fracture terminating against the NE-SW trending fracture (y-

node), and also notes an E-W trending fracture linking the two major fractures and cutting the third (three x-nodes). Participant 11 differed from Participant 18 by interpreting the NE-SW fracture trace as being offset by the NW-SE fracture, such that the x-node interpreted by Participants 21 and 18, was instead interpreted as two y-nodes. Area 2 is a complex intersection of a number of NW-SE fractures with part of the photographed exposure obscured by shadow (a clear limitation of interpreting the scanline from photographs rather than in the field). Participant 21 did not interpret the fractures obscured by shadow, whereas Participant 18 did. Participant 11 depicted a number of smaller fractures which Participants 18 and 21 did not identify. Area 3 is an intersection of two large fractures which is obscured by a coarse sand infill. Both Participant 18 and11 interpreted the obscured connection as a simple x-node, whereas Participant 21 felt that the fracture bifurcated to frame the area of no exposure. Participant 18 and 21 interpreted the other fully exposed connections similarly (although Participant 21 does not depict a fracture to the south of the sand fill), whereas once again Participant 11 identifies several additional smaller and complicated fractures and fracture connections, particularly y-nodes. In each case, it appears that participants effectively 'self-censored' their data according to their 'preferred' minimum trace length, and had different approaches to areas of shadow or obscured outcrop. The different geometry of the interpreted fracture intersections would result in significant differences in interpreted fracture development history.

When analysing the node classifications and interpreted trace lengths for all circles it was found that in many cases the fracture networks depicted or interpreted were not viable: in other words, there were undefined nodes or intersections that had a non-compatible number of branches entering the node (e.g. 4 nodes for a y-node or 5 for an x-node). Occurrences of these undefined or floating nodes were more common in WS1 than WS2, perhaps because WS2 participants were specifically asked to draw out the fracture network on their photographs.

## 4.6 The effect of working in groups

Large variability in the number of reported fractures in the field was also seen when linear scanlines were undertaken as pairs, for example for linear scanline 3 counts ranged from 21 (Participant C + D) up to 30 (Participant A + B). The groups are obviously made up of participants who have different 'eye for detail'. When working individually, Participants C and D both recorded small fracture counts, while Participant B recorded the highest. There is a suggestion in the data that when working as pairs, groups tended towards the more detailed member, for example Participant F recorded the lowest fracture count when working individually, however, in a group with Participant G recorded a higher than average fracture count. This was also discussed in the discussion following workshop 1.

No clear differences can be seen between data collected individually or as groups for either circular scanlines (Table 5; Fig.4) or window sampling (Table 6; Fig. 6b). Although the group circles have smaller y-counts and greater mean trace length values, the differences are not enough to be confident that the effects are due to working in groups rather than differences in the fracture network. This is due to the limited number of circles completed, the fact no circles which were completed individually were completed as a group and that there is a large spread in variability between participants observed between different circles. That said, groups generally reported more complex fracture networks with a higher reported number of small fractures. When working as groups that included a naturally detailed and naturally less detailed participant, the results tended to be more detailed: compare participants 2 and 11's recorded values when working individually or together as Group 3 (S7).

There is also no difference in the level of variability for any particular parameter reported for either topological sampling within a circular scanline or window sampling (e.g. y-node count, number of fractures etc.). For example, node counts display QCV values of 0.48 to 1.00 for individual circles and 0.40 to 1.00 for group circles. This suggests that working as a group does not affect the level of subjective bias in the dataset. Similarly, to when working individually, the majority of groups show high levels of internal consistency in the number of reported fractures (7 out of 12 groups). Groups also displayed internal consistency in the relative percentage of small fractures (Fig. 5b) and node types (Fig. 4b) reported across different sample circles.

**4.7 Time taken to collect data**

In the field, the time taken to undertake counts of n-points and nodes varied not only as a function of participant, but also the circle being sampled. It was not clear if it took longer for participants to count more n-points or nodes, with the trend being non-existent to very weak for n-points ($R^2$ ranging from 0.003 to 0.37) and non-existent to weak for nodes ($R^2$ ranging from 0.04 to 0.70). For workshop data no trend was observed between the time taken to record, or the variability in, the number of reported fractures observed (Fig. 8a). Both the time taken and magnitude of the variability was considerably greater in the workshops compared to the field. For example, Circle 5 took participants between 1 and 17 minutes in the workshop (QCV = 0.94), and 2 minutes 21 seconds to 4 minutes 26 (QCV = 0.64) seconds in the field.

Window sampling, which was undertaken in WS2, took longer than circular scanlines for the same circle in WS1, however, this difference is small. While it took 1.3 to 3.2 times as long to record n values, the relative time taken to undertake topological sampling within the circle is comparable for circles completed both as individuals (0.85 to 1.6) and in groups (0.95 to 1.05). Thus, although circular scanlines are often suggested as a quick way of gathering fracture data, it does not take significantly longer to trace out the fracture network. This observation suggests a similar amount of data could be collected using both methods.

While some participants took much longer than others, the participants were often (18 out of 29 Participants) internally consistent in the time taken to complete their tasks (Fig. 3 & 8). For example, C and G tended to take longer than A or D in the field, and in Workshop 2, Participant 29 consistently took longer than Participant 25. Although this was often observed, some participants displayed low to no consistency in time taken between scanlines. For example, Participant 25 ranked 3rd quickest for Circle 8 and 28th quickest for Circle 1 in the time taken to count n-points. No correlation was found between average rank position and range in rank position for the time taken to either recorded n-point data ($R^2 = 0.025$) or node data ($R^2 = 0.001$).

**4.8 Experience**

The relationship between experience and the number of node counts has a large amount of scatter (Fig. 8b). Generally, participants with less experience undertaking geological field work or collecting fracture data counted fewer nodes than more experienced participants, however the trend is very weak. Perhaps counter-intuitively, experience does not reduce the time taken to collect fracture data (Fig. 8b). However, for node counts, the fastest experts are still notably slower than the fastest inexperienced Participant. Also, more experienced participants do not appear to characterise with more detail than those with less geological training or experience. It is possible that participants with experience in fracture analysis will consider the connections they observe, whereas beginners will draw the traces that they see without considering the implications of those connections (i.e. implied cross-cutting relationships).

**5. Effect of subjective bias on the derived fracture statistics**

The variability in the collected fracture parameters will affect the derived fracture statistics in different ways. No particular equation for the calculated statistics (Table 1) has statistically sensitive relationship to subjective bias for a particular fracture attribute. To identify which fracture statistics are most susceptible to subjective bias, we discuss and compare the results from all methods in terms of the relative ranges of values.

The effect of subjective bias on *mean trace length* depends on the method that the statistic is being derived from. For linear scanlines the variability depends on the scanline being sampled. For example, small variability is seen for Line 2 where values range from 0.33 to 0.49 m (QCV = 0.17), compared to 0.89 to 3.70 m (QCV = 0.61) in Line 4. For topological sampling within a circular scanline low to very high variability is observed between participants in the field, with QCV ranging from 0.13 for Circle 3 to 0.82 in Circle 7. Variability is higher in workshop data, where moderate to high QCV values are observed

(0.34 to 0.72), with both group circles displaying moderate variability (0.34 and 0.38). Mean trace length derived from window sampling displays moderate variably across all Circles sampled (QCV 0.26 to 0.47) and displayed lower variability compared to trace length derived for the same circle using topological sampling. Mean trace length derived from window sampling was consistently less than that derived from circular scanlines of the same circle. For example, mean trace length for Circle 5 derived from window sampling ranged from 0.19 to 0.46 m (S8).

For linear scanlines, no correlation was observed between the number of observed fractures and fracture trace length. For example, Participants B and G both recorded 10 fractures intersecting Line 1, however, the derived mean trace lengths were 0.62 m and 0.25 m respectively (see, S5). This outcome contrasts with window sampling, where mean trace length decreases as fracture count increases ($R^2 = 0.79$ for Circle 8, see S8), and circular scanlines where mean trace length is a function of the number of fractures intersecting and terminating within a circle.

*Fracture density,* which is calculated for circular scanlines and window sampling has moderate to high variability between participants. For both methods the level of variability depended on the circle being sampled, along with whether the analysis was undertaken in the field or in the workshops. For example, fracture density derived from circular scanlines ranged from 3.82 to 7.48 F/A for Circle 3 (QCV = 0.13) up to 2.12 to 10.6 F/A for Circle 6 (QCV = 0.68) in the field and from 2.07 to 12.1 F/A for Circle 3 (QCV = 0.34) up to 0.48 to 6.53 F/A for Circle 1 (QCV = 0.79). For window sampling participant's statistics displayed moderate to very high variability within circles (QCV 0.44-0.76). A larger value for fracture density is obtained using window sampling is used for the same circle, as shown in Circle 8, where window sampling derived fracture density ranged from 22.9 to 68.8 F/A compared to 1.9 to 41.4 for circular scanlines. Variability between participants is lower for window sampling compared to circular scanlines when samples are undertaken individually, however, show more variability when undertaken as a group.

Across all methods, *fracture intensity* has the smallest amount of variability between participants, however, differences are still observed between methods. When linear scanlines are used the amount of variability depends on the scanline being sampled. For example, Line 4 ranges from 0.93 to 0.98 f/m (QCV = 0.03) whereas Line 1 ranges from 2.31 to 7.69 f/m QCV = 0.71), with the majority of scanlines displaying low to moderate variability. When fracture spacing, instead of number of fracture reported, is used to calculate fracture intensity more variability in values is observed, primarily due to the large difference in the minimum reported fracture spacing of participants across all circles. Unlike for linear scanlines, fracture intensity represents a robust statistic for both circular scanlines and window sampling. This is emphasised by the QCV values for circular scanlines, both in the field (QCV 0.03 to 0.21) and workshop (0.19 to 0.43), along with those for window sampling (0.11 to 0.21). Fracture intensity estimates using circular scanlines derived from field data generally provide a higher value than when the same circle is analysed in the workshop. For example, Circle 3 ranges from 4.75 to 7.5 f/m from field data and 3.5 to 5.75 from workshop data. Fracture intensity derived from window sampling is consistently lower than that derived from circular scanlines for the same circle.

The *connectivity* of the network (*percentage of connected fractures, Pf*) is highly variable for values gathered by participants using linear scanlines, with the magnitude of the variability dependent on the scanline being sampled. For circular scanlines and window sampling, where the percentage of connected branches (*Pc*) is used, connectivity represents a robust statistic with very low QCV values (e.g. 0.00 to 0.06 for field data). The maximum reported values for Pc remain the same when field and workshop data are compared, however, the lowest reported values are consistently lower in the workshops for any given circle.

Subjective bias impacts all data collection methods (Table 7). Window sampling appears to be is the method which is least effected by subjective bias. Out of the methods tested in the workshops, window sampling displays the lowest variability between participants for all of the parameters: intensity (low), density (moderate to high), mean trace length (moderate) and connectivity (very low). Additionally, because this method requires the network to be drawn out, it is possible to check for the

existence of 'floating nodes' and other irregularities in the recorded fracture network. Linear scanlines had the greatest variability between parameters.

The different fracture statistics also display different degrees of subjective bias. *Fracture intensity* represents the most robust statistic as it shows the least variability in data collected by different participants for a given scanline. In contrast, *mean trace length* and *fracture density* both display considerable variability in the reported data, particularly when derived from workshop data. The *connectivity* of the network was found to be robust for topological sampling, however, considerable variability existed in the values reported from linear scanlines. When participants traced out fractures while completing linear scanlines or window samples, it was possible for us to identify the causes of differences in participant observations; differences that affect the derived fracture statistics.

## 6. Discussion

Subjective bias in fracture data collection has implications for the validity or reliability of the models that the data informs, such as the derived fluid flow parameters, rock strength characteristics or paleostress conditions. Here, we explore these implications. Further, we draw on participants' discussions during the workshop and field activities to propose potential reasons for the differences in observations between participants.

## 6.1 Scanline validity and appropriate data collection method

As for all forms of sampling for data collection, scanlines must contain enough data points to be statistically valid, where the required number of data points depends on investigated characteristic of the fracture network. However, our data demonstrate that in addition to the fracture network characteristics, the required scanline size (length of a linear scanline, circumference of a circular scanline or area of a window sample) is also dependent on who is collecting the data.

Different participants clearly observed different numbers of fractures in the same scanline (Table 6, Fig. 2). Zeeb et al. (2013) suggest that a minimum of 225 fractures are sampled for linear scanlines and 110 fractures for window sampling. For Line 3 participants reported between 1.4 and 2.5 fractures per metre. If we apply Zeeb's recommendations, the cumulative length of scanline for the person who reported a lower number of fractures per metre would need to be nearly twice the length (160m) of the representative scanline for the person reporting higher fracture numbers (90m). The number of fractures in Circle 5 reported for window sampling ranged from 13 to 56, which means between 2 and 9 circles of this size would need to be analysed to statistically represent the network. The variation between how participants view the fractures therefore results in significantly different lengths of scanline or numbers of circles to capture a representative sample of that network. Our data show that there is not a great degree of difference in the time taken by participants to characterise the same fractures network, albeit with different detail. However, the simple fact that one geoscientist needs to find over 4 times more locations to draw out circles of the same radius on a particular outcrop will likely mean that collecting equivalent datasets may take longer for a less detail-oriented participant. Where a detailed-orientated operator may fall down, however, is when a fracture network displays a degree of heterogeneity or clustering. In this case, although a detailed-orientated operator would report the required number of fractures according to Zeeb et al. (2013), they may fail to cover enough ground to understand the spatial distribution of fractures the way a less detail-orientated operator would.

The appropriate *radius* of the sample window is also dependent on the sampling behaviour of the operator. For circular scanlines it is widely agreed that a minimum of 20-30 fracture terminations within a circle is appropriate to derive fracture statistics or undertake topological sampling, and the circle radius must be adjusted to capture enough fractures or fracture terminations (Procter and Sanderson, 2017; Rohrbaugh et al., 2002). Figure 10 shows the proportions of valid (capturing >30 terminations) and invalid (capturing <20 terminations) results for the circular scanlines in this study. Out of the 29 participants that collected data from Circle 8 in the workshops, 12 identified over 30 fractures and so report valid results, another 8 collected

over 20 fractures and their results are potentially valid, whereas 9 valid reported fewer than 20 fractures and so the statistics derived from their sample may be unrepresentative. Since the number of fractures identified in the field is generally higher than in workshops, a greater proportion of field participants reported sufficient terminations within the circle to be statistically valid. For example, all field participants report valid data for Circle 4, whereas only 3 of the 9 groups in the workshops do.

In this work, the location and radius of all scanlines except C6 were selected by Participant G/11, who tended to be more detailed than other participants. This participant recorded enough terminations to class their data as valid for all sampled circles. Therefore, this participant chose a circle radius appropriate to the level of detail to which *they* identify and characterise fractures, but which is not appropriate for other less detailed observers. This effect is demonstrated in Fig. 11, which shows a synthetic fracture set interpreted by an operator who gathered less detail-focused observations (Fig 11a) and an operator who

gathered more detailed information (Fig 11b). A statistically valid circular scanline (>30 fracture terminations) is drawn onto the interpreted network and the resulting differences in the fracture topology and the fracture statistics shown (Table C inset). For this example, for the scanline to be statistically valid, its radius must be 3 times larger for operator (a) than operator (b).

        How detailed a fracture network is interpreted to be by an operator therefore affects the derived fracture statistics (Fig. 11c). The more detail focused interpretation (Fig 11b) has more y-nodes, but similar counts of n, i-nodes and x-nodes.

As a result, the connectivity of the interpreted network in part (b) is greater than that in part (a). The other fracture statistics (intensity, density and trace length) are very different between different levels of interpretation detail. For example, the density of fractures in part (b) is over 18 times larger than that of part (a) and mean trace length reduces from 1.71 m for part (a) to 0.47 m for part (b). This variability is primarily due to the required circle radius, which is used to calculate fracture statistics using circular scanlines (Table 1), changed in order to capture the minimum number of fracture terminations. For our data, if

participants who recorded insufficient fracture terminations in their samples (i.e. less than 20) to be considered statistically valid are disqualified (i.e. removed from the dataset), the maximum trace length and density are more affected by subjective bias than the fracture intensity and connectivity. For example, the calculated maximum trace length for Circle 8 decreases from 2.88 to 0.92 m, and the maximum density for Circle 5 decreases from 46.5 to 12 f/A.

        Different fracture data collection methods are chosen depending on the aims of the study, the way the fracture network

is presented within the outcrop (or core) and the homogeneity of the fracture network. Our data suggest that window sampling is the least effected by subjective bias. In the process of drawing out the fracture network, the operator is required to consider the fracture geometries, evidence for fracture timing (e.g. cross-cutting mineral fill types), and the implications of this for the fracture statistics. There may be similarities with the findings of Macrae et al. (2016), who showed in a randomised controlled trial of industry experts that the quality of a seismic interpretation could be increased by explicitly requesting interpreters of

seismic data to describe the temporal geologic evolution of their interpretation.

### 6.2 Causes of subjective bias: operator bias and fracture network characteristics

        **Human factors:** We observe considerable variability between participant's interpretations, something which has also been observed by Peacock et al. (*in press*) in the reported values of joint intersections on a bedding plane, additionally our data shows individuals display a degree of internal consistency (Fig. 3 and 4). That is, individuals exhibited personal characteristics or traits through the data that they gathered: they were either more detail-orientated, or they were less detail-orientated.–

35 allowing them to focus on gathering a larger volume of data. We suggest that this reflects an operator's personal approach to data collection: variability in data that is collected by a single person is likely to be internally consistent from one data-gathering exercise to the next. Care therefore needs to be taken when comparing results from different operators. Our data shows it is important to consider whether you are working with a 'detailed' person who will likely wish to include data on smaller/more

detailed structures, or if you are working with a person who is more likely to focus on 'the big picture' and to gather a higher volume of data from a greater number of sample locations in the same amount of time.

It is interesting to consider why people tend to be internally consistent in their data gathering approach, yet different from each other. It is likely that they consciously or subconsciously construct their own protocols around how the data should be collected, and what features should or should not be included. These protocols will be shaped by: (a) Practical and physical factors such as the quality of an operator's eyesight, whether or not it is easy for them to repeatedly crouch down to get a closer view and stand up to move around, spatial co-ordination that affects the ease with which they cover the scanline, and the time available to gather the data; (b) Inherent cognitive or personality-related factors.

As an example, some participants focussed only on more pronounced fractures, ignoring, for example, smaller subsidiary fractures, closed or filled fractures, or thin 'hairline' fractures intersecting the scanline. This behaviour was particularly common where a large or clear fracture is present; the participant reports only the dominant feature. As one participant exclaimed during group discussion "*What do these tiny things matter - if you have a massive fracture?*". However, this viewpoint was not shared by all participants: others raised the importance of the spatial distribution of small fractures either as indicators of strain incompatibility, or as the locus of flow at fracture intersections. It is clear that decisions about "what feature counts" and whether a feature has geological origins are subjective judgements. Shipton et al. (*in press*) and Gibson et al. (2016) discuss the concept of Mental Models in the geosciences: a mental model is a simplified internal representations of some external event or process. We suggest that our participant's mental model of the processes that they are measuring may guide their attention to particular features, and so obscure or censor the network that they record. The mental model, and therefore the features – or scale of features - observed, may also be influenced also by the intended application for collected data (Shipton et al., *in press*), or the conceptual model that the participant is working to (Shipley and Tikoff, 2016).

While one may expect mental models be shaped by the experience levels of operators, this is not observed in our dataset. Scheiber et al. (2015) studied different participants' observations from a single LiDAR dataset, and found no correlation between experience and the reported number of bedrock lineaments. Similar to our work, Scheiber et al. (2015) found that participants who reported the largest number of lineaments observed the greatest number of small features, and these small features often did not follow the main orientation trends seen in the data. Biological studies also find no evidence for a relationship between level of experience and the detail or observations (e.g. Dickinson et al., 2010; Dunham et al., 2004).

We suggest that the cognitive style of the participant is more important than experience in how a participant interprets the studied media; the fracture network. Cognitive style refers to how an individual perceives, thinks, solves problems, learns, makes decisions and interacts with others (Witkin and Goodenough, 1977). The work of Carl Jung (2016, *original work published 1924*), particularly the use of the Myers-Briggs Type Indicator (Myers, 1962) to assess cognitive style, underpins much of this field. Jung's theory outlines three facets of cognitive style, each with end-member preferences (Myers et al., 1998): *Perception* dictates whether a person is either meaning-oriented (*intuitive*) or detail-oriented (*sensory); Judgement* dictates whether a person makes decisions based on analytical and logical means (*thinking)* or through a set of personal values (*feeling)*; and *Environment* dictates whether a person reacts to immediate and objective conditions (*extrovert*) or by looking inward to their internal and subjective reactions (*introvert*) when reacting to their environment. On top of these three facets, people often have an innate preference for either *perception* or *judgement* trains of thought such that a perception person has a tendency to use sensing and intuition orientated thought, while judgemental person uses a combination of thinking and feeling. It is well known that cognitive style can have an impact on how people both respond to stimuli and make decisions (Jung, 2016, *original work published 1924*). If a cognitive style is at odds with the task in hand, for example where an intuitive participant is required to undertake a detailed task which would be better suited to a sensory participant, a lower performance is to be expected (Chan, 1996). This has been reported in the case of auditors (Fuller and Kaplan, 2004) and air traffic controllers (Pounds and Bailey, 2001). A 'cogitative culture' is often observed in different professions and roles, where people aim to fit their cogitative style to the task or workplace environment (Armstrong et al., 2012). A misfit between cogitative style and the task tends to be associated with lower performance levels (Chilton et al., 2005).

While cogitative styles may not be clear-cut (e.g. Peterson et al., 2009), it is useful to adopt end-member styles to consider how the cognitive style of the data collector could, in theory, affect the fracture data they collect. For example, a *sensory* participant should show a high attention to detail, often observing small fractures and subtle features of the fracture network that may tend to be missed by intuitive participants. Conversely, while an *intuitive* participant may not record small features, they should update their conceptual model more frequently in response to new observations (e.g. a specific orientation of fracture is consistently mineralised), leading them to develop a more robust conceptual model of the subsurface (Shipley and Tikoff, 2016). A *thinking* participant may collect more consistent or transparent data than participants with other cogitative styles, for example, by developing and applying a set of logical and analytical rules.

The node data collected in our study is most consistently affected by cognitive biases (Figures 3, 4 & 6). Detailed-orientated participants reported a greater number and percentage of y-nodes compared to i- and x-nodes. One of the underlying reasons for this was identified in the workshop discussions, where sensory-type participants described reporting the small fractures at fracture intersections, whereas intuitive-type participants reflected that they did not report these features, since they believed (i.e. interpreted) that they would not contribute to flow. Similarly, jogs in the fracture were classified systematically differently by different participants, where some considered jogs to be the termination a fracture and initiation of another fracture, whereas others considered jogs to be a slight side-step of an otherwise continuous fracture. This would have consistently affected the number of nodes reported.

**Working in groups**: We observed that behaviour varied considerably between groups, and that the behaviour of groups depended on the cogitative styles of individuals within that group (pairs, in most cases). For example, in one group a participant explained *"[when we started working together] I very quickly …realised that [their partner] cares about tiny features, so, together we cared about tiny features…but I was aware that if I was working on my own, I would have done it differently"*. This group evidently consisted of participants with different levels of detail-orientated behaviour, and the participant who individually displayed a less sensory cogitative style tended towards the level of their partner. This is perhaps an example of herding behaviour (c.f. reference), often herding towards a more detailed approach. Another participant reflected *"I didn't find we [their group] were talking about 'does this fracture count?'. Instead, we were discussing whether something was a Y-Y or an X, or where exactly a fracture goes or where it terminates and so on"*. This group appears to be made up of two in intuitive-type participants, and worked well together discussing the meaning behind the fracture network.

The very knowledge that you are working together might be effective in itself. As one participant articulated *"the very knowledge that you are working with someone changes your approach. You want to engage together and so you need to defend or explain your choice, which makes you more alert to what you are doing and why"*. This suggests that for fracture analysis a group comprising of different cognitive styles could be advantageous in terms of capturing the range of perspectives and potential interpretation styles. Fracture network analysis is not simple; it requires not only the identification of fracture traces, but also a consideration of how these fractures traces from a network (Peacock and Sanderson, 2018). Parallels may be drawn to the findings of Cheng et al. (2003), who found that when participants were asked to complete a complex accounting task, groups comprised of different cogitative styles outperformed homogenous groups. That said, working in mixed groups can be a cause of conflict and introduce errors due to a negative effect on the ability to reach a consensus in the decision making process (Aggarwal and Woolley, 2013). In our study, some participants felt that working as a group slowed down the data collection process to a problematic degree. However, this was only observed in WS1; the sampling time was comparable for individuals and groups in WS2 (Fig. 3, Table 5). Interestingly, there are many different interpretations of what 'working together' means, or shapes that working together takes. While for many, this meant working through the scanline together, others elected to divvy-up the window or scanline, working separately and combining their results at the end, or for one person to be the data gatherer, and the other the data recorder (i.e. the scribe). For the latter two models of working, the potential benefits of discursive or deliberative group work (i.e. rationalising and laying bare thought processes) will not be leveraged.

**Projecting into areas of limited exposure**. The effect of subjective bias on the required length of linear scanlines, radius of circular scanlines and area of sample windows will have particular consequences in areas of limited exposure, where a detail-orientated operator may not be able to collect enough data to statistically represent the fracture network. In the discussions following WS1, several participants reflected that where exposure was limited or obscured, they did not attempt to interpret where the fracture went, nor the type of fracture intersection, since this was straying too far from quantitative observation into more qualitative interpretation. Other participants, however, did interpret the network despite these difficulties, which increased the number of nodes that they reported and decreased the number of illogical 'floating' nodes. Clearly some felt it was most appropriate to interpret in the face of uncertainty, so as not to discount nodes that could be logically inferred, while others felt that this would be over-interpretation. Both viewpoints have internally consistant reasoning, but will produce very different outcomes in terms of fracture network characteristics to be applied to analyses of fluid flow or rock strength.

In some cases, these uncertainties could easily be overcome in the field. For example, where a fracture was obscured by shadow or seaweed. Some field participants described 'feeling' for the trace of a fracture with fingers or pencils when obscured (e.g. by seaweed), or difficult to see. Some also describe inferring fracture trace by extrapolating from the exposed traces, triangulated by observing the general fracture trends. Such 'exposure bias' is recognised when studying fault zones, where, by their nature, the fault rocks are often preferentially obscured and therefore good continuous exposures of fault zones are very rare (Shipton et al., *in press*).

**The scale of observation:** In the workshops, participants were provided with a 2D 'birds eye' view of the full circle being sampled. In the field, only the tallest operator will be able to observe the full circle, while all others would have a more limited field of view. In the field, the participant can potentially crouch down and 'get their eye in' to the detail within a complex fracture network. The ability to adjust the scale of observation during data collection in the field is most likely the reason for more nodes reported in the field than in the workshop for the same circular window (Fig. 10). Similarly, it is important that the same scale of observation is maintained when using remote sensing methods. For example, it is important that an operator does not zoom into areas of interest, unless they do so systematically.

**Using pre-set data cut-offs:** It is clear that a meaningful quantitative analysis of fractures requires a certain degree of consistency. This is particularly relevant for combining or comparing data collected by a number of individuals, including for meta-analyses. Participants in WS1 discussed whether their collected data could be more readily compared or combined if a minimum trace length cut-off was applied to the data. However, no consensus could be reached about the scale of or style of the cut-off to be applied because (i) it would not be an accurate representation for flow and/or rock strength; and (ii) more attention should not be paid to simpler, larger, and more isolated structures that could have almost no flow or mechanical significance. The use of size cut-offs has been used in scanline studies which investigate aperture size distribution (e.g. Hooker et al., 2014; Ortega et al., 2006). Fracture trace length however differs from aperture studies in that what you are measuring (the number of fractures) is not a clearly defined parameter (i.e. aperture size) but instead highly subjective. This stems from the fact that most opening mode fractures show evidence of growth through the linkage of several smaller fractures, and, due to the fractal nature of fractures, a single fracture tends to be comprised of several smaller fractures (e.g. Bonnet et al., 2001), and so the fracture count is dependent on the scale of observation. We observe similar effect in our dataset, where participants differ in their interpretations of where a fracture starts and ends, and whether fractures with jogs should be classified as one continuous feature or multiple fractures.

Another knock-on effect of having no data cut-off is that the derived statistics for *fracture intensity* or *fracture density* from reported data can return wildly differing results (Ortega et al., 2006). From our findings, it is clear people 'self-censor' according to a minimum trace length, and this minimum cut-off is variable in scale. That said, we find that the range in reported values decreases towards 10 to 15% of the diameter of a circular scanlines or window sampling. For example, for Circle 8 data (S8), the range in the number of reported fractures is 36, however, when fractures <5 cm trace length are removed the range

falls to 19. The range stabilises if only fractures >10 cm length are considered. This effect is amplified for fracture density, which is calculated using the number of reported fractures. The raw density statistics range by a factor of 3 (23 to 69 f/A), however, as you apply cut-off's to the data the values decrease and converge so that when all fractures less than 10 cm length are removed, the difference between minimum and maximum values reduces to 1.3 (18 to 25 f/A). This suggests that it should be possible, depending on the aim of the study, to apply a cut-off to the minimum trace length included in the dataset, however, it is vital that this approach is reported, otherwise the data reported will not be replicable.

## 6.3 Recommendations for reducing subjective bias

We encourage reflective critique of the fracture data collection process, including identification of potential uncertainties when collecting new data, and when collating or comparing fracture statistics from different field studies. Drawing on our results, we propose the following approaches to assess, reduce, and report the potential subjective bias in the data that geoscientists collect:

*1) Understand your style of data collection:* It is vital that when collecting fracture data, either in the field or from photographs (or e.g. remote sensing), that the 'go to' style of data collection is understood; i.e. detail-orientated vs data volume orientated approaches. In relatively homogenous fracture networks a detailed operator will characterise a network quicker as less circle is required (i.e. detail-orientated will be preferable). In areas of regional heterogeneity, however, it is better to undertake more circles covering larger fractures (i.e. be more data-orientated). Finally, but most importantly it is vital that we report our own biases and methods used to reduce bias in the field reports, to enable replicability and comparison of studies.

*2) Select your fracture data collection methods to limit subjective bias:* While all methods of collecting fracture data are susceptible to subjective bias, we find that window sampling is the least effected. The approach does not take much longer than topology sampling (the time taken is on par with topology sampling when working in groups, and <1.6 times as long as topology sampling when working individually,). Thus, we recommend that, where possible, a window-sampling approach is adopted to collect fracture data. In addition, regardless of which approach is adopted (circular, window, linear), the fracture network should be traced out; either on a printed photograph/tablet or with chalk on the outcrop. Doing so for at least some of the sample windows would allow participants to examine their own biases in how they classify fractures, and critique their collection approach. Since we find that the window radius, to some extent, governs the size of the fractures observed and reported by different individuals, we recommend that, if using circular scanlines, the radius of the circle is kept the same across a sample area since we find that the circle radius, to some extent, governs the size of the fractures observed and reported. However we recognise that this could be problematic in areas of drastically different fracture intensities where a 'valid' circle size for one sample location would not collect valid data at other locations.

*3) Define what fracture features to include early on:* Prior to the collection of field data, or as the first step of field data collection, the sampling strategy should be reflected upon and agreed, in line with the goals of the study and the characteristics of the locality. For example, in fluid flow studies it is vital that information for all *connected* fractures are included in the dataset, in which case, the location of small fractures that contribute to the network becomes key: simply stating this may induce people to focus more on the small features (c.f. Macrae et al., 2016). The spatial distribution, not just the relative percentage, of fracture terminations within a network should be assessed and recorded when reporting fracture statistics. In the case where small fractures may be important, then it is important that all the observed fractures are collected, however, sub-sets based on fracture trace length should be used when comparing data. One could take the approach that everything should be collected and only after collection should the data potentially be censored for the purpose of further analysis (e.g. to investigate the intensity of fractures above a certain trace length). However, not every sampling campaign necessarily needs the same level of detail, and so adopting this approach could lead to the collection of a large amount of unnecessary data as a function of campaign goals. If the level of detail collected is superfluous to the needs of the study, the overall data quality

could suffer in terms of the extent of outcrop studied (i.e. the number of detailed sample windows completed over a given area is less than the number that would have been completed if the level of detail relevant to the study was collected).

*4) Agree how to address data collection in areas of limited exposure:* We recommend that operators take steps to ensure that the fracture network they collect is complete (i.e. all node types have the correct number of branches and the counts of parameters are checked) and consistent with the network observed in areas of full exposure. This could be achieved though the extrapolation of trends from outside the sample area, or through ensuring the consistency of the network within the sample area (e.g. are EW trending joints consistently connected to NS joints by y-nodes?). It is important that areas of no exposure (see Fig. 7d) are interpreted as best possible, otherwise estimates of trace length and connectivity will be unrepresentative of the network. This approach is also important as it enables the operator to gain further insight into the development of the fracture network: for example, a better understanding of the age relationships between fracture traces (Procter and Sanderson, 2017). If this is completed as the first step of fieldwork, sources of counting errors can be identified and minimised. Regardless, the sampling or counting error identified should be communicated as part of the data reporting.

*5) Where possible, collecting fracture data from field exposures is preferable to interpreting field photographs:* We find that there is less variability in fracture data collected by different participants when data is collected in the field, rather than collected from field photographs. Field-based observations have a number of advantages over photo-based approaches: the operator can change position and distance for more complex fractures, remove obstructing material, adjust so that something isn't in shadow, physically feel for the fracture, check if a feature rubs off, or if it is continuous into another plane of the outcrop. A further advantage of collecting data in the field is the ability to look outside the sample area, to ensure that the fracture network within the sample area is consistent with the wider network, and to enable kinematic data to be collected. A caveat to this recommendation is that in the field the quality of observations can be negatively affected by environmental factors (e.g. rain, cold, heat etc.) which are not encountered during analysis undertaken in the office. Recording such factors and the likely effect on one's field approach is good practice.

*6) Working as a group:* Working as a group is preferable to working individually to collect fracture data, since we find less variability and fewer inconsistent nodes in data collected as groups. However, group work should be considered a collaborative and dialogic process, where participants discuss their rationale or reasoning before, during and after data gathering, as opposed to divvying up tasks to be completed individually in a team. In the former, working together allows for the identification and reconciliation of differences in interpretational approach, while improving the mechanics of the data gathering, thereby reducing the potential for subjective bias by increasing the detail of observations. The quality of the data collected will be more consistent as a result. In line with this, a group comprised of different cognitive types is preferable. In particular, sensory-type operators should be paired with intuitive-type operators, and encouraged to work collaboratively to tease out whether and how the detail observed by sensory participant is identified and interpreted. The level of geological experience is not relevant to consider when selecting groups, but the relationship dynamics within the group should be managed such that the less experienced individuals feel comfortable to actively discuss with those more experienced than them, rather than simply consent to their views or defer to their judgement.

If data are to be collected separately and then combined, then the sampling behaviour of members of the team should be assessed prior to data collection to establish if data from the individuals can indeed be meaningfully combined. The sampling strategy should be conceived such that the minimum number of moderate-scale, obvious, fractures should be captured (i.e. when using a circular approach, the radius should capture 20-30 terminations of the major fracture sets), with the small fractures still recorded. If conducting collaborative fieldwork, where operators are working individually to collect data from different sampling sites, the team must first characterise their own biases, then agree on a unified approach and classification system, the process of determining sample location and dimensions, and what to do when, e.g. a particular fracture intersection is obscured. It is important to characterise the way participants differentiate fracture terminations and distribution of reported trace lengths.

*7) **Define a data cut-off**:* Because all fractures larger than 10 to 15% of the circle diameter are typically well defined by all data gatherers, all data above this size can be confidently compared between operators with different fracture judgements. The circle radius should be set and reported on prior to the start of the collection of field data. It is vital that the scanline is large enough to cover enough fractures for the least detailed-orientated member of a group to still collect sufficient number of fractures. We recommend that the scale of observation is kept consistent throughout the survey and if a minimum fracture trace length cut-off is chosen that it is clearly reported in field reports and publications.

The procedure could be further improved, and tested through either (a) using a set of calibration scanlines prior to data collection to test personal biases and familiarise the operator with the technique or (b) have a scanline, or sample area which is used as a marker and completed regularly throughout data collection procedure to test replicability, as also advised by Peacock et al., (*in press*). While the above procedure outlined above is undoubtedly helpful and goes someway to providing consistency in fracture data collection, it also does not take into account that behaviours may change through time (e.g. Scheiber et al., (2015)). Such changes may be due to such things as experience of the data gathering procedure, experience of trends in the fracture network being classified, subsequent training (e.g. the introduction of minimum trace length cut-offs) or when undertaking fracture data collection with differing survey goals (e.g. paleo-stress analysis vs fluid flow studied). Due to this the procedure should be repeated regularly and assigned to 'single events' such as a day in the field or a single data collection session.

*8) **Communicate the steps taken to manage bias in data collection:*** Steps one to seven should be communicated as part of data reporting and publication.

## 6.4 Wider geoscientific implications

While this work concentrates on a 'field-based approach', which uses several 'data points' (sampling areas) to collect data from outcrop, many of our findings are also relevant to the collection of data from broad scale approaches such as UAV or remote sensing derived maps. With the advent of digital-image analysis techniques and UAV technology, it can seem preferable to perform digital fracture mapping, however, uncertainties regarding, say, hairline fractures, potential weathering features, or vegetation obscuring the fracture network can be more easily explored by direct field observations. While one may expect marginal error, which is a function of the sample size, to be reduced by digital fracture mapping, since digital mapping allows for a much larger number of (and area of) fractures to be sampled in a given time. We instead suggest this to not be the casse because each participant is in effect using their own method to identify and classify features on the digital image being studied, many of the subjective biases that we observed in our work will be applicable to remote mapping methods. This corroborates work by Scheiber et al. (2015), who investigated the number of lineaments identified by six participants interpreting the same LiDAR dataset (at the same resolution). Extreme variability was observed between participants, who counted between 74 and 607 lineament traces (COV = 1.61). Indeed, concern about consistencies in image interpretation was raised in early work on remote imagery; Huntington and Raiche (1978) suggested that inter-operator variability in the interpretation of lineaments from LANDSAT imagery could be so significant that it may seem as if different scenes with different geologies had been interpreted.

In this work, we have demonstrated, for the first time, the clear need for geoscientists to develop consistent and transparent protocols for collecting field data that is scientifically rigorous. We find that the type and scale of subjective biases that affect how we identify, classify, and report on fracture characteristics are independent of experience, and appear to be related to personal character traits. It is vital that the geoscientific community become more aware of the potential for subjective bias, the subsequent effect on scientific uncertainty, and options to manage biases. Indeed, we feel that these issues should be discussed openly from the very first time that students collect field data. Training schemes and procedures should be developed that not only consider the relative differences between methods (as in Watkins et al., 2015) but also the inherent human factors which effect data collection. These schemes will differ based on the specific aims of the study, however, approaches to manage

subjective uncertainty in data must be communicated openly so as to enable the study's findings to be replicable, and to facilitate comparison with other field data.

In fact, we propose that a series of reasoned recommendations or protocols derived from and adopted by the scientific community could prove valuable to streamline the data collection process and reduce the uncertainty in observation-based sciences. The recommendations for field-based fracture data collection may be different to those for remote sensing images. Any such workflow should not be so prescriptive as to be inhibitive, or to limit the scope of study, however, should be supportive enough such that the results obtained by the adopted method are replicable. Since the type and scale of subjective bias is independent of the level of experience or expertise, a suitable workflow should enable crowd sourcing or citizen science to be a useful medium for fracture data collection and analysis in such a way that is commonplace in ecological studies (Dickinson et al., 2010). Indeed, our work has implications beyond the geoscience discipline; for example, to garner maximum potential from Big Data, these subjective uncertainties and any protocols to manage them must be reported. However, our work also demonstrates the clear need for further work in this field, to test the effects of subjective or operator bias on the collection of fracture data, both in the field and using maps generated from remote sensing, in addition to investigating the role of subjective bias in other forms of geological data and beyond.

## 7. Conclusions

In Arthur Conan Doyle's *Silver Blaze* (1892), Sherlock Homes states "I only saw it because I was looking for it". We observe that this behaviour may be common in geoscience data collection and has the potential to impart subjective biases in the data collected, introducing uncertainty in the geological information derived from these data and potentially affecting the ability to replicate studies. We demonstrate that geologists' own subjective biases influences the data they collect, and, as a result, different participants collect different fracture data from the same scanline or sample area. This has consequent effect on the fracture statistics that are derived from these data and that are used to inform geological models. Although we find that participants can collect a range of data, we observe internal consistency in the classification of and number of fractures gathered by each participant. This consistency is not related to geological expertise or experience, nor the time taken to complete the scanline, so we propose that the underlying control on the subjective bias relates to the individual's personal characteristics (detailed vs pragmatic) and also the process that the data will inform (bulk fluid flow? Scale of relevant observation?). Major fracture sets tend to be captured by all participants, and so the subjective bias mostly affects the smaller-scale fracture features. We find that the effect of subjective bias on the fracture statistics derived from the observed fracture attributes can be large, and that trace length and fracture density are the parameters that are most susceptible to subjective bias.

The subjective biases in how features are identified, classified, and reported have implications for how data should be collected and collated. Firstly, for the characteristics of a fracture network to be statistically valid, a circular scanline should aim to capture a minimum number of fractures in its area, and the radius adjusted to ensure that these conditions are met. However, to meet the necessary validity criteria, individuals who pay particular attention to small features could potentially use a circular scanline with much smaller radius (and consequently, can collect data from smaller outcrops) than individuals who tend to dismiss small fractures. Secondly, by comparing fracture data collected in the field and from field photographs, we find that if possible fracture data should be collected in the field, where the type of connections present can be examined in more detail.

Drawing on the quantitative and qualitative data in this study, we propose a series of methods for managing subjective bias. As well as supporting individuals to understand – and so mitigate - their own biases, there are other practical steps that can be taken. For example, we suggest that the perceived fracture network should be drawn out, either onto printed field photos or using a tablet computer, to minimise bias by prompting the operator to consider and report the trace length distribution and network topology. Doing so also records not just the number of terminations and individual trace lengths, but also where in

the scanline/are the values recorded, and also makes clearer the rationale behind the interpreted fractures. For similar reasons, we also propose that people should work collaboratively in (small) groups when gathering fracture data, and preferably with people who have different personal characteristics to them. A series of protocols could be developed to streamline fracture data collection and reduce uncertainties introduced by subjective biases, but, ultimately, the steps taken to manage bias in data collection should be communicated as standard during data reporting and publication.

This study is the first to quantitatively illuminate and discuss the scale of and potential causes of subjective bias in the collection of geological field data. As the implications of our findings has relevance for a range of observation-based sciences beyond geoscience, from digital mapping to Big Data, our study is, ultimately, a call for further work in this area.

**Data availability**

The workshop documents and data collected as part of this study is available in the supplementary information.

**Supplement link (Will be included by copernicus)**

**Team List**

**Author contributions**

Initial discussions and planning of the paper was undertaken by all authors, with BA and JRR designing the workshops. The paper was prepared by BJA and JRR, with contributions from all authors.

**Competing interests**

The authors declare that they have no conflict of interest.

**Disclaimer**

**Special issue Statement (Will be included by Copernicus)**

**Acknowledgements**

This work was funded through Billy J. Andrews' PhD studentship, supported by the Environmental and Physical Sciences research council (EPSRC, award number EP/L016680/1); the ENOS project: H2020-EU.3.3.2.3. Develop competitive and environmentally safe technologies for CO2 capture, transport, storage and re-use, Record Number: 664337; UKCCSRC, funded by the EPSRC (EP/K000446/1, EP/P026214/1) as part of the RCUK Energy Program; and the University of Strathclyde Global Engagement Fund. The authors also wish to thank all the participants who attended the workshops and took part in the fieldwork as part of this study.

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

**Tables**

| Fracture statistic | Notation | Definition (unit) | Input parameters and calculation | | |
|---|---|---|---|---|---|
| | | | Linear | Circular scanline | Window sampling |
| Density (D) | Areal (P20) | Number of fractures per unit area (m⁻²) | - | $D = \frac{(Ni+Ny)}{2\pi r^2}$ | $D = \frac{N}{A}$ |
| Intensity (I) | Linear (P10) | Number of fractures per unit length (m⁻¹) | $I = \frac{n}{L} = \frac{1}{S}$ | $I = \frac{n}{4r}$ | - |
| | Areal (P21) | Fracture length per unit area (m x m⁻²) | - | - | $I = \frac{\sum tl}{A}$ |
| Spacing (S) | Linear | Spacing between fractures (m) | $S = \frac{\sum s}{(N-1)} = \frac{1}{I}$ | - | - |
| Mean trace length (TI) | TI | Mean fracture length (m) | $Tl = \frac{\sum l}{N}$ | $Tl = \frac{n}{(Ni+Ny)} \times \frac{\pi r}{2}$ | $Tl = \frac{\sum l}{N}$ |
| Network topology | Topological sampling | Defining fracture nodes as I, y and x. | - | Yes | Yes |
| Connectivity | Using node topology (Pc) | Percentage of connected branches | - | $Pc = \frac{3Ny + 4Nx}{Ni + Ny + Nx}$ | $Pc = \frac{3Ny + 4Nx}{Ni + Ny + Nx}$ |
| | Using trace end classification (Pf) | Percentage of connected fractures | $Pf = \frac{F}{R + F} \times 100$ | - | - |
| Trace length distribution | TI distribution (tl) | Distribution of individual fracture trace lengths | Yes | - | Yes |

**Table 1: Summary and definition of fracture statistics that can be derived from methods used in this work. Table adapted from Zeeb et al. (2013). Ni = number of i-nodes, Ny = number of y-nodes, Nx = number of x-nodes, r = radius of circular scanline, N = number of fractures, A = Area, n = number of fracture intersections with the scanline (either linear or circular), L = length of scanline, s = spacing between adjacent fracture traces on the scanline, tl = individual fracture trace length, F = fracture abuts against another fracture, R = fracture terminates into rock (n.b. some authors also distinguish stratabound fracture terminations), 'Yes' for trace length distribution & network topology indicates you can use that method to carry out the technique.**

| Method | | Field | | | Workshop | | | | Length or radius/m |
|---|---|---|---|---|---|---|---|---|---|
| | | Completed? | i | g | Completed? | i | g | Order | |
| Circular | C1 | ✓ | ✓ | | ✓ (WS1&2) | ✓ | | 3 | 1.0 |
| | C2 | ✓ | ✓ | | ✗ | | | | 1.0 |
| | C3 | ✓ | ✓ | | ✓ (WS1&2) | | ✓ | 5 | 1.0 |
| | C4 | ✓ | ✓ | | ✓ (WS1&2) | | ✓ | 4 | 1.0 |
| | C5 | ✓ | ✓ | | ✓ (WS1&2) | ✓ | | 2 | 1.0 |
| | C6 | ✓ | ✓ | | ✗ | | | | 0.73 |
| | C7 | ✓ | ✓ | | ✗ | | | | 1.21 |
| | C8 | ✗ | | | ✓ (WS1&2) | ✓ | | 1 | 0.5 |
| Linear | L1 | ✓ | ✓ | | ✗ | | | | 1.0 |
| | L2 | ✓ | | ✓ | ✗ | | | | 1.0 |
| | L3 | ✓ | | ✓ | ✗ | | | | 15.0 |
| | L4 | ✓ | | ✓ | ✗ | | | | 7.5 |
| | L5 | ✗ | | | ✓ (WS1&2) | | ✓ | | 6.55 |
| | L6 | ✗ | | | ✓ (WS1&2) | ✓ | | | 1.45 |
| Window sampling | C1 | | | | ✓ P1,3,11 & WS2 | ✓ | | 3 | 0.5 |
| | C3 | | | | WS2 | | ✓ | 5 | 1.0 |
| | C4 | | | | WS2 | | ✓ | 4 | 1.0 |
| | C5 | | | | ✓ P1,3,11 & WS2 | ✓ | | 2 | 0.5 |
| | C8 | | | | ✓ P1,3,11 & WS2 | ✓ | | 1 | 0.5 |

**Table 2: Summary of circular (C) and linear (L) scanlines completed in the field and workshops (WS1 & WS2). Whether these were completed individually (i) or in groups (g) is noted. 'Order' refers to the order the scanlines were completed in the workshops. Four of the circular scanlines (C2,3,4,5) were completed both in the field and in the workshop, but none of the linear scanlines were completed in both, due to workshop time constraints. Window sampling, whereby participants drew out the interpreted fractures as well as completing topological sampling, was only completed by Participants 1, 3, 11 and all of Workshop 2 (WS2). The workbooks used in this study are supplied in the supplementary information (S3 & S4).**

| Group | # participants | Geological training | | | | | Familiarity with geological fieldwork | | | | | Familiarity with collecting fracture data | | | | |
|---|---|---|---|---|---|---|---|---|---|---|---|---|---|---|---|---|
| | | None | Low | Medium | High | (Other) | None | Low | Medium | High | (Other) | None | Low | Medium | High | (Other) |
| Field | 7 | 1 | 0 | 3 | 3 | 0 | 1 | 0 | 3 | 3 | 0 | 1 | 0 | 3 | 3 | 0 |
| WS1 | 11 | 2 | 2 | 3 | 2 | 2 | 2 | 1 | 5 | 1 | 2 | 3 | 2 | 5 | 1 | 0 |
| WS2 | 18 | 3 | 0 | 6 | 9 | 0 | 3 | 6 | 3 | 6 | 0 | 6 | 5 | 5 | 2 | 0 |

**Table 3. Summary of the level of geological training, and experience in geological fieldwork and fracture data collection, reported by field and workshop (WS) participants. Individual participant responses are provided in the Supplementary Information (S2).**

| Scanline | | Individual/ | # | Fracture count | | | | Trace length (m) | | | | | | Time (minutes) | | | |
|---|---|---|---|---|---|---|---|---|---|---|---|---|---|---|---|---|---|
| | | | | Min | Max | Median | QCV | Min | Max | Mean | QCV | Median | QCV | Min | Max | Median | QCV |
| L1 | Field | i | 6 | 3 | 10 | 7.0 | 0.71 | 0.03 | 2.22 | 0.58 | 0.36 | 0.40 | 0.15 | 5:32* | 9:00* | 7:16* | 0.24 |
| L2 | Field | G | 3 | 7 | 14 | 12.0 | 0.29 | 0.01 | 1.78 | 0.43 | 0.17 | 0.26 | 0.21 | - | - | - | - |
| L3 | Field | G | 3 | 21 | 38 | 26.0 | 0.33 | 0.04 | 23.08 | 1.21 | 0.69 | 0.54 | 0.18 | 10:00 | 13:00 | 10:00 | 0.15 |
| L4 | Field | G | 2 | 18 | 19 | 18.5 | 0.03 | 0.05 | 14.4 | 2.29 | 0.61 | 1.17 | 0.69 | - | - | - | - |
| L6 | WS1 | i | 11 | 10 | 23 | 14 | 0.39 | 0.02 | 0.61 | 0.21 | 0.39 | 0.19 | 0.43 | 2:17 | 8:40 | 4:58*** | 0.33 |
| | WS2 | i | 18 | 9 | 25 | 21 | 0.38 | 0.03 | 0.72 | 0.24 | 0.28 | 0.23 | 0.27 | 1:51 | 24:00 | 6:12** | 0.66 |
| L5 | WS1 | G | 5 | 22 | 31 | 22 | 0.23 | 0.12 | 2.72 | 0.86 | 0.73 | 0.70 | 0.82 | 5:57 | 9:35 | 7:33 | 0.24 |
| | WS2 | G | 7 | 15 | 28 | 20 | 0.40 | 0.14 | 2.43 | 0.96 | 0.21 | 0.86 | 0.47 | 5:00 | 13:00 | 8:17 | 0.57 |

**Table 4: Summary table of raw linear scanline results where i = individual, G = groups, # = number of participants/groups. *only two participants recorded time for this scanline **P10 did not record time taken to count nodes ***P23 did not trace fractures so only have spacing and time information.**

| | | i/g | n-point | | | | | | Node Count | | | | | | | | | | | |
|---|---|---|---|---|---|---|---|---|---|---|---|---|---|---|---|---|---|---|---|---|
| | | | n | | | t (sec) | | | i-node | | | y-node | | | x-node | | | t (sec) | | |
| | | | Range | median | QCV | Range | Median | QCV | Range | Median | QCV | Range | Median | QCV | Range | Median | QCV | Range | Median | QCV |
| C1 | Field | i | 15-21 | 17 | 0.16 | 19-42 | 29.5 | 0.39 | 0-3 | 0.5 | 3.5 | 12-21 | 19 | 0.22 | 6-14 | 7 | 0.32 | 137-230 | 172 | 0.33 |
| | WS1 | i | 14-23 | 18 | 0.22 | 36-99 | 68 | 0.87 | 0-12 | 1 | 2.0 | 1-38 | 19 | 0.74 | 4-11 | 7 | 0.43 | 119-447 | 240 | 0.77 |
| | WS2 | i | 11-25 | 18.5 | 0.30 | 15-295 | 82 | 1.41 | 0-6 | 1 | 2 | 4-34 | 18 | 0.81 | 4-14 | 7.5 | 0.5 | 82-1140 | 289.5 | 1.11 |
| C5 | Field | i | 14-19 | 15.5 | 0.16 | 14-43 | 21 | 0.64 | 4-8 | 5 | 0.45 | 28-47 | 34.5 | 0.38 | 2-8 | 3.5 | 1.07 | 127-245 | 165.5 | 0.32 |
| | WS1 | i | 7-18 | 12 | 0.25 | 20-120 | 47 | 0.78 | 3-14 | 5 | 0.9 | 4-34 | 20 | 0.53 | 1-6 | 2 | 0.75 | 150-1177 | 317 | 0.46 |
| | WS2 | i | 9-18 | 12 | 0.08 | 20-298 | 67.5 | 0.99 | 0-32 | 4.5 | 1.39 | 7-41 | 14 | 0.5 | 0-11 | 1.5 | 1.17 | 60-1050 | 281 | 1.09 |
| C8 | WS1 | i | 10-25 | 23 | 0.17 | 29-180 | 78 | 0.77 | 2-11 | 5 | 0.5 | 1-60 | 26 | 0.5 | 2-22 | 10 | 0.3 | 150-780 | 378 | 0.8 |
| | WS2 | i | 16-32 | 24 | 0.22 | 45-240 | 107 | 0.48 | 1-16 | 4 | 0.5 | 5-45 | 19.5 | 0.92 | 5-18 | 10.5 | 0.48 | 30-1440 | 599 | 0.64 |
| C4 | Field (i) | i | 12-20 | 15 | 0.13 | 24-50 | 41 | 0.41 | 5-19 | 13 | 0.31 | 20-34 | 29 | 0.17 | 0-4 | 0 | - | 147-215 | 167 | 0.35 |
| | WS1 | g | 11-18 | 14 | 0.36 | 60-330 | 97 | 0.49 | 7-19 | 9 | 0.22 | 6-27 | 11 | 0.73 | 1-4 | 3 | 0.67 | 324-521 | 405 | 0.16 |
| | WS2 | g | 10-18 | 14 | 0.39 | 64-323 | 129 | 0.96 | 5-23 | 5 | 0.9 | 5-27 | 11 | 0.50 | 0-3 | 1 | 1.5 | 115-720 | 290 | 1.35 |
| C3 | Field | i | 19-30 | 22 | 0.05 | 24-58 | 38.5 | 0.68 | 3-15 | 7 | 1.0 | 21-33 | 29 | 0.24 | 6-16 | 8 | 0.5 | 162-282 | 261 | 0.15 |
| | WS1 | g | 18-22 | 19.5 | 0.17 | 55-90 | 77.5 | 0.2 | 4-20 | 5.5 | 0.86 | 19-24 | 23 | 0.05 | 5-11 | 5.5 | 0.41 | 208-521 | 322 | 0.29 |
| | WS2 | g | 14-23 | 16 | 0.13 | 52-713 | 129 | 1.77 | 2-54 | 7 | 1.43 | 11-22 | 18 | 0.39 | 3-10 | 4 | 0.38 | 143-600 | 360 | 0.63 |

**Table 5: Summary of fracture data and time taken for circular scanlines 1, 5 and 8, in the field and workshop, either working individually (i) or in groups (g). The data are presented in the order scanlines were completed in the workshops.**

| Circle | Number of participants | Number of fractures | | | Trace length (m) | | | | | |
|---|---|---|---|---|---|---|---|---|---|---|
| | | Range | Median | QCV | Min | Max | Mean | QCV | Median | QCV |
| 8 (i) | 20 | 18-54 | 30.5 | 0.49 | 0.01-0.10 | 0.70-0.98 | 0.27 | 0.31 | 0.17 | 0.43 |
| 5 (i) | 20 | 13-56 | 22.5 | 0.48 | 0.02-0.12 | 0.68-1.05 | 0.33 | 0.40 | 0.24 | 0.39 |
| 1 (i) | 20 | 9-40 | 23.5 | 0.44 | 0.01-0.40 | 0.67-1.03 | 0.37 | 0.37 | 0.30 | 0.95 |
| 4 (g) | 7 | 11-29 | 17 | 0.76 | 0.02-0.11 | 1.89-1.95 | 0.69 | 0.47 | 0.52 | 0.60 |
| 3 (g) | 7 | 18-50 | 25 | 0.46 | 0.04-0.22 | 1.82-2.01 | 0.61 | 0.26 | 0.38 | 0.26 |

**Table 6: Summary of fracture parameters reported for window sampling. Data is presented in the order the scanlines were undertaken within the workshops. (i) and (g) denote whether the scanline was undertaken individually or as a group.**

| Statistic | Circular Scanline – topology | Circular Scanline - Window | Linear Scanline |
|---|---|---|---|
| Intensity | Very low to low variability when derived from field data and low to moderate when workshop data is used. For Circle 1, 4 and 5 the calculated intensity from workshop and field data were very similar, however, the calculated intensity for Circle 3 was much lower in the workshop. In all cases ranges are greater when workshop data is used, particularly for Circles 1 and 5. | Low spread between participants within circles. In all cases, apart from Circle 4, intensity calculated using window sampling is lower than that derived for node counting for a given circle. | Variability, which ranged from very low to high, depends on the scanline being sampled. For example, Lines (Line 1, Line 6) than others (Lines 3 - 5, all low intensity, have small range). |
| Density / Spacing | Low to high spread when derived from field data and moderate to very high when workshop data is used. Density calculated from workshop in all cases apart from Circle 1 is lower than when calculated from field data. | Moderate to high spread. Values consistently higher in workshop data when window sampling data is used compared to node counting, particularly Circle 8. Can be both comparable to field density (Circle 4) or considerably higher (Circle 1). | Variability in mean spacing values depends on the scanline being sampled, ranging from very low to very high. Maximum reported spacing had low spread, whereas, minimum spacing ranged from low to extreme variability depending on the scanline being sampled. Equally large range in workshops and field. |
| Mean trace length | Low to moderate spread when derived from field data and moderate to high when workshop data is used. How similar the range in reported values are between workshop and field data varies for different circles. | Moderate spread across all circles. The extremes in the ranges observed in mean trace length estimates are considerable lower than for node counting. Of all methods window sampling provides the smallest estimate for mean trace length. | Moderate to Highly variable for most scanlines. Equally large range in workshops and field. Maximum reported trace lengths generally much larger than for other methods, due to the different scale of observation. |
| Connectivity | Very low spread, both between circles, between methods, and settings (field vs workshop). | Not assessed separately from node classifications. | Spread depends on the scanline being sampled and ranges from very low to extremely variable. Equally large range in workshops and field. |

**Table 7: Summary of the broad trends in fracture statistics derived from the three methods we explored, presented in Fig. 9.**

**Figures**

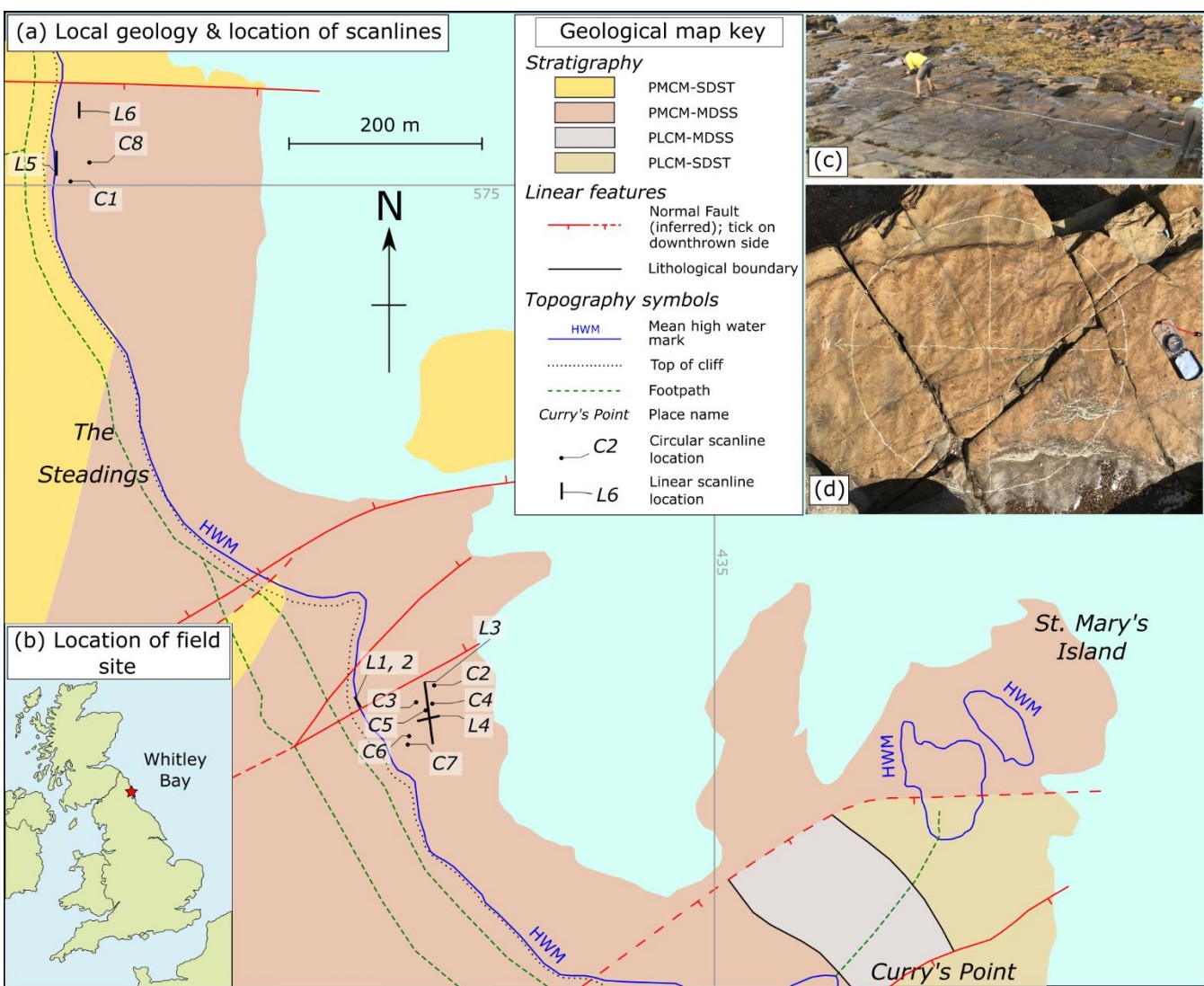

Figure 1: Location map highlighting (a) the local geology and (b) the location of the study area, located near Whitley Bay, Northumberland (UK). Grid lines are annotated with UK national grid numbers. Field photographs of both linear (c) and circular (d) scanline methods are also shown (L3 [NZ34717545] and C8 [NZ34377609] respectively). The geological map is modified from Geological Map Data BGS © UKRI (2018), where stratigraphy is as follows: PLCM-SDST = Pennine Lower Coal Measures – Sandstone; PLCM-MDSS = Pennine Lower Coal Measures – Mudstone, siltstone and Sandstone; Pennine Middle Coal Measures – Sandstone; PLCM-MDSS = Pennine Middle Coal Measures – Mudstone, siltstone and Sandstone.

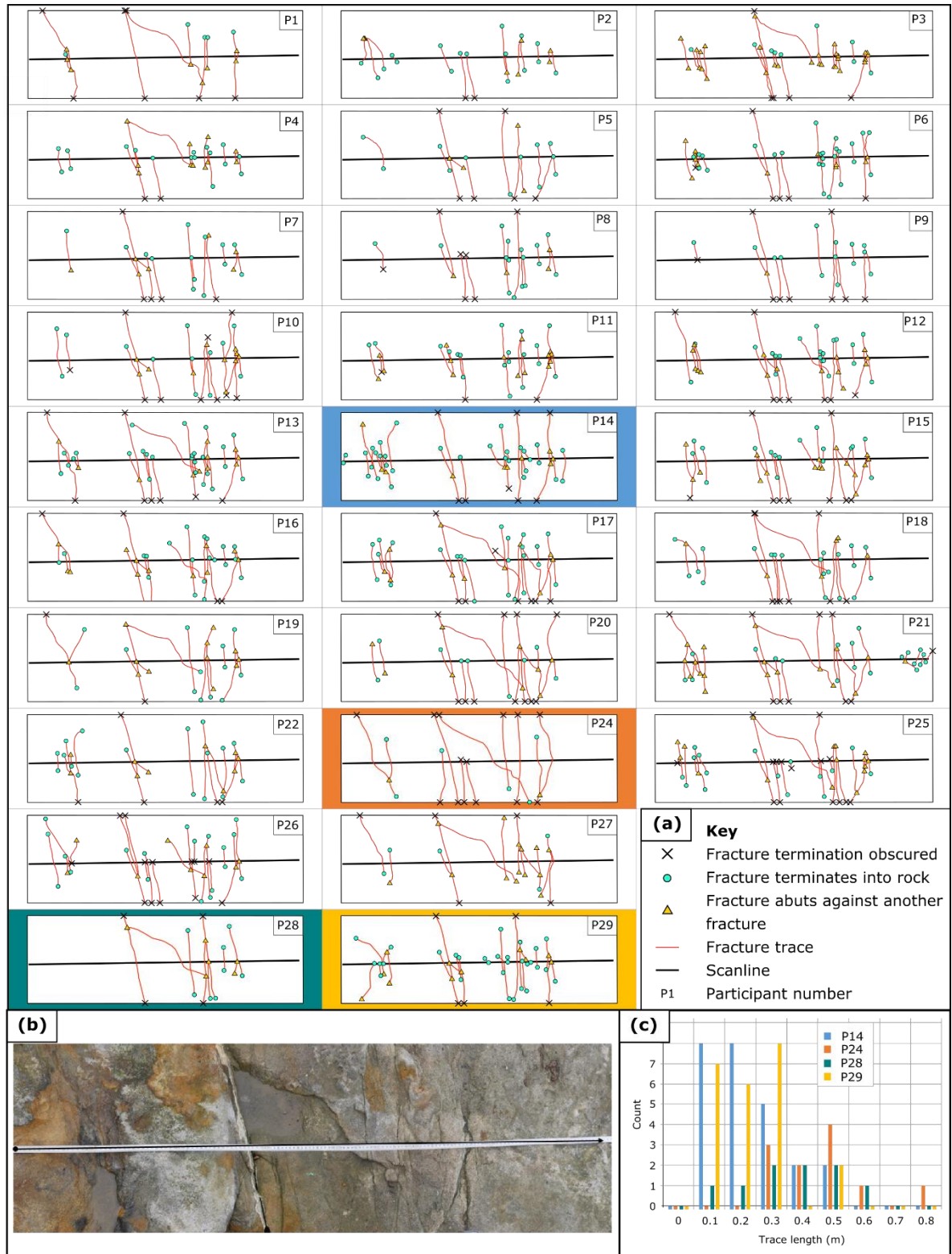

**Figure 2: The interpreted fracture traces for Line 6 (length 1.45 m). (a) The digitised fracture networks for all workshop participants. (b) Field photograph of Line 6. (c) Fracture trace length histograms (bin = 0.1 m) for participants who recorded a low to high number of fractures. The corresponding digitised fracture trace is also highlighted in the appropriate colour. Key differences in the interpreted fracture networks are highlighted using participants who selected a low (Participant 28, 9 fractures), medium (Participant 10, 17 fractures) and high (Participant 14, 25 fractures) number of fractures.**

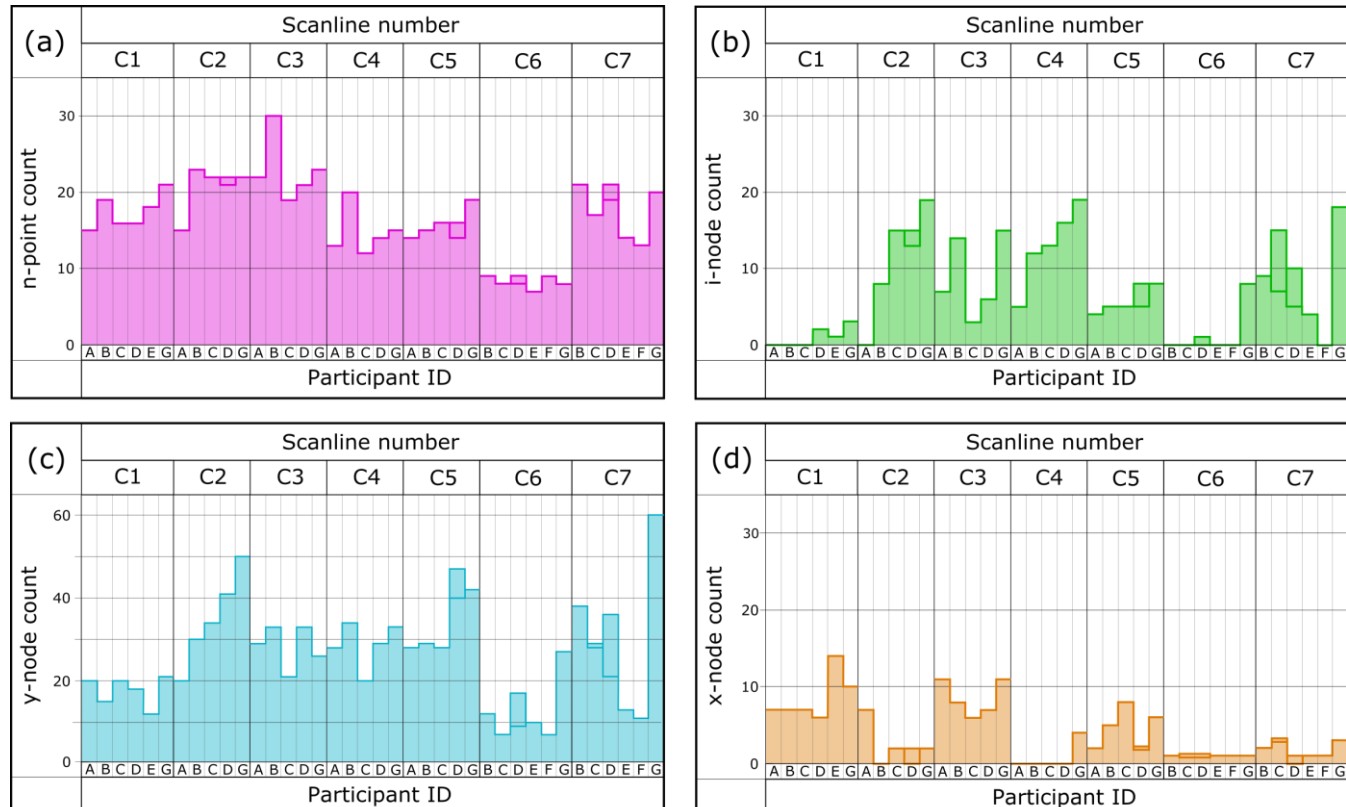

**Figure 3: Results of the fracture data from circular scanlines (C1-7) collected in the field by 7 participants (labelled A-G, though A, E and F did not complete all of the scanlines). (a) the number of fractures that intersected the circular scanlines (n). (b) fractures that terminated in rock (i-nodes). (c) fractures that terminated against another fracture (y-nodes). (d) fractures that intersect another fracture (x-nodes). Participants C and D repeated some of their measurements for selected circles and this is indicated by two bars in their column for that circle.**

**(a)**

| | P | n-point C8 | n-point C5 | n-point C1 | i-node C8 | i-node C5 | i-node C1 | y-node C8 | y-node C5 | y-node C1 | x-node C8 | x-node C5 | x-node C1 | Time n-point C8 | Time n-point C5 | Time n-point C1 | Time Node C8 | Time Node C5 | Time Node C1 |
|---|---|---|---|---|---|---|---|---|---|---|---|---|---|---|---|---|---|---|---|
| Workshop 1 | 1 | 23 | 11 | 21 | 4 | 9 | 2 | 22 | 24 | 32 | 12 | 2 | 7 | 78 | 70 | 68 | 540 | 324 | 337 |
| | 2 | 11 | 8 | 16 | 2 | 4 | 2 | 1 | 4 | 1 | 3 | 3 | 8 | 107 | 59 | 99 | 378 | 317 | 259 |
| | 3 | 24 | 14 | 20 | 5 | 12 | 1 | 60 | 34 | 38 | 10 | 2 | 7 | 46 | - | 72 | 460 | 1177 | 447 |
| | 4 | 22 | 12 | 16 | 6 | 3 | 0 | 28 | 17 | 18 | 9 | 1 | 7 | 106 | 53 | 50 | 602 | 333 | 119 |
| | 5 | 10 | 7 | 14 | 5 | 3 | 1 | 1 | 7 | 5 | 2 | 1 | 5 | 83 | 70 | 70 | 172 | 150 | 120 |
| | 6 | 25 | 14 | 23 | 4 | 5 | 12 | 27 | 26 | 29 | 12 | 1 | 11 | 52 | 32 | 51 | 312 | 330 | 416 |
| | 7 | 20 | 11 | 17 | 2 | 3 | 0 | 13 | 20 | 16 | 9 | 2 | 10 | 120 | 30 | 60 | 480 | 300 | 300 |
| | 8 | 25 | 16 | 19 | 6 | 7 | 3 | 26 | 12 | 20 | 15 | 6 | 8 | 36 | 28 | 38 | 150 | 150 | 211 |
| | 9 | 25 | 14 | 16 | 5 | 5 | 0 | 33 | 24 | 12 | 12 | 4 | 5 | 180 | 120 | 60 | 780 | 480 | 240 |
| | 10 | 21 | 12 | 18 | 2 | 5 | 0 | 23 | 19 | 19 | 22 | 1 | 4 | 29 | 20 | 49 | 171 | 186 | 141 |
| | 11 | 24 | 18 | 19 | 11 | 14 | 2 | 47 | 31 | 27 | 10 | 2 | 5 | 47 | 41 | 36 | 242 | 184 | 125 |
| Workshop 2 | 12 | 24 | 13 | 18 | 4 | 4 | 2 | 42 | 26 | 25 | 8 | 1 | 6 | 102 | 298 | 295 | 1200 | 235 | 290 |
| | 13 | 26 | 18 | 25 | 15 | 32 | 6 | 45 | 41 | 34 | 18 | 6 | 10 | 180 | 60 | 60 | 1380 | 900 | 540 |
| | 14 | 28 | 12 | 21 | 4 | 7 | 1 | 23 | 16 | 18 | 9 | 1 | 9 | 109 | 80 | 107 | 705 | 451 | 538 |
| | 15 | 25 | 16 | 22 | 2 | 5 | 1 | 31 | 32 | 34 | 16 | 5 | 8 | 129 | 64 | 80 | 864 | 737 | 528 |
| | 16 | 24 | 13 | 19 | 5 | 13 | 1 | 14 | 14 | 20 | 12 | 0 | 4 | 105 | 89 | 230 | 660 | 600 | 259 |
| | 17 | 19 | 11 | 20 | 3 | 6 | 2 | 20 | 15 | 13 | 13 | 7 | 14 | 94 | 58 | 48 | 622 | 310 | 509 |
| | 18 | 26 | 12 | 19 | 3 | 2 | 0 | 19 | 13 | 10 | 15 | 2 | 7 | 134 | 71 | 84 | 504 | 235 | 186 |
| | 19 | 22 | 9 | 20 | 4 | 4 | 3 | 26 | 14 | 32 | 8 | 1 | 4 | 210 | 112 | 176 | 598 | 350 | 430 |
| | 20 | 16 | 12 | 18 | 1 | 2 | 1 | 5 | 23 | 18 | 5 | 2 | 6 | 45 | 240 | 254 | 125 | 325 | 217 |
| | 21 | 25 | 14 | 22 | 4 | 7 | 4 | 6 | 10 | 12 | 18 | 11 | 8 | 55 | 33 | 45 | 295 | 237 | 289 |
| | 22 | 18 | 11 | 13 | 5 | 3 | 0 | 7 | 11 | 4 | 7 | 1 | 10 | 98 | 131 | 74 | 730 | 517 | 550 |
| | 23 | 25 | 11 | 15 | 16 | 11 | 6 | 8 | 9 | 10 | 6 | 2 | 7 | 120 | 60 | 120 | 300 | 120 | 540 |
| | 24 | 22 | 12 | 11 | 4 | 3 | 0 | 7 | 7 | 9 | 12 | 2 | 6 | 120 | 120 | 120 | 600 | 180 | 240 |
| | 25 | 23 | 12 | 16 | 2 | 2 | 0 | 18 | 11 | 21 | 6 | 0 | 6 | 70 | 20 | 40 | 240 | 60 | 180 |
| | 26 | 32 | 12 | 17 | 8 | 11 | 2 | 29 | 14 | 25 | 12 | 3 | 8 | 121 | 34 | 46 | 458 | 165 | 138 |
| | 27 | 20 | 12 | 15 | 4 | 2 | 0 | 21 | 18 | 12 | 9 | 0 | 7 | 52 | 25 | 32 | 527 | 252 | 213 |
| | 28 | 16 | 12 | 13 | 1 | 0 | 0 | 7 | 9 | 5 | 9 | 1 | 10 | 46 | 21 | 15 | 30 | 60 | 82 |
| | 29 | 27 | 14 | 21 | 8 | 9 | 1 | 22 | 18 | 25 | 13 | 1 | 10 | 240 | 90 | 180 | 1440 | 1050 | 1140 |

**(b)**

| | G | n-point C4 | n-point C3 | i-node C4 | i-node C3 | y-node C4 | y-node C3 | x-node C4 | x-node C3 | Time n-point C4 | Time n-point C3 | Time Node C4 | Time Node C3 |
|---|---|---|---|---|---|---|---|---|---|---|---|---|---|
| Workshop 1 | 1 | 14 | 22 | 7 | 20 | 17 | 24 | 3 | 11 | 330 | 90 | 521 | 521 |
| | 2 | 13 | 18 | 11 | 4 | 11 | 19 | 1 | 6 | 62 | 82 | 324 | 208 |
| | 3 | 18 | - | 19 | - | 27 | - | 4 | - | 97 | - | 405 | - |
| | 4 | 11 | 18 | 9 | 5 | 9 | 23 | 2 | 5 | 60 | 73 | 357 | 332 |
| | 5 | 18 | 21 | 9 | 6 | 6 | 23 | 4 | 5 | 110 | 55 | 420 | 312 |
| Workshop 2 | 6 | 18 | 23 | 14 | 11 | 13 | 17 | 1 | 6 | 120 | 60 | 600 | 360 |
| | 7 | 18 | 18 | 5 | 3 | 27 | 22 | 3 | 3 | 129 | 129 | 720 | 600 |
| | 8 | 14 | 14 | 5 | 7 | 14 | 18 | 1 | 4 | 323 | 713 | 115 | 143 |
| | 9 | 12 | 16 | 5 | 16 | 11 | 22 | 1 | 3 | 184 | 389 | 445 | 168 |
| | 10 | 10 | 16 | 5 | 4 | 5 | 13 | 2 | 4 | 116 | 113 | 290 | 465 |
| | 11 | 12 | 18 | 5 | 2 | 8 | 11 | 0 | 10 | 300 | 240 | 120 | 360 |
| | 12 | 17 | 16 | 23 | 54 | 8 | 22 | 3 | 4 | 64 | 52 | 140 | 205 |

**Key**

| Rank for n-point and node counts | Rank for n-point and node time |
|---|---|
| Lowest | Fastest |
| Medium | Medium |
| Highest | Slowest |

**Figure 4: Recorded fracture data (n, and node counts) and the time taken to undertake n and node counts for workshop (WS) participants (P) and groups (G). The data for each attribute has been colour-coded according to where the reported value for the parameter ranked for that circle. Data are presented in the order that they were completed in the workshop.**

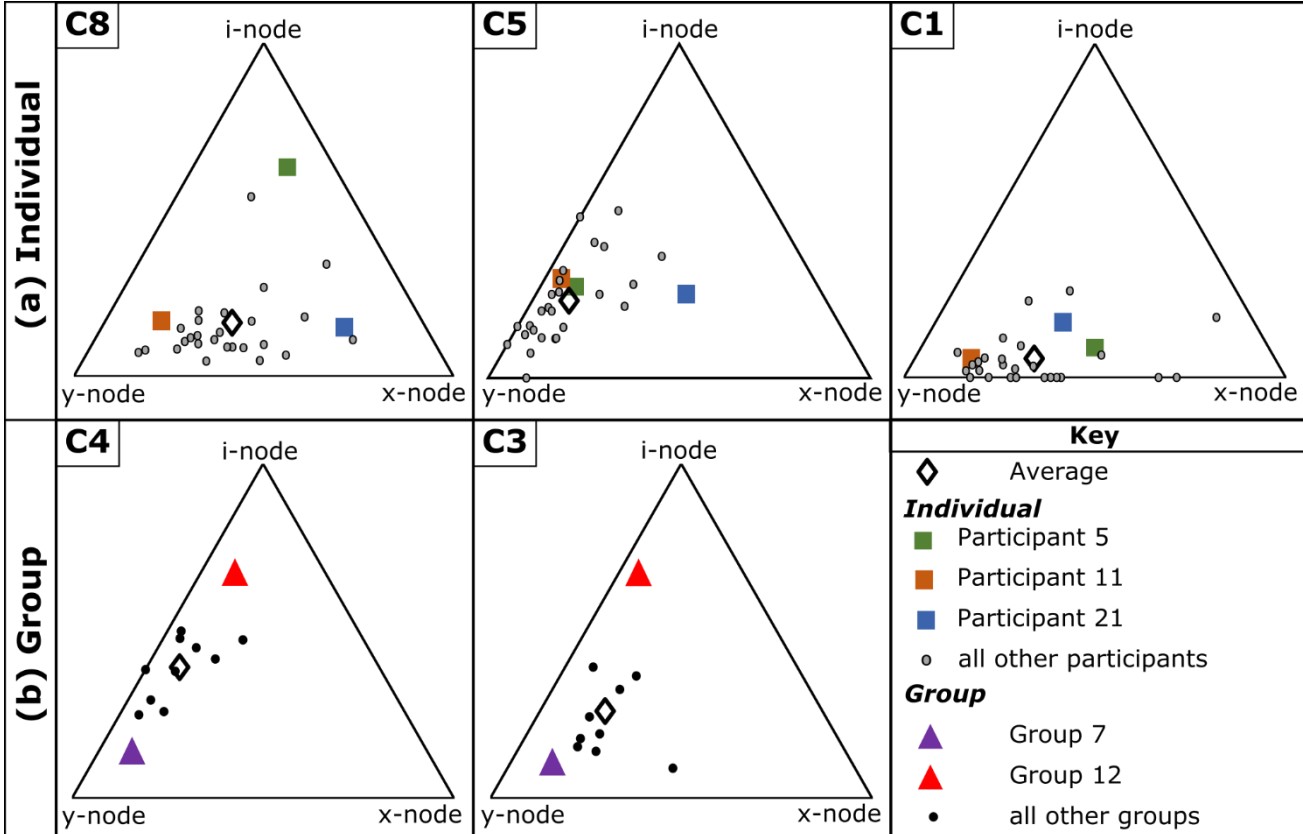

**Figure 5: Node triangles for workshop participants and groups.** For individual circles (a), Participants 5, 21, and 11 were highlighted to show the consistency the way participants classified nodes. Participants were selected according the whether they reported a low (P5), medium (P21) or high (P11) node count. Similarly, for group circles (b) Groups 7 and 12 were highlighted as groups who recorded a high and low node count.

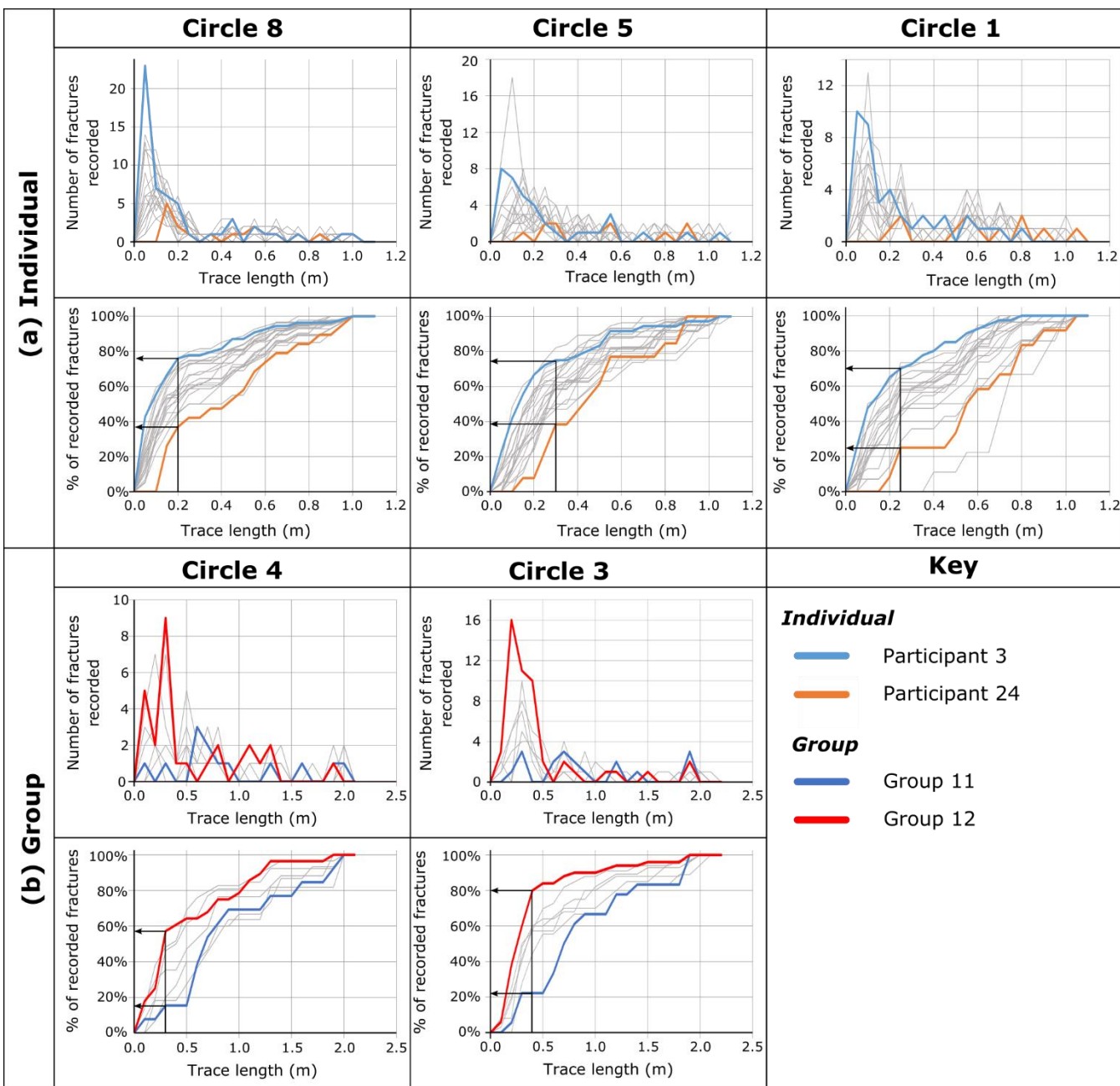

**Figure 6: Fracture trace length distributions for (a) individual and (b) group window sampling data. The results are presented as both histograms and normalised cumulative frequency curves of fracture trace length with bin widths of 0.05 m for individual and 0.1 m for group window sampling data. The range in the relative percentage of small fractures observed in the data is highlighted using Participants and groups who consistently observed a high and low percentage of small fractures (Participant's 3 and 24 and Groups 12 and 11 respectively).**

d

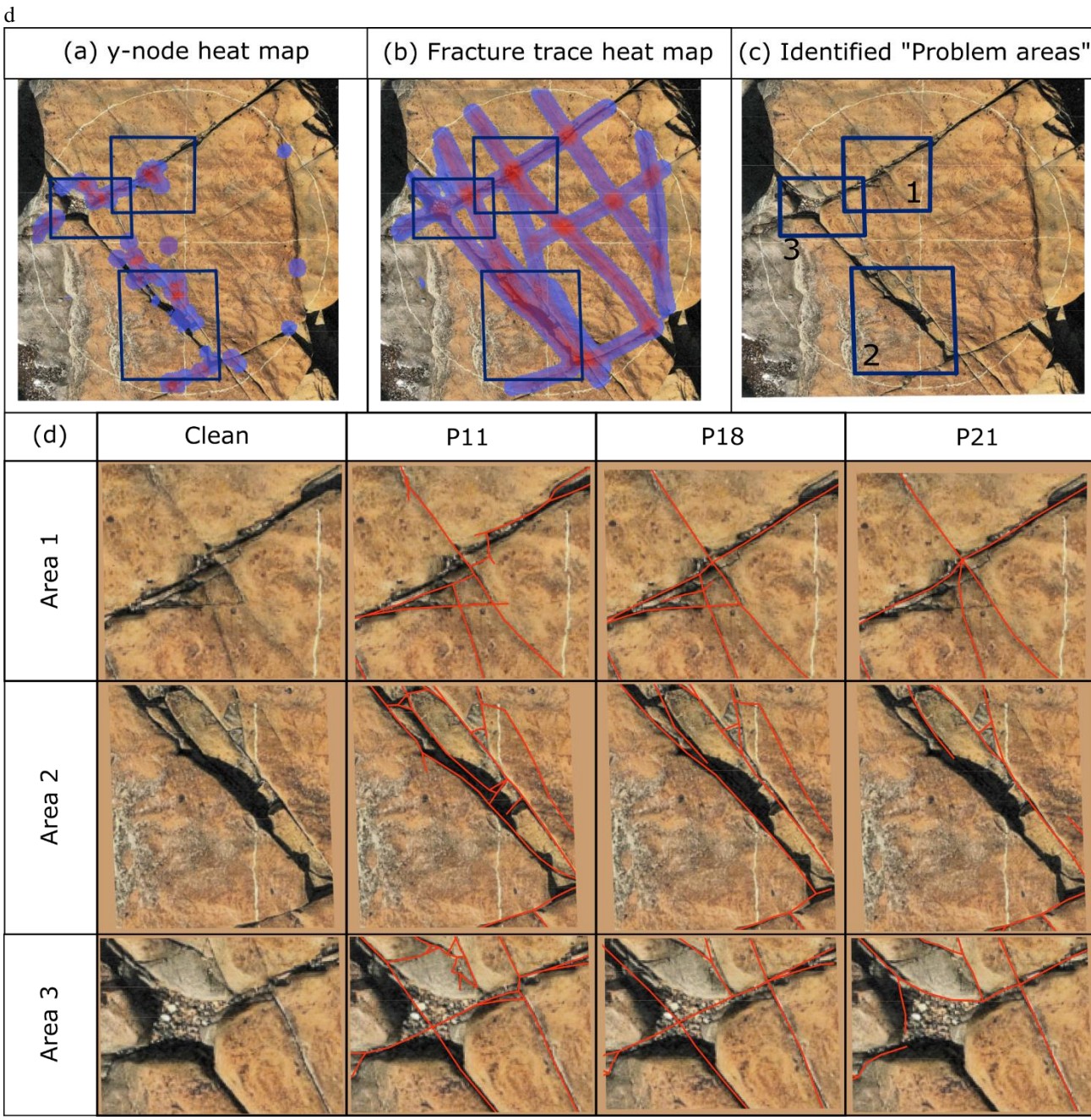

**Figure 7: A detailed study of the areas which cause increased uncertainty in Circle 8. The figure comprises of clean field photographs of Circle 8 with the (a) heat map of y-node point density, (b) heat map of fracture trace density and (c) areas identified as problem areas. In panel (d) the close up of areas 1, 2 and 3 along with the features recorded by Participants 11, 18 and 21 are shown. See text for full description.**

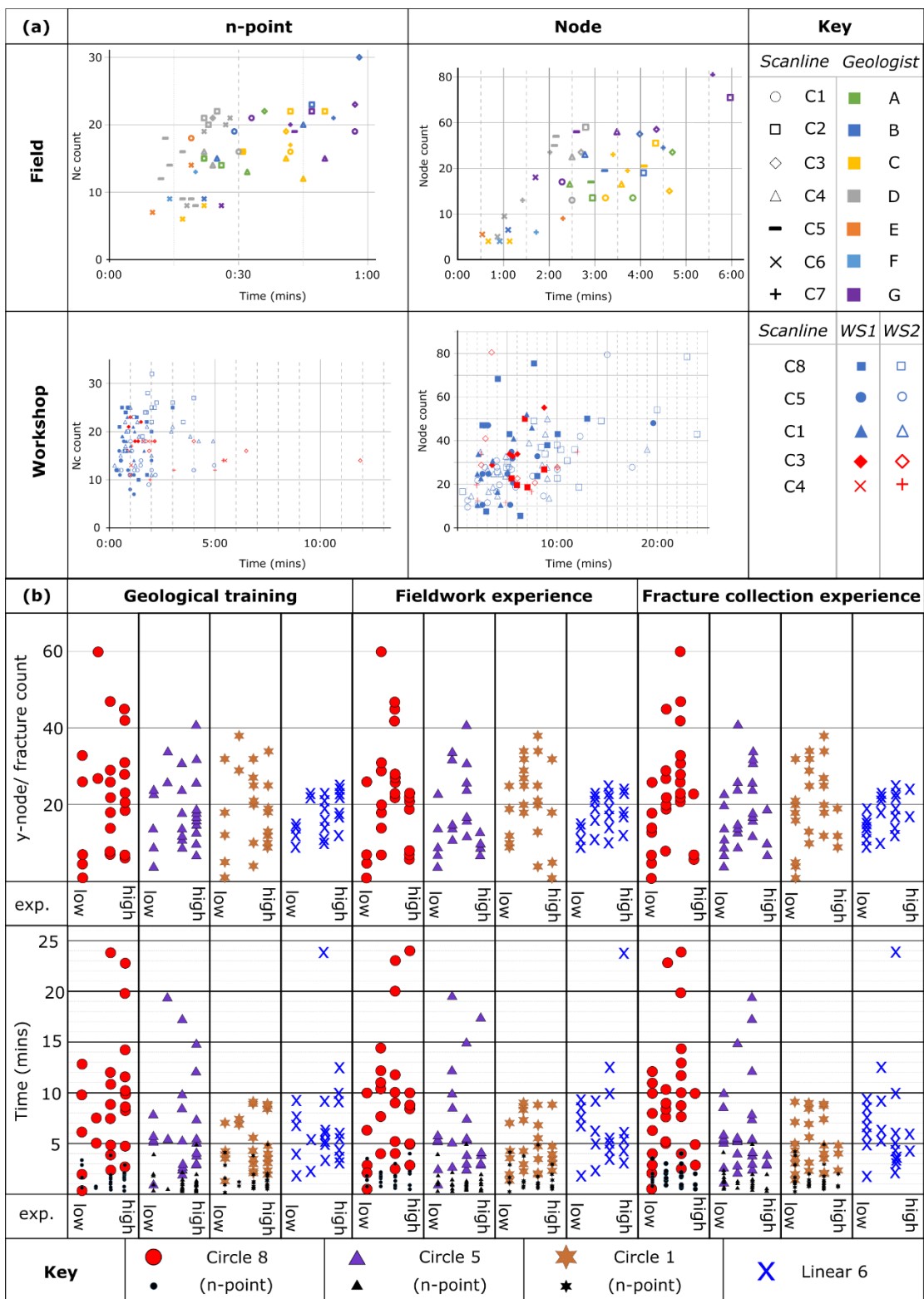

**Figure 8: The impact of participant experience on the collection of fracture data. (a) The time taken in seconds to record fracture data (n and node counts) from circular scanlines both in the field and workshops. (b) The impact of experience on the recorded y-count and number of fractures in individual scanlines and the time taken to complete the workshop tasks.**

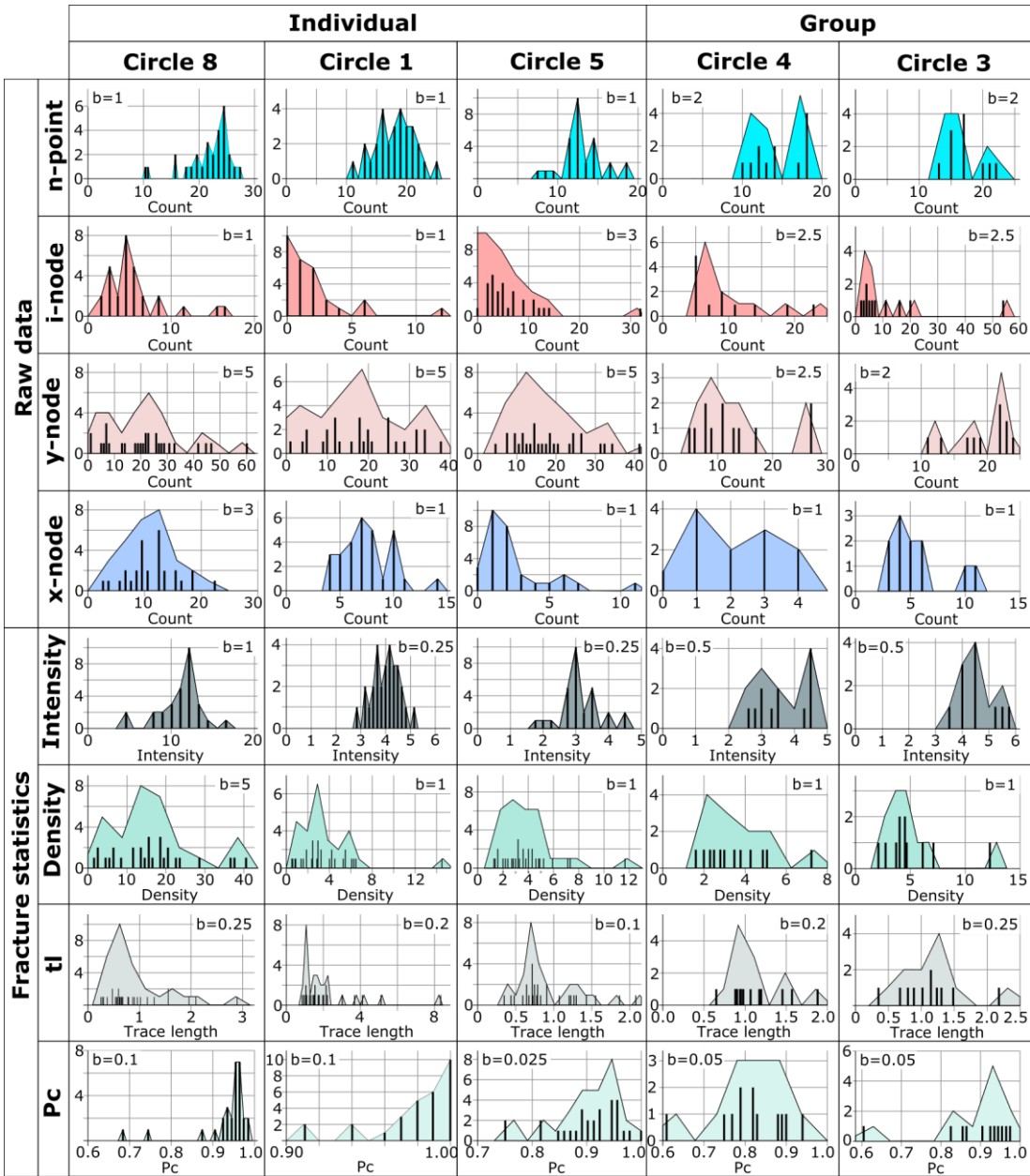

**Figure 9: Topological sampling results for individuals and groups for circular scanlines 1, 3, 4, 5 and 8. Each histogram reports the results for all workshop participants. The statistics have been derived from the data for each participant. Data is presented as both bar charts and shaded histograms with the bin width, b, indicated on the chart (please note the bin width varies between circles as a function of the range in reported or calculated values). In all cases the y-axis represents frequency and is scaled so the shape of the distributions can be assessed.**

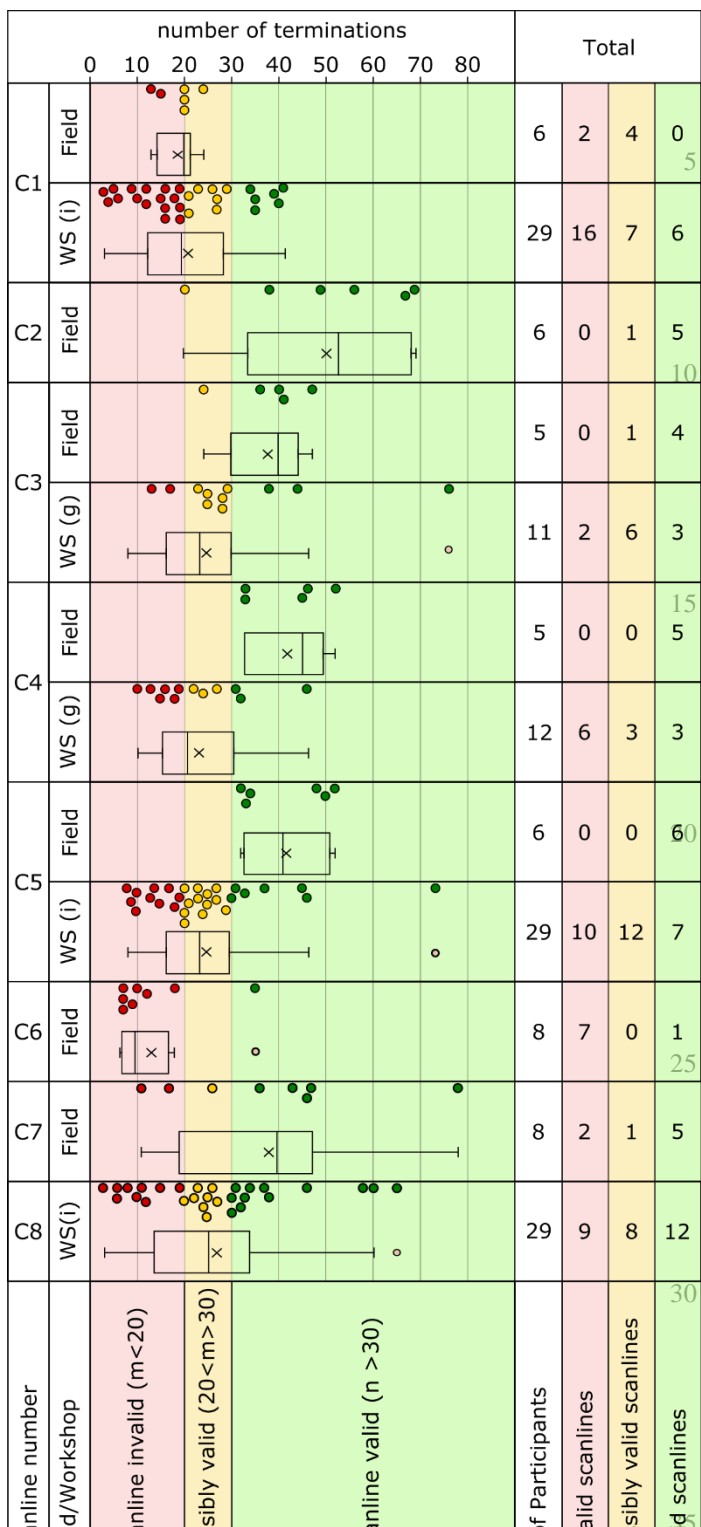

| Scanline number | Field/Workshop | number of terminations | # of Participants | invalid scanlines | possibly valid scanlines | valid scanlines |
|---|---|---|---|---|---|---|
| C1 | Field | | 6 | 2 | 4 | 0 / 5 |
| C1 | WS (i) | | 29 | 16 | 7 | 6 |
| C2 | Field | | 6 | 0 | 1 | 5 / 10 |
| C3 | Field | | 5 | 0 | 1 | 4 |
| C3 | WS (g) | | 11 | 2 | 6 | 3 / 15 |
| C4 | Field | | 5 | 0 | 0 | 5 |
| C4 | WS (g) | | 12 | 6 | 3 | 3 |
| C5 | Field | | 6 | 0 | 0 | 6 / 60 |
| C5 | WS (i) | | 29 | 10 | 12 | 7 |
| C6 | Field | | 8 | 7 | 0 | 1 / 25 |
| C7 | Field | | 8 | 2 | 1 | 5 |
| C8 | WS(i) | | 29 | 9 | 8 | 12 / 30 |

**Figure 10: The effect of subject bias on the validity of circular scanlines.** The number of terminations recorded by individuals or groups is displayed for each circle and colour coded depending on where a valid (>30, green), possibly valid (20-30, yellow) or invalid (<20, red) number of terminations were recorded.

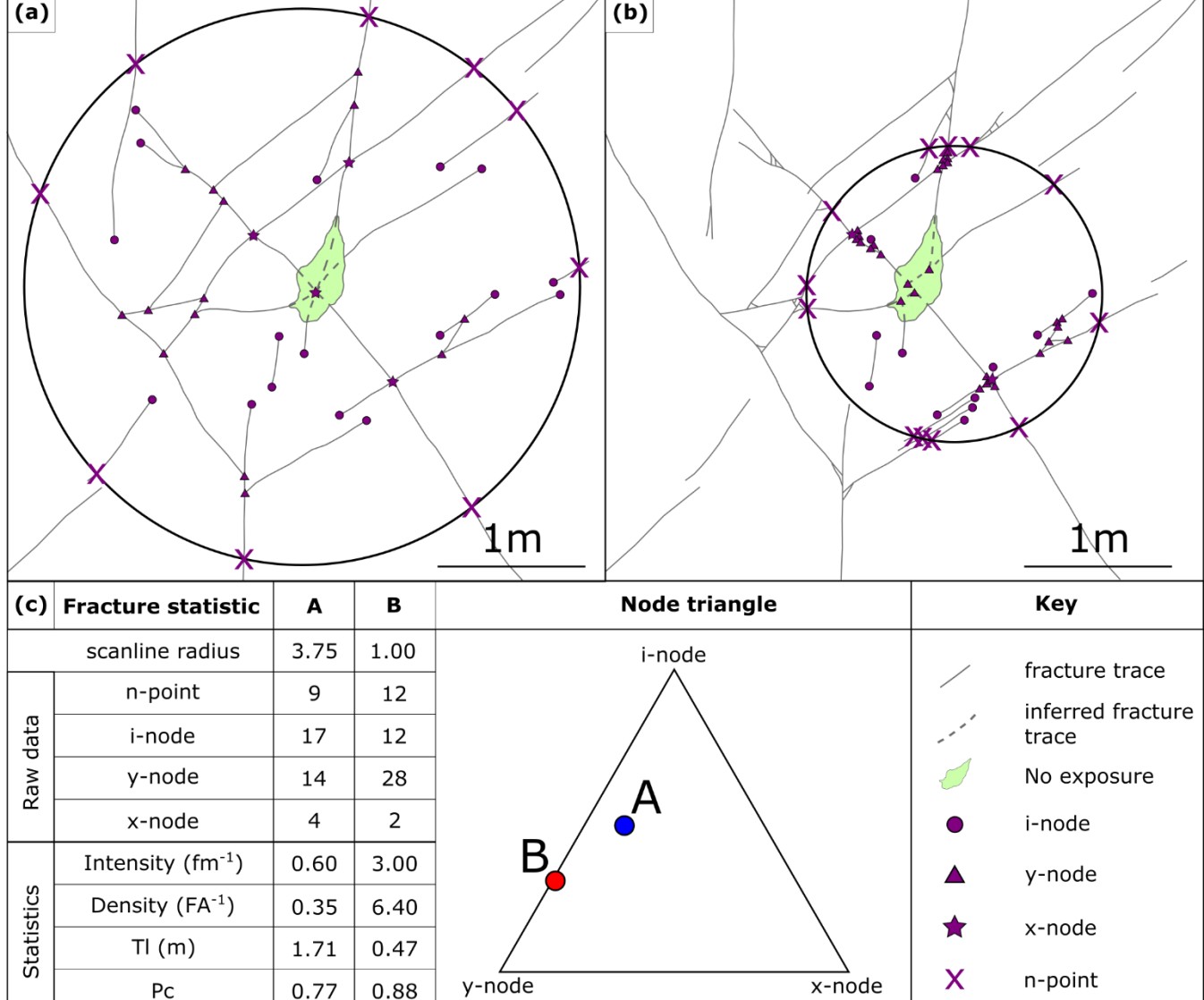

**Figure 11: The impact of interpreter style on fracture statistics of a synthetic fracture network. (a) statistically valid topological sampling within a circular scanline for a fracture network which only considers the large scale fracture network. (b) statistically valid topological sampling within a circular scanline for the same large scale fracture network as (a), however, also capturing small scale fractures at fracture intersections. (c) The topology attributes (n, i-, y- and x-nodes), derived fracture statistics and node triangle of the different interpretations of the fracture network.**