# Peer review of "How do we see fractures? Quantifying subjective bias in fracture data collection."

_Solid Earth, 2018_

## Referee Comment (RC1) · Roberto Emanuele Rizzo (Referee) · 5 Feb 2019

The manuscript presents a thorough and detailed work illustrating the issues related to subjective biases during fracture collection processes. A variety of acquisition methods for fracture attributes (i.e. scan-line, window sampling, circular scan-line and topological sampling) are introduced and compared to assess their response to subjective biases during collection. The results obtained in three different 'interpretation sessions' (two workshops using images and one in the field) are then used by the authors to draw conclusions on the impact that subjective biases induce on the parameter estimation acquisition methods to and to build a protocol to reduce the effect of subjective biases during fracture data collection. The concepts will be of great interest to the Solid Earth readership since fractures play fundamental roles in many applied settings and I

suggest minor corrections for the manuscript.

Major comments: In my opinion, a fundamental underlying issue that has not been addressed by the authors directly relates to use of their data to draw conclusions on the accuracy in the parameter estimations of acquisition methods. In particular, do the authors have taken into consideration the possibility that the errors and uncertainties related with subjective biases can scale with the number of fractures measured in the network? Letting interpret a larger fracture dataset to participant would have reduced the uncertainties in the estimation of the fracture attributes, independently of the acquisition method used?

Although the authors clearly state that it is not in their aim to "collect sufficient fractures to represent the fracture network" and that "the tested scanlines were not designed to be statistically representative" (page 5 lines 28 – 30), at the same time they do "consider the effect of the variation on fracture statistics derived from data collected..." (page 3 lines 3 and 4) and they dedicate a Section on the "Effect of subjective bias if the derived fracture statistics" (Section 4). I fear that their conclusions on this specific matter lack of statistical robustness, because of small number of fractures in the samples.

A well-known behaviour in statistics is what is called the 'marginal error': the size (N) of a statistical sample affects the standard deviation (i.e. variability) of the same sample (Moor, D.S., McCabe G.D. "Introduction to the practise of statistics", 1999; pages 294-295; 391-392). The variability shown in this work suffers the relatively small sample size in the number of fractures interpreted by the participants. The variability in a sample (the spread of the sample distribution) matters as much as bias when building a robust and significant dataset. Because N (i.e. the fracture sample) is the denominator of the sample st.dev. formula (s= $\sqrt{((\sum(x\_1 - x)^2)/(N-1))})$ the st.dev. decreases as N increases. It follows then that having less data in your sample gives more variation (and

I understand that reviewing the manuscript in the light this comment may take con-

siderable time (due to re-running tests), therefore I suggested minor revisions for this manuscript; however, I advise the authors to account for these possible biases in their interpretation on fracture statistics throughout the manuscript.

Minor comments: Page 2, Lines 13-14: Can you please review this sentence? As written it is not very clear.

Page 3, Line 20: Please check the use of 'Nc': should not it be written as 'n-points' in accordance with nomenclature in the following sentence?

Page 4, Line 4: Please check the sentence: is there a 'where' missing between '. . . a technique' and 'all fractures. . .'?

Page 4, Line 14: I would suggest to add '. . . and window sampling' to the listed methods: "Trace lengths may be measured directly with linear scanlines and window sampling, or estimated. . ."

Page 5, Lines 10-11: Can you please fully explain how you measure connectivity in linear scanlines? Are you using only x- and y-connections?

Page 5, Lines 18-19: Can you please add the trending attitude for the sub-vertical joint set? Is this a third set?

Page 5; Lines 24-25: Can you please write the size of the used circles? In this context, looking at Table 2: why did the size of circle change between different localities? How did you choose the size of the circle?

Page 5, Line 26: Missed a capital letter 'P' in Participants. For consistency, please consider changing the 'Nc' (throughout the manuscript) to 'n' or vice-versa.

Page 7, Line 30: Consider changing 'Does not' instead of 'doesn't'.

Page 8, Lines 10-14: How do the authors assess variability? Does 'variability' refer to the statistical variance within a single scan-circle? To show the variability in your sample, I would suggest to accompany the mean values shown in the tables with variance

or standard deviation. The word 'variability' has been used by the authors throughout their text, however any 'variance/ standard deviation' is never statistically evaluated.

Page 10, Lines 12-13: 'Participant 11 depicted' instead of 'depicts' and 'Participant 18 and 20' did not. . .' instead of 'do not', consistently with previous sentence.

Page 11, Line 29: How do the authors assess the trend? Only visually?

Page 11, Lines 31 – 32: How do you evaluate indicators for trends?

Page 12, Lines 9-10: The authors refer to two mean trace length values derived from two participants measurements, however I could not find these numbers. Can you please indicate to which table are you referring to?

Page 12, Line 11: Can the authors, please, mention to what the 'Rˆ2' values stands for? Is it a coefficient of correlation?

Page 12, Line 22: Please check 'al', should it not be 'all'?

Page 12, Lines 25-26: Can you please further explain why window sampling is less subjected to biases?

Page 15, Lines 4-5: This sentence raises the question: what can be considered a 'tall geologist'? I would suggest to avoid this kind of assertion if not fully accompanied with demonstrations and scientific data.

Page 16, Line 21: Please consider changing 'won't' with 'will not'

Page 17, Lines 28-30: Please consider re-phrasing, as written the sentence is a bit obscure

Page 17, Lines 30-32: Would not this be known only after having analysed the whole fracture networks?

Page 27, Table 4: For the fracture count, it would be interesting to see the spreading of the data: i.e., the DeltaN (difference between Min and Max). Instead, for the trace

length data do you have taken into consideration min and max within each individual/group observation? Is the Min and Max reported into the table a mean of the Min values? Similarly for the Max?

Page, 42 Figure 9: Why does the bin size vary within the same attribute?

Please also note the supplement to this comment:
https://www.solid-earth-discuss.net/se-2018-135/se-2018-135-RC1-supplement.pdf

---

## Short Comment (SC1) · 5 Feb 2019

Stephen Laubach

steve.laubach@beg.utexas.edu

This is an interesting article. Using outcrops to understand the attributes of fractures at depth is a very important challenge right now for structural geology. With the advent of drone-based outcrop imaging, many fracture trace pattern data sets are being collected, interpreted, and used to build DFN models. So it is critical to understand biases in the data collection and interpretation. This is a useful contribution to understanding those observations. Some of the problems documented in this paper, like how to objectively document length, are ones that need further thought. Marrett (probably in Marrett and Ortega) concluded that length and connectivity were too subjective to measure meaningfully, which is part of the reason he advocated linear scanlines and careful aperture size measurement (as in Ortega et al. 2006). It should be standard

practice to specify aperture size cut offs in linear scanline data, and using cutoffs and other rules may be useful for acquiring reproducible length data sets.

In a report on subjectivity in fracture data collection, there are some other important problems that should at least be mentioned. As S. Holmes said in Silver Blaze, 'I saw it because I was looking for it' and the fracture community seems to have some highly obscuring blinders on when it comes to some aspects of fractures. In my opinion because we're used to looking at fracture patterns in a certain limited way (Laubach et al. 2010, J. Struct. Geol.) For example, if we're interested in constructing DFN models for fractures at depth, are barren joints in outcrop a useful structure to measure in the first place? Fractures in core commonly have some amount of mineral lining; they've been subject to hot fluids for sometimes millions of years (references in Lander & Laubach 2015, GSA Bulletin). The problems with the specific methods we use may not be as important as the unexamined subjectivity of the choices we make about which outcrop to study.

In the interest of supporting this open comment format, here are a few additional remarks keyed to page and line number (why not use continuous numbering?).

2/9 In general I think the advantages of circular scanlines tend to get over sold, at least as applied to sedimentary rocks outside of intensely deformed fold and fault zones. For regional fractures, which may have a few or only one simple fracture set with sparse, widely spaced fractures, linear scanlines may be the only way to get meaningful data on fracture occurrence; they are not subject to the interpretation problem of picking fracture 'ends' that affect 2D approaches, and they are directly comparable to the 1D data sets available from wellbores. And there are methods available for looking at fracture spatial arrangement in a rigorous way (i.e., Marrett et al. 2018, J. Struct. Geol.)

2/25 And the diagenesis of the fracture network. In many sedimentary rock fracture systems, diagenesis is the principle control on fracture network connectivity (i.e., Olson

et al. 2009, AAPG Bulletin, as you cite later) and this is often overlooked by structural geologists with their mechanics and geometry disciplinary blinders on. This is a great example of 'subjective uncertainty'.

3/30 The actual overview article should be cited (Laubach et al. 2018) rather than the very short editorial.

Laubach, S.E., Lamarche, J., Gauthier, B.D.M., Dunne, W.M., and Sanderson, D.J., 2018. Spatial arrangement of faults and opening-mode fractures. Journal of Structural Geology 108, 2-15. doi.org.10.1016/j.jsg.2017.08.008

4/10 I agree that trace length is vital to measure, and it's a fracture parameter that can only come from outcrops. But the biggest limitation, and one that seems to be in a blind spot—is the finite size of outcrops. Production and tracer data from the subsurface show that fractures capable of rapidly transmitting fluids can be really long - kilometers long probably in some instances. Outcrops of such size that are also good analogs for subsurface fractures are rare. An example, though, is shown in Li et al. (2018, J. Struct. Geol.) where extremely long fracture trace lengths are visible. The finite size of good outcrop analogs is a big challenge if the aim is guiding DFNs. Is it part of 'subjective uncertainty' what we settle for in terms of outcrop type?

4/20 I think it's worth appreciating that the reason Marrett focused on aperture measurements rather than 'length' was that he appreciated that determining 'length' was (and is) subjective.

4/27 But the concept of 'number of fractures' can also be quite subjective. For example, examined microscopically, most opening-mode fractures show evolution by linkage. Where does one fracture start or end?

And specifying fracture 'size' in connection with defining intensity should be standard practice, and should be noted in contexts like this, following the work for example of Ortega et al. (2006). Not to do so may be another hidden, subjective bias for the fol-

lowing reason. 'Joints' (that is, barren opening-mode fractures) typically have a very narrow aperture size range, so why bother to try to measure aperture sizes? But many fracture populations found in core from sedimentary basins have been known since the late 1980s to typically have wide aperture size ranges. It's possible, therefore, to use the aperture size distribution, which can be measured in outcrop and the subsurface target, to decide how similar outcrop and target likely are. Moreover, if you don't account for size in defining what you measure, you can get wilding different results for intensity. These observations are partly what motivated Ortega et al.'s work. It should be standard practice to specify a size measure when describing 'intensity'. And the potential bias of working on easily visible but possibly misleading joints as guides to the subsurface is a topic that a report on subjectivity in fracture studies ought to at least consider.

5/10 Again, this measure of 'connectivity' ignores cement.

6/5 I wonder how many participants had experience describing fractures in core?

9/4 This seems highly likely. In some of the older fracture 'topology' literature (which seems to have been missed by recent papers) this scale-of-observation effect was explicitly taken into account in connectivity measures. It's another area where it should be a matter of course to take size into account in descriptions.

9/7 Some problems like this can be taken into account by explicitly specifying size cut offs, a procedure that is a regular part of scanline studies focused on aperture size distributions (e.g., Ortega et al. 2006; Hooker et al. 2014).

15/15 These problems can be minimized with linear scanlines and explicit thresholds. Restricts you to measuring aperture sizes, though, so the problems remain for 'defining' length.

15/19 It would really introduce problems to try to determine if fractures in outcrop are fluid conduits or not. Some of the best outcrop analogs for the subsurface may have

completely mineral filled fractures: they are fossilized fracture systems. In outcrop, the open, fluid conductive fractures may preferentially be obscured by vegetation, etc.

17/30 Fractures that are 'not connected' are only unimportant for flow if the host rock is completely tight. The arrangement and length distribution - including that of small fractures - is important if the rock has finite porosity and permeability (the typical situation for even 'tight' sedimentary rocks). See Philip et al. 2005, SPE Reservoir Evaluation & Engineering 8/4, 300-309.

SEL

---

## Referee Comment (RC2) · William Dunne (Referee) · 10 Feb 2019

"How do we see fractures? Quantifying subjective bias in fracture data Collection" by Billy J. Andrews, Jennifer J. Roberts, Zoe K. Shipton, Sabina Bigi, Maria C. Tartarello, and Gareth O. Johnson. https://doi.org/10.5194/se-2018-135

The paper tackles an important general topic in scientific research and choses to use the characterization of natural fracture networks on two-dimensional surfaces as the framework for analysis and discussion. The topic is "the effect of the biases of the observers on the observations that will form the data population for a scientific analysis". The approach is to have a range of participants apply four different established methods for characterizing the networks. The participants and their differences in data

populations that they gathered, are considered in terms of factors such as amount of professional experience, individual or group work, or a participant's preference to gather detailed data carefully or larger data sets quickly. The paper provides extensive and appropriate documentation, and focuses its analysis particularly on the impact to the state of the gathered data as a function of whether observers tend to be "more detailed" or "less detailed" with respect to their data gathering. For the particular case, the authors provide recommendations for which sampling methods best overcome observer bias, sizing the sampling technique, best approaches to data gathering by a group, and integrating project goals into the data-gathering plan to minimize observer bias. \*\*\*\*\*\*\*\*\*\*\*\*\*\*\*\*\*\*\*\*\*\*\*\*\*\*\*\*\*\*\*\*\*\*\*\*\*\*\*\*\*\*\*\*\*\*\*\*\*\*\*\*\*\*\*\*\*\*\*\*\*\*\*\*\*\*\*\*\*\*\*\*\*

Major Comments: Section 3 - A careful set of data are collected about participant performance for factors such as amount of data collection, type of data collection, patterns of data collection, time taken to collect data, and data collection performance as a function of individual or group data collection. These data are well documented. However, the analysis of this data in Section 3 is somewhat vague with statements such as "reasonably consistent", "a suggestion in the data", "differences are not enough to be confident", or "the trend is very weak". No framework for a quantitative and/or qualitative approach is established at the outset of the data presentation and analysis in this section. Presenting this framework and then utilizing it would be a critical for improving the rigor of the present paper. Presenting the framework will likely lead to similar results and will do so in a manner that creates greater confidence in the results presented in this Section.

Page 17, Line 27 to Page 18, Line 5 (End of Discussion) - This text should be replaced by more ambitious text that speaks both more generally than just the mechanics of resolving data gathering differences between observers in the context of "detail" and also connects to real-world situations that apply to the readers beyond just those for the particulars of gathering fracture-related data. So, it is certainly worthwhile constructing experiments that directly test for effects related to subjective bias or operator bias concerning the collection of geological data. Yet, how do experimental results apply to real situations involving data collection? For example, how do the results provide value to an instructor working with a group of students who are performing field data collection for the first time vs. to an individual or team that are applying a rules-based data collection process with specific training prior to the first field deployment to ensure familiarity with the rules-based approach vs. to a computer-based observer utilizing virtual 3-D outcrops from photogrammetric data who has no prior field experience with the data set vs. to building a data set by crowd-sourcing. In this context, the present paper would be a stronger contribution if it explicitly considered the application of its outcomes to real-world circumstances of value and interest to readers. Replacing the existing text at the end of Discussion and embracing this opportunity for expanding the import of the narrative should bring greater recognition to the contribution of the authors and greater interest from the readership. Also, this revised text would address comments made on Page 14 – Line 29, Page 15 – Line 9, and Page 15 – Line 27, where the authors need to extend their work and provide more guidance about the meaning and application of their results.

The Discussion also has a few key locations where the work of others should be included and considered. Please see "Other Comments" for details.
* * *
Other Comments: Page 2, Lines 8-11 – It seems odd to list four methods and only provide citations for one of the four methods. Quality citations exist for all of the methods and the manuscript would be more useful for readers if each method was paired in the text here with at least two appropriate citations.

Page 2, Lines 11-14 – The annotated PDF for this review of the paper provides suggestions for strengthening the statement of the purpose of this contribution.

Page 2, Lines 25-29 - These two sentences consider observational resolution and limitations to the quality of observations that can be made as a function of the exposed rock. These two points would relate to both objective and subjective uncertainty, and as such seem out of place in the narrative flow. Given the text in Lines 22 to 25 that is focusing on subjective uncertainty, any text, if any is needed, after Line 25 in this paragraph should only consider factors the relate to subjective uncertainty. It might be best to eliminate this text and just continue with the text in the new paragraph starting on Line 30 that focuses on the subjective uncertainty and further introduces the paper.

Page 3, Lines 15-18 – Suggested text revisions are offered to completely and correctly state the contribution of Zeeb et al., (2013) to defining the number of measurements needed to provide an estimated value for a characteristic that is statistically significant.

Page 4, Line 3 – It would be useful to explicitly state for the reader why plotting topology data in a triangular diagram is useful.

Page 6 Line 15 - Specify the dimensions of A3 paper as it is not a universally used paper size.

Page 7 Lines 4 to 6 – Suggestions offered in the annotated PDF for this review to improve the clarity and purpose of this text related to methodology and then the approach to statistical characterization.

Page 7, Line 19 - How is "reasonable amount of consistency" quantitatively defined or qualitatively recognized?

Page 7 Line 30 to Page 8 Line 4 - Suggestions are offered to improve the precision and the clarity of the text describing locations and causes of increased observational uncertainty as a function of the participants for Scanline 6.

Page 8, Lines 23-24 - How often were participants "internally consistent"? What is the measure/criterion for "internal consistency"? What is the measure/criterion for defining the occurrence of "often"?

Page 9, Line 6 - What defines or qualifies "varied considerably"? What is the standard or basis for comparison?

Page 9 Line 14 - Words added because an operational preference for a participant to report more smaller fracture traces does not necessitate that the reported small fracture traces are the actual small fractures.

Page 10, Line 30 - The meaning of "the joint highest" is not clear.

Page 10, Line 30 - What defines "a suggestion in the data"? This characteristic or attribute is not defined or explained. More explanation is needed here.

Page 11, Lines 3-4 - While I agree, the paper would be better if the authors explained why they believe "differences are not enough to be confident that this is due to working in groups rather than differences in the fracture network".

Page 11, Line 15 - "correlation is weak" - Is this statement backed by statistical analysis or is that a judgment call by the authors. Additional explanation is needed here.

Page 11, Line 29 - What is the statistical or qualitative basis for stating that "however the trend is very weak". Further explanation is needed.

Page 12, Lines 19-25 - Suggested revisions offered to focus the narratives on the reporting by participants. It is the values as reported by the participants rather than the values themselves that is the focus of this work and the connection of the participants to the values should be explicitly maintained throughout the narrative.

Page 12, Line 25 - The narrative should be more direct and avoid the use of the word "suggest" that is vague and lacking in framework.

Page 12, Line 26 - Revision offered to the latter part of this sentence to clearly and explicitly relate "spreads" in Table 7 to main text, and then clearly state the interpretation that the authors have derived from considering this population of spreads as a function of sampling method and subjective bias.

Page 12, Line 27 - Apologies for my confusion but how does "most robust" relate to "displaying considerable variability". Previously, "robustness" related to similar reported values by participants with limited variability. What am I missing in the text here?

Page 13, Lines 11-12 - Why does the variation in reported values by different participants for the same sampling method directly correlate to the size of the tool for the method, and hence the sample size. Is it not the case that if a statistically valid sample size is going to be collected by each participant that both the sample size and the tool size need to be specified prior to measurement by all participants? Or is there an implication that the dimension of the scanline or window needs to be set on the basis of minimum values to be expected from the range of values due to subjective bias? I suspect that the core problem with this clause of this sentence is that it should come at the end of the paragraph after the key observations are offered, so that a summative comment can then be made and justified. So, this text should be relocated. The comment does also need some improvement in text to provide greater clarity.

Page 14, Line 2 - the participants are not less or more detailed. Their observations are. Suggested text revisions are offered to clarify this point. This approach should be adopted at other appropriate locations in the narrative.

Page 14, Line 5 - A clear conclusion and useful point is reached at the end of this paragraph. What are recommendations for operationalizing the observation with respect to a future data-sampling campaign? How does the needed level of detail for a campaign fit into this operationalization? After all, not every sampling campaign necessarily needs the same level of detail as a function of campaign goals. Or put differently, more or less detail is not always best!

Section 4 - Subsections misnumbered because two Section 4.2's are identified.

Page 14, "First" Section 4.2 - This short section is out of place and effectively encompassed in the later sections including particularly Section 4.4. It should be deleted.

Page 15 Lines 22-23 - Is this suggestion about a preference for using field-based data rather than photo-based data a first occurrence in the literature. If not, prior work should be cited and most likely, briefly discussed.

Page 16, Lines 9-10 - The conclusion about consistency of results for single observers is dependent on the observer not changing their approach to data gathering as a function of experience gained by data gathering, subsequent training, and/or subsequent interaction with other data gatherers. The conclusion seems too simplistic vs. reality. It might be applicable to "single events" such as one day of fieldwork or a single workshop, but is likely to be less applicable with the passage of time and the occurrence of multiple events, particularly if they have differing goals.

Page 16, Lines 11-13 - Text revisions offered to be less judgmental and to more clearly state "driving philosophies" for "less detailed" vs. "more detailed" participants.

Page 16, Lines 14-15 - The work by others around this point should be cited here and included. Much of it may be in the literature for Psychology, but a useful entry point may be contributions involving Shipley & Tikoff.

Page 16, Lines 29-30 - This "part" sentence is a little odd. It is probably not needed (could place the colon after "collect"). Yet, if it is going to be retained, it should "go large" and not "small" (why focus on folks who do paleostress analysis?). The recommendations are relevant to all persons collecting structural data or utilizing the data products/analyses of others (go large!).

Abstract and Conclusions - Likely will need minor revisions if the main text is revised along these recommendations. ***********************************************

Additional Comment: Please see separately submitted PDF with annotations showing comments suggesting detailed improvements to the text of the main document.

Please also note the supplement to this comment:
https://www.solid-earth-discuss.net/se-2018-135/se-2018-135-RC2-supplement.pdf

[Figure]

**Supplement:**

[revised manuscript text omitted]

---

## Author Comment (AC1) · 25 Mar 2019

**Response to Reviewers**

**Ref: Se-2018-135**

**Title: How do we see fractures? Subjective bias in fracture data collection**
**Journal: Solid Earth**

Our paper has benefitted from two reviewers, Roberto Rizzo (R1) and William Dunne (R2). We also received supportive comments from Stephen Laubach. As we incorporate suggestions for improvement presented in these comments, we will refer to Prof Laubach as Reviewer 3 (R3).

In our response, we refer to the manuscript that was submitted for review as the 'original manuscript', which has been edited following comments from the reviewers to become the 'revised manuscript'.

The following tables documents each of the comments received from each reviewer and our response. For completeness, the reviewer's full original comments are detailed at the end of this document.

We would like to suggest that Bill Dunne is nominated for excellence in reviewing – his review was extremely thorough and very helpful.

**Review R1: Roberto Emanuele Rizzo**

| # | Pg(line) | Comment | Response |
|---|---|---|---|
| 1 | - | In my opinion, a fundamental underlying issue that has not been addressed by the authors directly relates to use of their data to draw conclusions on the accuracy in the parameter estimations of acquisition methods. In particular, do the authors have taken into consideration the possibility that the errors and uncertainties related with subjective biases can scale with the number of fractures measured in the network? Letting interpret a larger fracture dataset to participant would have reduced the uncertainties in the estimation of the fracture attributes, independently of the acquisition method used? | Points 1, 2 and 3 are essentially arguing that our sample size is not large enough to demonstrate an effect robustly. However, the effect discussed by the reviewer of Marginal error is not what we were aiming to investigate. Rather we are looking at the difference in a given sample (e.g. circle) or a population (fracture network) measured by operators using slightly different methods (e.g. detail-oriented operator, vs less detail-oriented operator). Essentially though the sample is the same – different operators are reporting varying data for that sample. If two operators collected larger volumes of data, we maintain that this operator bias would simply roll through into the larger dataset. |
| 2 | - | Although the authors clearly state that it is not in their aim to "collect sufficient fractures to represent the fracture network" and that "the tested scanlines were not designed to be statistically representative" (page 5 lines 28 – 30), at the same time they do "consider the effect of the variation on fracture statistics derived from data collected. . ." (page 3 lines 3 and 4) and they dedicate a Section on the "Effect of subjective bias if the derived fracture statistics" (Section 4). I fear that their conclusions on this specific matter lack of statistical robustness, because of small number of fractures in the samples | However, if an operator changed their level of detail through time this would not hold. An operator might reduce detail through time and speed up in order to meet a deadline. Conversely an operator may become more detailed through time as they become more 'familiar' with a fracture network. If one was measuring a homogeneous fracture network then this might become |

| # | | | |
|---|---|---|---|
| 3 | - | A well-known behaviour in statistics is what is called the 'marginal error': the size (N) of a statistical sample affects the standard deviation (i.e. variability) of the same sample (Moor, D.S., McCabe G.D. "Introduction to the practise of statistics", 1999; pages 294-295; 391-392).

The variability shown in this work suffers the relatively small sample size in the number of fractures interpreted by the participants. The variability in a sample (the spread of the sample distribution) matters as much as bias when building a robust and significant dataset. Because N (i.e. the fracture sample) is the denominator of the sample st.dev. formula ($s = \sqrt{(\sum(x_1 - x)\Theta 2)/(N-1)}$) the st.dev. decreases as N increases. It follows then that having less data in your sample gives more variation (and less precision) in the result of your statistics: it can appear that big variations occur between participants counting, for example, number of fractures, however this spread in the data reduces considerably with the number of fractures that can be counted in the network. To reduce the variability of a statistic a large sample needs to be used: large sample almost always give an estimate that is close to the true value. I understand that reviewing the manuscript in the light this comment may take considerable time (due to re-running tests), therefore I suggested minor revisions for this manuscript; however, I advise the authors to account for these possible biases in their interpretation on fracture statistics throughout the manuscript. | apparent through time. However, when measuring fault-related fracture networks, the networks should be expected to vary spatially, and therefore vary through time during a field campaign. In this case unless measures are taken to explicitly mitigate against a 'drift' in the operator's detail orientation, it may not be possible to distinguish between such drift and actual variation in the fracture network in space (and therefore through the time of the field campaign).

See also reviewer 2 comment 12(19-25) – where the reviewer correctly noted that *"It is the values as reported by the participants rather than the values themselves that is the focus of this work and the connection of the participants to the values should be explicitly maintained through the narrative."*

We have added the following to page 2, line 40 to ensure clarity. "It is the values as reported by the participants rather than the underlying statistics of the measured fracture networks that is the focus of this work."

This is made explicated in Page 6, line 24 to 28 where it now says "In this paper, we are not interested in defining that 'true' value, rather 25 we wish to explore the ranges in reported values from different participants, showing the scale of subjective bias for the collected data, and the factors that affect this range. Therefore, we define the uncertainty, or level of variability, present in fracture data collection and the related statistics as a function of the observers/operators."

We have also reiterated our belief that marginal error is separate from the effect of subjective bias through the addition of a section in '6.4 Wider geoscientific implications' outlining our reasoning and backing this up using the work of Scheiber et al (2015). See page 20, Line 24 to 35. |

| # | Page (line) | Minor comment/edits | Response |
|---|---|---|---|
| 4 | 3(20) | Please check the use of 'Nc': should not be written as 'n-points' in concordance with nomenclature in the following sentence? | This has been changed to be consistent with other literature in this field. |

| 5 | 5(10-11) | Can you please fully explain how you measure connectivity in linear scanlines? Are you using only x- and y-connections? | This is explained in Table 1. We make this more explicit to the reader by referencing Table 1 in this sentence (Page 4 Line 38 to 39). |
|---|---|---|---|
| 6 | 5(24-25) | Can you please write the size of the used circles? In this context, looking at Table 2: why did the size of circle change in different localities? How did you choose the size of the circle? | This was previously alluded to in in section 4, however R1 is correct, this it should be noted in the methodology. We included a sentence to explain this, see Page 5 Line 14 to 16 of revised manuscript. |
| 7 | 8(10-14) | How do the authors assess variability? Does 'variability' refer to the statistical variance within a single scan-circle? To show the variability in your sample, I would suggest to accompany the mean values shown in the tables with variance or standard deviation. The word 'variability' has been used by the authors throughout their text, however 'variation' is never statistically evaluated. | This has been addressed through the addition of a 'Framework to describe results' See R2 comment 1 for elaboration.

We reported the mean values since this is a widely adopted statistic. However, given that the data does not follow a normal distribution, describing the data using the mean values is not statistically appropriate. We have now reported median values in all tables apart from for mean trace length. For clarity, we make this clear in the text, see Pg. 5 line 2 to 3 of revised manuscript. |
| 8 | 9(29) | How do the authors assess the trend? Only visually? | Please see R2 comment 1. |
| 9 | 9(31-32) | How do you evaluate indicators for trends? | Please see R2 comment 1. Lack of trend also visible in Figure 8b, as noted in revised text (Page 11 Line 23). |
| 10 | 12(9-10) | The authors refer to two mean trace length values derived from two participant's measurements, however I could not find these numbers. Can you please indicate to which table are you referring? | Referred readers to Supplementary information which details this Page 12 Line 4 to 5. |
| 11 | 12(11) | Can the authors, please, mention to what this 'R2' values stands for? Is it a coefficient of correlation? | We have added this to section 3.4 'Analytical framework' see Page 6 Line 29 to Page 7 Line 22. |
| 12 | 12(25-26) | Can you please further explain why window sampling is less subjected to biases? | Firstly, the phrasing of this sentence has changed in the revised manuscript, so that this is clearer.

We add a paragraph to explain this here (see page 12 Line 39 to Page 13 Line 2 of revised manuscript) |
| 13 | 12(4-5) | This sentence raises the question: what can be considered a 'tall geologist'? I would suggest to delete this assertion if not fully accompanied with demonstrations and scientific data. | In our text we are not defining what makes a tall or short geologist (!), but that human factors such as height will affect the scale of observation. The tallest in a group will have a wider field of vision than the shortest. Raising such human factors is an important point of this paper, and these differences were observed (qualitatively) in the field. For |

| # | Pg(line) | | response |
|---|---|---|---|
| | | | these reasons, we do not remove this sentence. |
| 14 | 14(30-32) | Would not this be known only after having analysed the whole fracture networks? | Yes. Which is why we suggest removing any unnecessary data *after* data collection, rather than *omitting* these fractures from the data collected. |
| 15 | 27(T4) | For the fracture count, it would be interesting to see the spreading of the data: i.e., the ⏉N (difference between Min and Max). Instead, for the trace length data do you have taken into consideration min and max within each individual/group observation? Is the Min and Max reported into the table a mean of the Min values? Similarly for the Max? | Table 5 has been amended to display the median, and QCV for fracture count. The range was not given in this table, however, is available in the Supplementary information (S8). We do not think reporting the range is appropriate in Table 4 as it would need to either be a range in ranges, or the mean range by participants. Table 5 (Table 4 in the submitted MS) records the minimum, or maximum, value recorded by a participant and does not represent the mean of those values. The reported mean and median values for each scanlines represent the mean values across all participants. QCV represents the variability in that attribute between participants. |
| 16 | 42(F9) | Why does the bin size vary within the same attribute? | It is normal for a histogram bin width to vary in response to the data collected. The number of fractures in each data point changes, and we change the bin width accordingly. Due to this we have kept F9 the same, however, outlined why the bin widths change in the figure caption. (see Page 42 line 5 to 6) |
| **#** | **Pg(line)** | **Minor text changes** | **response** |
| 17 | 2(13-14) | Can you please review this sentence? As written it is not very clear. | Sentence has been rewritten to make this clearer, following suggested edits from R2. See Pg. 2 line 13 to 15 of revised manuscript. |
| 18 | 4(4) | Please check the sentence: is there a 'where' missing between '... a technique' and 'all fractures...'? | Implemented. |
| 19 | 4(14) | I would suggest to add '... and window sampling' to the listed methods: "Trace lengths may be measured directly with linear scanlines and *window sampling*, or estimated..." | Implemented. |
| 20 | 5(18-19) | Can you please add the trending attitude for the sub-vertical joint sent? Is this a third set? | Clarified the wording to avoid confusion about the number of fracture sets. Page 5 Line 6 to 8. |
| 21 | 5(26) | Missed a capital letter 'P' in Participants. For consistency, please consider changing the 'Nc' (throughout the manuscript) to 'n' or *vice-versa*. | Done following minor text edits to clarify this section. See comment #4 above regarding the change from Nc to n-points. |

| 22 | 7(30) | 'Does not' instead of 'doesn't'. | Done (Page 7, Line 41) |
|---|---|---|---|
| 23 | 10(12-13) | 'Participant 11 depicted' instead of 'depicts' and 'Participant 18 and 20' did not…' instead of 'do not', consistently with previous sentence. | Section edited to give greater clarity on the results (Page 10 Line 5 to 11) |
| 24 | 12(22) | Please check 'al', should it not be 'all'? | Section reworded in order to apply the analytical framework. |
| 25 | 14(21) | Please consider changing 'won't' with 'will not' | Incorporated into text edits of the section recommended by R2. |
| 26 | 14(28-30) | Please consider re-phrasing, as written the sentence is a bit obscure | Section edited during revisions of the discussion. |

**Review R2: William M. Dunne**

Comments in black are those provided in written response, comments in blue are those provided on the commented PDF. R2 also edited the document extensively, and we have incorporated many of these edits.

| # | Pg(line) | Comment | Response |
|---|---|---|---|
| 1 | - | **Section 3** - A careful set of data are collected about participant performance for factors such as amount of data collection, type of data collection, patterns of data collection, time taken to collect data, and data collection performance as a function of individual or group data collection. These data are well documented. However, the analysis of this data in Section 3 is somewhat vague with statements such as "reasonably consistent", "a suggestion in the data", "differences are not enough to be confident", or "the trend is very weak". No framework for a quantitative and/or qualitative approach is established at the outset of the data presentation and analysis in this section. Presenting this framework and then utilizing it would be a critical for improving the rigor of the present paper. Presenting the framework will likely lead to similar results and will do so in a manner that creates greater confidence in the results presented in this Section. | Reviewer 2 raises a fair point. In our original manuscript we do not provide a semi-quantitative framework for describing our results.

This links to, and also addresses, several of R1 comments, and also a number of other comments raised by R2.

In response to the reviewers' helpful comments, we provide a semi-quantitative framework to improve the rigour of work, described in section 3.4 Analytical framework (Page 6 Line 29 to Page 7 Line 22).

This framework describes (quantitatively) what we mean, in terms of visual trends in the range, $R^2$ and consistency) by given descriptors.

We now semi- quantitatively assess our data using the following themes: Spatial distribution and node triangle space, Range/variability, co-variance and consistency. All themes are described using a number of descriptors outlined in the text.

Following the reviewer comments, we felt that it was important to use a quantitative approach to assess variably, however, because most data displayed non-normal distributions the use of the standard coefficient of variance was not appropriate. It was decided that the quartile based coefficient of variance would be applied |

| | | | |
|---|---|---|---|
| | | | to all data (QCV = Interquartile range / median). |
| | | | It was still required to describe some of the data qualitatively, for example where participant experience is used, and this is explicitly outlined on Page 7 Line 19 to 22. |
| | | | This framework provided very similar results to those reported in the submitted manuscript and give greater confidence in the results and we would like to thank the reviewer for raising this very valid point. We feel in this paper of all papers we should not be subjective in our analysis of our own data! |
| 2 | - | Page 17, Line 27 to Page 18, Line 5 (End of Discussion) - This text should be replaced by more ambitious text that speaks both more generally than just the mechanics of resolving data gathering differences between observers in the context of "detail" and also connects to real-world situations that apply to the readers beyond just those for the particulars of gathering fracture-related data. So, it is certainly worthwhile constructing experiments that directly test for effects related to subjective bias or operator bias concerning the collection of geological data. Yet, how do experimental results apply to real situations involving data collection? For example, how do the results provide value to an instructor working with a group of students who are performing field data collection for the first time vs. to an individual or team that are applying a rules-based data collection process with specific training prior to the first field deployment to ensure familiarity with the rules-based approach vs. to a computer-based observer utilizing virtual 3-D outcrops from photogrammetric data who has no prior field experience with the data set vs. to building a data set by crowd-sourcing. In this context, the present paper would be a stronger contribution if it explicitly considered the application of its outcomes to real-world circumstances of value and interest to readers. Replacing the existing text at the end of Discussion and embracing this opportunity for expanding the import of the narrative should bring greater recognition to the contribution of the authors and greater interest from the readership. Also, this revised text would address comments made on Page 14 – Line 29, Page 15 – Line 9, and Page 15 – Line 27, where the authors need to extend their work and provide more guidance about the meaning and application of their results. | We are delighted that R2 sees such value in our work! Following your advice, and to better across the message that we present, the discussion section has been rearranged in the revised manuscript as follows;

 6.1: Scanline validity and appropriate data collection method

 6.2 Causes of subjective bias: operator bias and fracture network characteristics

 6.3 Recommendations for reducing subjective bias

 6.4 Wider geoscientific implications.

 The restructuring enabled us to streamline our argument, include several key pieces of literature which were missing in the original text and expand the finale of the discussion, drawing on ideas raised by R2 and the supportive comment of R3.

 The key areas where we expanded our discussion was on the 'Human factors' section (Page 14 Line 32 to Page 16 Line 16), how these will effect group work (Page 16 Line 17 to Line 42). We also expanded our discussion on whether pre-set rules should be implemented (Page 17 Line 25 to Page 18 Line 6).

 The key references added to strengthen our discussion included works from the cognitive style literature, along with a number of paper which came to our attention during the discussion period as outlined in comment 3.
 . |

| 3 | - | The Discussion also has a few key locations where the work of others should be included and considered. Please see "Other Comments" for details. | A number of key references have been added as outlined below, by subject [Page & Line numbers indicated]; |
| --- | --- | --- | --- |
| | | | *Previous work on inter-operator variability:* |
| | | | Burns and Brown 1978 [Pg 2, Ln 30] |
| | | | Burns et al 1976 [Pg 2, Ln 30; Pg 2 Ln 36] |
| | | | Hillier et al 2015 [Pg 2, Ln 30] |
| | | | Huntington and Raiche, 1977 [Pg 2, Ln 30; Pg 20, Ln 32] |
| | | | Peacock et al (*in press*) [Pg 14, Ln 33; Pg 20, Ln 10] |
| | | | Scheiber et al., 2015 [Pg 1, Ln 31; Pg 1, Ln 34; Pg 15, Ln 21; Pg 15, Ln 22; Pg 20, Ln 12; Ph 20, Ln 29] |
| | | | *Cognitive style (individual and group)* |
| | | | Aggarwal and Woolley, 2013 [Pg 16, Ln 36] |
| | | | Armstrong et al 2012 [Pg 15, Ln 42] |
| | | | Chan, 1996 [Pg 15, Ln 40] |
| | | | Cheng et al. 2003 [Pg 16, Ln 33] |
| | | | Chilton et al., 2005 [Pg 15, Ln 43] |
| | | | Fuller and Kaplan, 2004 [Pg 15, Ln 40] |
| | | | Jung, 2016 [Pg 15, Ln 28; Pg 15, Ln 38] |
| | | | Myers, 1962 [Pg 15, Ln 29] |
| | | | Myers et al 1998 [Pg 15, Ln 30] |
| | | | Peterson et al, 2009 [Pg 16, Ln 1] |
| | | | Pounds and bailey, 2001 [Pg 14, Ln 41] |
| | | | Shipley and Tikoff, 2016 [Pg 15, Ln 18-19; Pg 16, Ln 6-7] |
| | | | Witkin and Goodenough, 1977 [Pg 15, Ln 28] |
| | | | *Mental models* |
| | | | Gibson et al., 2016 [Pg 15, Ln 14] |
| | | | Macrae et al, 2016 [Pg 14, Ln 28; Pg 18, ln 34] |
| | | | *Fracture references* |
| | | | Bonnet et al, 2001 [Pg 17 Ln 35] |
| | | | Hooker et al. 2014 [Pg 17, Ln 31] |
| | | | Laubach et al, 2018 [Pg 3, Ln 32, *changed from editorial to full article*] |
| | | | Sanderson and Nixon, 2018 [Pg 1, Ln 11] |
| | | | *Wider references/experience* |
| | | | Dickinson et al, 2010 [Pg 15, Ln 25; Pg 21, Ln 10]. |

| | | | Doyle and Paget, 1892 [Pg 21, Ln 16] Dunham et al. 2004 [Pg 15, Ln 25] |
|---|---|---|---|
| | **Page (line)** | **Minor comment/edits (Blue & italic represents comments on the attached PDF).** | **Response** |
| 4 | 2(8-11) | It seems odd to list four methods and only provide citations for one of the four methods. Quality citations exist for all of the methods and the manuscript would be more useful for readers if each method was paired in the text here with at least two appropriate citations. | We agree. We have added in references to each method |
| 5 | 4(3) | It would be useful to explicitly state for the reader why plotting topology data in a triangular diagram is useful. | We have added a sentence to the revised text. See Page 3 line 36 to 39 of the revised manuscript. |
| 6 | 7(4-6) | Suggestions offered in the annotated PDF for this review to improve the clarity and purpose of this text related to methodology and then the approach to statistical characterization. | Many of the suggested changes have been incorporated, or have been addressed by our response to comment #1. |
| 7 | 7(19) | How is "reasonable amount of consistency" quantitatively defined or qualitatively recognized? | See response to comment #1. |
| 8 | 8(23-24) | How often were participants "internally consistent"? What is the measure/criterion for "internal consistency"? What is the measure/criterion for defining the occurrence of "often"? | See response to comment #1, which addresses this comment. |
| 9 | 9(6) | What defines or qualifies "varied considerably"? What is the standard or basis for comparison? | See response to comment #1, which addresses this comment. |
| *10* | *10(30)* | *The meaning of "the joint highest" is not clear.* | Removed joint for clarity. |
| *11* | *10(30)* | *What defines "a suggestion in the data"? This characteristic or attribute is not defined or explained. More explanation is needed here.* | See response to comment #1, which addresses this comment. |
| *12* | *11(3-4)* | *While I agree, the paper would be better if the authors explained why they believe "differences are not enough to be confident that this is due to working in groups rather than differences in the fracture network".* | Sentence added |
| *13* | *11(15)* | *"correlation is weak" - Is this statement backed by statistical analysis or is that a judgment call by the authors. Additional explanation is needed here.* | See response to comment #1, which addresses this comment. |
| *14* | *11(29)* | *What is the statistical or qualitative basis for stating that "however the trend is very weak". Further explanation is needed.* | See response to comment #1, which addresses this comment. |

| | | | |
|---|---|---|---|
| 15 | 12(25-31) | *This paragraph is a key paragraph in the narrative of the paper. It really needs to be clearly written and drive home key points in a manner that is easy for readers to fully understand. I encourage you to give this paragraph its due and provide a revision that does the paper justice and hopefully, leads to a more engaged readership with a stronger understanding of your work.* | We have re-written this paragraph to improve readability, and in fact split it into two: one about which techniques suffers least form subjective bias and the second about which statistic seems more robust in the face of subjective uncertainty. |
| 16 | 12(25) | *The narrative should be more direct and avoid the use of the word "suggest" that is vague and lacking in framework.* | See response to comment #1, which addresses this comment. |
| 17 | 12(26) | *Revision offered to the latter part of this sentence to clearly and explicitly relate "spreads" in Table 7 to main text and then clearly state the interpretation that the authors have derived from considering this population of spreads as a function of sampling method and subjective bias.* | See response to comment #1, which addresses this comment. |
| 18 | 12(27) | *Apologies for my confusion but how does "most robust" relate to "displaying considerable variability". Previously, "robustness" related to similar reported values by participants with limited variability. What am I missing in the text here?* | This has been rephrased to avoid confusion. |
| 19 | 13(11-12) | *Why does the variation in reported values by different participants for the same sampling method directly correlate to the size of the tool for the method, and hence the sample size. Is it not the case that if a statistically valid sample size is going to be collected by each participant that both the sample size and the tool size need to be specified prior to measurement by all participants? Or is there an implication that the dimension of the scanline or window needs to be set on the basis of minimum values to be expected from the range of values due to subjective bias? I suspect that the core problem with this clause of this sentence is that it should come at the end of the paragraph after the key observations are offered, so that a summative comment can then be made and justified. The comment does need some improvement in text to provide greater clarity.* | In response to this comment we have clarified the text describing how participants effect the sampling strategy (Page 13 line 20 to 34)

Our recommendations are now clearly outlined in a separate section (Page 18, Line 25 to 27 & aspects of our other recommendations e.g. Page 18, line 39 to Page 19, line 2). |
| 20 | 14(2) | *the participants are not less or more detailed. Their observations are. Suggested text revisions are offered to clarify this point. This approach should be adopted at other appropriate locations in the narrative.* | We agree and have looked to adopt this throughout the MS. |
| 21 | 14(5) | *A clear conclusion and useful point is reached at the end of this paragraph.* | Following your recommendation, our discussion has been rearranged to make |

| | | | |
|---|---|---|---|
| | | *What are recommendations for operationalizing the observation with respect to a future data-sampling campaign? How does the needed level of detail for a campaign fit into this operationalization? After all, not every sampling campaign necessarily needs the same level of detail as a function of campaign goals. Or put differently, more or less detail is not always best!* | it clearer what our recommendations are and the limitations (e.g. not always best to collect more detail if it does not fit the purposes of the study). |
| 22 | 4 | *"First" Section 4.2 - This section is out of place and effectively encompassed in the later sections including particularly Section* | Removed from revised MS. |
| 23 | 14(29) | *So, what should be the rules for data gathering with respect to dealing with interpretation in the presence of limited exposure for a data-gathering campaign?* | Added to the recommendation section (Page 19, Line 3 to 12). |
| 24 | 15(9) | *So, is training an answer and recognizing the goals of the data-gathering campaign an important constraint?* | We instead feel that a clear communication of the suggested methods, as outlined in our steps 1-7, is key (Page 20 line 17 to 18). |
| 25 | 15(22-23) | *Is this suggestion about a preference for using field-based data rather than photo-based data a first occurrence in the literature? If not, prior work should be cited and most likely, briefly discussed.* | We have not been able to find any specific literature on this, and believe it will be subject to professional opinion. We have made the argument more balanced in our recommendations to cover such things as bad weather which can be detrimental to field-based data collection (Page 19 Line 13 to 22) |
| 26 | 15(27) | *The narrative should be expanded to provide guidance or cautions based in the outcomes of the paper with respect to the use of data from UAV-based data-gathering strategies......* | This has been added to the section '6.4 Wider geoscientific implications'. See Page 20 Lines 20 to 34 of the revised manuscript. |
| 27 | 16(9-10) | *The conclusion about consistency of results for single observers is dependent on the observer not changing their approach to data gathering as a function of experience gained by data gathering, subsequent training, and/or subsequent interaction with other data gatherers. The conclusion seems too simplistic vs. reality. It might be applicable to "single events" such as one day of fieldwork or a single workshop, but is likely to be less applicable with the passage of time and the occurrence of multiple events, particularly if they have differing goals.* | R2 makes a very valid point, and while we believe users are likely to have their 'go to' data collection style that this may change with differing goals or specific training. We have added a sentence regarding this as a caveat of the findings & that if it is used then you should recheck data collection behaviours regularly. The text has been amended accordingly (Page 20 Line 10 to 16). |
| 28 | 16(14-15) | *The work by others around this point should be cited here and included. Much of it may be in the literature for* | References have been added, see Comment #3. |

| # | Pg(line) | | response |
|---|---|---|---|
| | | *Psychology, but a useful entry point may be contributions involving Shipley & Tikoff.* | |
| 29 | 16(29-30) | *This "part" sentence is a little odd. It is probably not needed (could place the colon after "collect"). Yet, if it is going to be retained, it should "go large" and not "small" (why focus on folks who do paleostress analysis?). The recommendations are relevant to all persons collecting structural data or utilizing the data products/analyses of others (go large!).* | The part sentence has been removed as part of the restructuring of the discussion. |
| 30 | 17(27) to 18(5) | *This text should be replaced by more ambitious text that speaks both more generally than just the mechanics of resolving data gathering differences between observers in the context of "detail" and also connects to real-world situations that apply to the readers beyond just those for the particulars of gathering fracture-related data.* | The end of the discussion has been broadened through the addition of section 6.4 Wider geoscientific implications. We also make it clear that this work expands far beyond that of fracture analysis through the closing sentence of the manuscript: *"As the implications of our findings has relevance for a range of observation-based sciences beyond geoscience, from digital mapping to Big Data, our study is, ultimately, a call for further work in this area."* |
| 31 | Abs & Concl. | *Abstract and Conclusions - Likely will need minor revisions if the main text is revised along these recommendations.* | Conclusion & abstract have been rewritten following changes to the MS. |
| # | Pg(line) | Minor text changes | response |
| 32 | 2(11-14) | The annotated PDF for this review of the paper provides suggestions for strengthening the statement of the purpose of this contribution. | Thank you. We have added to this section to sing about our work more loudly! |
| 33 | 2(25-29) | These two sentences consider observational resolution and limitations to the quality of observations that can be made as a function of the exposed rock. These two points would relate to both objective and subjective uncertainty, and as such seem out of place in the narrative flow. Given the text in Lines 22 to 25 that is focusing on subjective uncertainty, any text, if any is needed, after Line 25 in this paragraph should only consider factors the relate to subjective uncertainty. It might be best to eliminate this text and just continue with the text in the new paragraph starting on Line 30 that focuses on the subjective uncertainty and further introduces the paper. | These sentences are removed, as suggested. |
| 34 | 3(15-18) | Suggested text revisions are offered to completely and correctly state the contribution of Zeeb et al., (2013) to defining the number of measurements needed to | Thank you. Done. |

| | | provide an estimated value for a characteristic that is statistically significant. | |
|---|---|---|---|
| 35 | 6(15) | Specify the dimensions of A3 paper as it is not a universally used paper size. | Added, see Page 5 Line 36 to 37. |
| 36 | 7 (10-11) | *The second part of this sentence is not clear and does not seem necessary, so deleted.* | Agree. |
| 37 | 7 (14-16) | *A suggested text revision is offered to succinctly state what is needed.*

*Also, no need to introduce Section 4, because if the narrative is sufficient, the narrative will flow into Section 4 and the reader will understand without needing this "mile marker" sentence at the end of Section 2.* | Agree.

We disagree here (we think the reviewer was referring to the short introduction to the discussion – originally section 5 in the submitted MS and now section 6 due to correction of mis-numbering. We prefer to retain an introduction, partly because this is a long and complex paper and some readers will move straight to the discussion. |
| 38 | 7(30)-8(4) | Suggestions are offered to improve the precision and the clarity of the text describing locations and causes of increased observational uncertainty as a function of the participants for Scanline 6. | Thank you. Incorporated into the text. |
| 39 | 8(4) | *The replacement text makes the more important point that does need to be stated.* | Thank you. Incorporated into the text. |
| 40 | 8(10-19) | *Suggestions offered to improve the precision and clarity of the narrative.* | Thank you. Incorporated into the text. |
| 41 | 9(14) | *Words added because an operational preference for a participant to report more smaller fracture traces does not necessitate that the reported actual small fracture traces.* | The suggested edits have been incorporated into the revised text. |
| 42 | 9(16) | *Suggested text modification to provide greater clarity about the meaning of this sentence.* | The suggested edits have been incorporated into the revised text. |
| 43 | 12(5) | *Unneeded sentence. The narrative is more effective if the story is simply told without these "mile markers".* | Agree, removed as suggested. |
| 44 | 12(19-25) | *Suggested revisions offered to focus the narratives on the reporting by participants. It is the values as reported by the participants rather than the values themselves that is the focus of this work and the connection of the participants to the values should be explicitly maintained through the narrative.* | The suggested edits have mostly been incorporated into the revised text. |

| 45 | 12(29) | *provide a more complete explanation with respect to the actions of the participants in the sentence here.* | This has been rephrased as suggested. |
|----|--------|---------------------------------------------------------------------------------------------------------------|---------------------------------------|
| 46 | 12 (29-30) | *Suggested text revisions to increase narrative clarity and more explicitly link to participant observations.* | The suggested edits have mostly been incorporated into the revised text. |
| 47 | 13(11) | *important to add "different" as a participant reported a value for a particular data-gathering event for a single sampling tool, so no "difference" could occur for single participants and single sampling events.* | Due to the re-structuring of the discussion some of these points have been removed, or incorporated into the next text. We would like to thank R2 for the incredibly diligent and extensive review, which has led to a much tidier discussion. |
| 48 | 14(9) | *Perhaps not best to use the term "invalid". Were the participants provided with guidance that they needed to exceed a certain number of observations for their data sample to be considered to be statistically valid. If not, they should not be judged as "invalid". Alternative text is offered.* | |
| 49 | S4 | *Subsections misnumbered because two Section 4.2's are identified.* | |
| 50 | 15(2) | *What is this notation? Is it the same as "pers. comm." or does it reference a particular presentation or paper. The meaning of "n.d." is not clear.* | |
| 51 | 15(6) | *This insertion of text is quite important to clearly making the point for readers.* | |
| 52 | 15(25) | *important addition to the end of the sentence.* | |
| 53 | 16(11-13) | *Text revisions offered to be less judgmental and to more clearly state "driving philosophies" for "less detailed" vs. "more detailed" participants.* | |
| 54 | -- | *Overall changes to text in the PDF not included as specific comments* | We want to thank R2 for an extensive and detailed edit of the manuscript's text which has in many places dramatically improved how we deliver the message. The majority of these edits have been incorporated into the revised MS. |

**Reviewer R3: Stephen Laubach**

These were posted as a Supportive Comment.

|  | Comments | Response |
|---|---|---|
| 1 | Some of the problems documented in this paper, like how to objectively document length, are ones that need further thought.

Marrett (probably in Marrett and Ortega) concluded that length and connectivity were too subjective to measure meaningfully, which is part of the reason he advocated linear scanlines and careful aperture size measurement (as in Ortega et al. 2006). It should be standard practice to specify aperture size cut offs in linear scanline data, and using cutoffs and other rules may be useful for acquiring reproducible length data sets. | We agree that questions surrounding subjective bias, while in part tackled in this paper, remain and that further work by the fracture community is required if we are going to tackle the issue. |
| 2 | In a report on subjectivity in fracture data collection, there are some other important problems that should at least be mentioned. As S. Holmes said in Silver Blaze, 'I saw it because I was looking for it' and the fracture community seems to have some highly obscuring blinders on when it comes to some aspects of fractures.

In my opinion because we're used to looking at fracture patterns in a certain limited way (Laubach et al. 2010, J. Struct. Geol.) For example, if we're interested in constructing DFN models for fractures at depth, are barren joints in outcrop a useful structure to measure in the first place? Fractures in core commonly have some amount of mineral lining; they've been subject to hot fluids for sometimes millions of years (references in Lander & Laubach 2015, GSA Bulletin).

The problems with the specific methods we use may not be as important as the unexamined subjectivity of the choices we make about which outcrop to study. | R3 makes a very valid point that this is a particular problem for fracture data collection and that the quote from Silver Blaze is very apt in the description of subjective bias. We have added it to the start of our conclusion (Pg 21 Line 16).

We agree that this is another form of subjective bias which can creep into fracture data collection, however, feel that it comes under the remit of the 'mental model' or 'what features count' sections. We have included a reference here w.r.t. this in the relevant section.

We feel this is a very important point regarding fracture data collection in general, however, we feel that it is beyond the scope of this paper and something that requires further work in future publications. |
| **Page**

**(line)** | **Minor comment/edits** | **Response** |
| 2(9) | In general I think the advantages of circular scanlines tend to get over sold, at least as applied to sedimentary rocks outside of intensely deformed fold and fault zones. For regional fractures, which may have a few or only one simple fracture set with sparse, widely spaced fractures, linear scanlines may be the only way to get meaningful data on fracture occurrence; they are not subject to the interpretation problem of picking fracture 'ends' that affect 2D approaches, and they are directly comparable to the 1D data sets available from wellbores. And there are | In response to this comment we ensured that we have discussed the methods in equal measures. This work does not attempt to understand the relative differences between methods, who's relevance is dependent on both the fracture network and the studies aims, but instead how data collected by each method is effected by subjective bias. We have attempted to improve how we provide our recommendations such that all methods are included, along with being more balanced in our description of linear scanlines. |

| | | |
|---|---|---|
| | methods available for looking at fracture spatial arrangement in a rigorous way (i.e., Marrett et al. 2018, J. Struct. Geol.) | |
| 2(25) | And the diagenesis of the fracture network. In many sedimentary rock fracture systems, diagenesis is the principle control on fracture network connectivity (i.e., Olson et al. 2009, AAPG Bulletin, as you cite later) and this is often overlooked by structural geologists with their mechanics and geometry disciplinary blinders on. This is a great example of 'subjective uncertainty'. | We agree that this is another form of subjective bias which can creep into fracture data collection, however, feel that it comes under the remit of the 'mental model' or 'what features count' sections. |
| 3(30) | The actual overview article should be cited (Laubach et al. 2018) rather than the very short editorial.

> Laubach, S.E., Lamarche, J., Gauthier, B.D.M., Dunne, W.M., and Sanderson, D.J., 2018. Spatial arrangement of faults and opening-mode fractures. Journal of Structural Geology 108, 2-15. doi.org.10.1016/j.jsg.2017.08.008 | We apologise for this and have amended the citation, thank you for bringing this to our attention. |
| 4(10) | I agree that trace length is vital to measure, and it's a fracture parameter that can only come from outcrops. But the biggest limitation, and one that seems to be in a blind spotâ˘Tis the finite size of outcrops. Production and tracer data from the subsurface ˘ show that fractures capable of rapidly transmitting fluids can be really long - kilometers long probably in some instances. Outcrops of such size that are also good analogs for subsurface fractures are rare. An example, though, is shown in Li et al. (2018, J. Struct. Geol.) where extremely long fracture trace lengths are visible. The finite size of good outcrop analogs is a big challenge if the aim is guiding DFNs. Is it part of 'subjective uncertainty' what we settle for in terms of outcrop type? | While we agree that this could be seen as a form of bias in fracture data collection, it is beyond the scope of the paper and should be considered in future publications. |
| 4(20) | I think it's worth appreciating that the reason Marrett focused on aperture measurements rather than 'length' was that he appreciated that determining 'length' was (and is) subjective. | As aperture was not taken into account in this study, which is a potential limitation of this work, we don't feel we can adequately cover this in the discussion. |
| 4(27) | But the concept of 'number of fractures' can also be quite subjective. For example, examined microscopically, most opening-mode fractures show evolution by linkage. Where does one fracture start or end?

And specifying fracture 'size' in connection with defining intensity should be standard practice, and should be noted in contexts like this, following the work for example of Ortega et al. (2006). Not to do so may be another hidden, subjective bias for the following reason. 'Joints' (that is, barren opening-mode fractures) typically have a very narrow aperture size range, so why bother to try to measure aperture sizes? But many fracture populations found in core from sedimentary basins have been known since the late 1980s to typically have wide aperture size ranges. It's possible, | R3 makes a very good point, and we have expanded out discussion into the use of pre-set data cut-offs (Pg 17 Ln 28 to Pg 18 Line 6) and have added this to our recommendations (Pg 20, Ln 1 to 16). Although the work on aperture is certainly interested and appropriate to this work it was decided to not add this to the discussion. |

| | | |
|---|---|---|
| | therefore, to use the aperture size distribution, which can be measured in outcrop and the subsurface target, to decide how similar outcrop and target likely are. Moreover, if you don't account for size in defining what you measure, you can get wilding different results for intensity. These observations are partly what motivated Ortega et al.'s work. It should be standard practice to specify a size measure when describing 'intensity'. And the potential bias of working on easily visible but possibly misleading joints as guides to the subsurface is a topic that a report on subjectivity in fracture studies ought to at least consider. | |
| 5(10) | Again, this measure of 'connectivity' ignores cement. | We feel this is a question of method, and not the subjective bias introduced by operators on the process. |
| 6(5) | I wonder how many participants had experience describing fractures in core? | I wonder, too! But this data wasn't collected. |
| 9(4) | This seems highly likely. In some of the older fracture 'topology' literature (which seems to have been missed by recent papers) this scale-of-observation effect was explicitly taken into account in connectivity measures. It's another area where it should be a matter of course to take size into account in descriptions. | The importance of scale of observation has been heavily made in this MS, with special reference to the statistics covered in this work. |
| 9(7) | Some problems like this can be taken into account by explicitly specifying size cut offs, a procedure that is a regular part of scanline studies focused on aperture size distributions (e.g., Ortega et al. 2006; Hooker et al. 2014). | This has been incorporated into the discussion of the paper, see comment 4(27). |
| 15(15) | These problems can be minimized with linear scanlines and explicit thresholds. Restricts you to measuring aperture sizes, though, so the problems remain for 'defining' length. | We want to remain as broad as we can w.r.t. to the methods used and such have chosen to not include this. |
| 15(19) | It would really introduce problems to try to determine if fractures in outcrop are fluid conduits or not. Some of the best outcrop analogs for the subsurface may have completely mineral filled fractures: they are fossilized fracture systems. In outcrop, the open, fluid conductive fractures may preferentially be obscured by vegetation, etc. | This point is similar to the mental model section and will inform the participants conceptual model. |
| 17(30) | Fractures that are 'not connected' are only unimportant for flow if the host rock is completely tight. The arrangement and length distribution - including that of small fractures - is important if the rock has finite porosity and permeability (the typical situation for even 'tight' sedimentary rocks). See Philip et al. 2005, SPE Reservoir Evaluation & Engineering 8/4, 300-309. | |

---

## Author Comment (AC2) · 25 Mar 2019

Dear Dr. William Dunne,

We would like to extend our greatest thanks to you for your extensive and hugely helpful review of our manuscript. Your comments have certainly helped improve the message we want to present, along with improving the rigor of our science such that we weren't 'subjective' in our interpretation of 'subjective bias'. In response to your review we have undertaken a restructuring of the Discussion section, and added a number of key references which help strengthen our argument. Please find our full reviewer response attached as supplementary information.

Kind regards, Billy J Andrews

[Figure]

Please also note the supplement to this comment:
https://www.solid-earth-discuss.net/se-2018-135/se-2018-135-AC2-supplement.pdf

---

## Author Comment (AC4) · 25 Mar 2019

[revised manuscript text omitted]

**(a)**

| | P | Count Nc C8 | C5 | C1 | i-node C8 | C5 | C1 | y-node C8 | C5 | C1 | x-node C8 | C5 | C1 | Time (s) Nc C8 | C5 | C1 | Node C8 | C5 | C1 |
|---|---|---|---|---|---|---|---|---|---|---|---|---|---|---|---|---|---|---|---|
| Workshop 1 | 1 | 23 | 11 | 21 | 4 | 9 | 2 | 22 | 24 | 32 | 12 | 2 | 7 | 78 | 70 | 68 | 540 | 324 | 337 |
| | 2 | 11 | 8 | 16 | 2 | 4 | 2 | 1 | 4 | 1 | 3 | 3 | 8 | 107 | 59 | 99 | 378 | 317 | 259 |
| | 3 | 24 | 14 | 20 | 5 | 12 | 1 | 60 | 34 | 38 | 10 | 2 | 7 | 46 | - | 72 | 460 | 1177 | 447 |
| | 4 | 22 | 12 | 16 | 6 | 3 | 0 | 28 | 17 | 18 | 9 | 1 | 7 | 106 | 53 | 50 | 602 | 333 | 119 |
| | 5 | 10 | 7 | 14 | 5 | 3 | 1 | 1 | 7 | 5 | 2 | 1 | 5 | 83 | 70 | 70 | 172 | 150 | 120 |
| | 6 | 25 | 14 | 23 | 4 | 5 | 12 | 27 | 26 | 29 | 12 | 1 | 11 | 52 | 32 | 51 | 312 | 330 | 416 |
| | 7 | 20 | 11 | 17 | 2 | 3 | 0 | 13 | 20 | 16 | 9 | 2 | 10 | 120 | 30 | 60 | 480 | 300 | 300 |
| | 8 | 25 | 16 | 19 | 6 | 7 | 3 | 26 | 12 | 20 | 15 | 6 | 8 | 36 | 28 | 38 | 150 | 150 | 211 |
| | 9 | 25 | 14 | 16 | 5 | 5 | 0 | 33 | 24 | 12 | 12 | 4 | 5 | 180 | 120 | 60 | 780 | 480 | 240 |
| | 10 | 21 | 12 | 18 | 2 | 5 | 0 | 23 | 19 | 19 | 22 | 1 | 4 | 29 | 20 | 49 | 171 | 186 | 141 |
| | 11 | 24 | 18 | 19 | 11 | 14 | 2 | 47 | 31 | 27 | 10 | 2 | 5 | 47 | 41 | 36 | 242 | 184 | 125 |
| Workshop 2 | 12 | 24 | 13 | 18 | 4 | 4 | 2 | 42 | 26 | 25 | 8 | 1 | 6 | 102 | 298 | 295 | 1200 | 235 | 290 |
| | 13 | 26 | 18 | 25 | 15 | 32 | 6 | 45 | 41 | 34 | 18 | 6 | 10 | 180 | 60 | 60 | 1380 | 900 | 540 |
| | 14 | 28 | 12 | 21 | 4 | 7 | 1 | 23 | 16 | 18 | 9 | 1 | 9 | 109 | 80 | 107 | 705 | 451 | 538 |
| | 15 | 25 | 16 | 22 | 2 | 5 | 1 | 31 | 32 | 34 | 16 | 5 | 8 | 129 | 64 | 80 | 864 | 737 | 528 |
| | 16 | 24 | 13 | 19 | 5 | 13 | 1 | 14 | 14 | 20 | 12 | 0 | 4 | 105 | 89 | 230 | 660 | 600 | 259 |
| | 17 | 19 | 11 | 20 | 3 | 6 | 2 | 20 | 15 | 13 | 13 | 7 | 14 | 94 | 58 | 48 | 622 | 310 | 509 |
| | 18 | 26 | 12 | 19 | 3 | 2 | 0 | 19 | 13 | 10 | 15 | 2 | 7 | 134 | 71 | 84 | 504 | 235 | 186 |
| | 19 | 22 | 9 | 20 | 4 | 4 | 3 | 26 | 14 | 32 | 8 | 1 | 4 | 210 | 112 | 176 | 598 | 350 | 430 |
| | 20 | 16 | 12 | 18 | 1 | 2 | 1 | 5 | 23 | 18 | 5 | 2 | 6 | 45 | 240 | 254 | 125 | 325 | 217 |
| | 21 | 25 | 14 | 22 | 4 | 7 | 4 | 6 | 10 | 12 | 18 | 11 | 8 | 55 | 33 | 45 | 295 | 237 | 289 |
| | 22 | 18 | 11 | 13 | 5 | 3 | 0 | 7 | 11 | 4 | 7 | 1 | 10 | 98 | 131 | 74 | 730 | 517 | 550 |
| | 23 | 25 | 11 | 15 | 16 | 11 | 6 | 8 | 9 | 10 | 6 | 2 | 7 | 120 | 60 | 120 | 300 | 120 | 540 |
| | 24 | 22 | 12 | 11 | 4 | 3 | 0 | 7 | 7 | 9 | 12 | 2 | 6 | 120 | 120 | 120 | 600 | 180 | 240 |
| | 25 | 23 | 12 | 16 | 2 | 2 | 0 | 18 | 11 | 21 | 6 | 0 | 6 | 70 | 20 | 40 | 240 | 60 | 180 |
| | 26 | 32 | 12 | 17 | 8 | 11 | 2 | 29 | 14 | 25 | 12 | 3 | 8 | 121 | 34 | 46 | 458 | 165 | 138 |
| | 27 | 20 | 12 | 15 | 4 | 2 | 0 | 21 | 18 | 12 | 9 | 0 | 7 | 52 | 25 | 32 | 527 | 252 | 213 |
| | 28 | 16 | 12 | 13 | 1 | 0 | 0 | 7 | 9 | 5 | 9 | 1 | 10 | 46 | 21 | 15 | 30 | 60 | 82 |
| | 29 | 27 | 14 | 21 | 8 | 9 | 1 | 22 | 18 | 25 | 13 | 1 | 10 | 240 | 90 | 180 | 1440 | 1050 | 1140 |

**(b)**

| | G | Count Nc C4 | C3 | i-node C4 | C3 | y-node C4 | C3 | x-node C4 | C3 | Time (s) Nc C4 | C3 | Node C4 | C3 |
|---|---|---|---|---|---|---|---|---|---|---|---|---|---|
| Workshop 1 | 1 | 14 | 22 | 7 | 20 | 17 | 24 | 3 | 11 | 330 | 90 | 521 | 521 |
| | 2 | 13 | 18 | 11 | 4 | 11 | 19 | 1 | 6 | 62 | 82 | 324 | 208 |
| | 3 | 18 | - | 19 | - | 27 | - | 4 | - | 97 | - | 405 | - |
| | 4 | 11 | 18 | 9 | 5 | 9 | 23 | 2 | 5 | 60 | 73 | 357 | 332 |
| | 5 | 18 | 21 | 9 | 6 | 6 | 23 | 4 | 5 | 110 | 55 | 420 | 312 |
| Workshop 2 | 6 | 18 | 23 | 14 | 11 | 13 | 17 | 1 | 6 | 120 | 60 | 600 | 360 |
| | 7 | 18 | 18 | 5 | 3 | 27 | 22 | 3 | 3 | 129 | 129 | 720 | 600 |
| | 8 | 14 | 14 | 5 | 7 | 14 | 18 | 1 | 4 | 323 | 713 | 115 | 143 |
| | 9 | 12 | 16 | 5 | 16 | 11 | 22 | 1 | 3 | 184 | 389 | 445 | 168 |
| | 10 | 10 | 16 | 5 | 4 | 5 | 13 | 2 | 4 | 116 | 113 | 290 | 465 |
| | 11 | 12 | 18 | 5 | 2 | 8 | 11 | 0 | 10 | 300 | 240 | 120 | 360 |
| | 12 | 17 | 16 | 23 | 54 | 8 | 22 | 3 | 4 | 64 | 52 | 140 | 205 |

**Key**

| Rank for Nc and node counts | Rank for Nc and node time |
|---|---|
| Lowest | Fastest |
| Medium | Medium |
| Highest | Slowest |

| (a) | P | Count n-point C8 | C5 | C1 | i-node C8 | C5 | C1 | y-node C8 | C5 | C1 | x-node C8 | C5 | C1 | Time (s) n-point C8 | C5 | C1 | Node C8 | C5 | C1 |
|---|---|---|---|---|---|---|---|---|---|---|---|---|---|---|---|---|---|---|---|
| Workshop 1 | 1 | 23 | 11 | 21 | 4 | 9 | 2 | 22 | 24 | 32 | 12 | 2 | 7 | 78 | 70 | 68 | 540 | 324 | 337 |
| | 2 | 11 | 8 | 16 | 2 | 4 | 2 | 1 | 4 | 1 | 3 | 3 | 8 | 107 | 59 | 99 | 378 | 317 | 259 |
| | 3 | 24 | 14 | 20 | 5 | 12 | 1 | 60 | 34 | 38 | 10 | 2 | 7 | 46 | - | 72 | 460 | 1177 | 447 |
| | 4 | 22 | 12 | 16 | 6 | 3 | 0 | 28 | 17 | 18 | 9 | 1 | 7 | 106 | 53 | 50 | 602 | 333 | 119 |
| | 5 | 10 | 7 | 14 | 5 | 3 | 1 | 1 | 7 | 5 | 2 | 1 | 5 | 83 | 70 | 70 | 172 | 150 | 120 |
| | 6 | 25 | 14 | 23 | 4 | 5 | 12 | 27 | 26 | 29 | 12 | 1 | 11 | 52 | 32 | 51 | 312 | 330 | 416 |
| | 7 | 20 | 11 | 17 | 2 | 3 | 0 | 13 | 20 | 16 | 9 | 2 | 10 | 120 | 30 | 60 | 480 | 300 | 300 |
| | 8 | 25 | 16 | 19 | 6 | 7 | 3 | 26 | 12 | 20 | 15 | 6 | 8 | 36 | 28 | 38 | 150 | 150 | 211 |
| | 9 | 25 | 14 | 16 | 5 | 5 | 0 | 33 | 24 | 12 | 12 | 4 | 5 | 180 | 120 | 60 | 780 | 480 | 240 |
| | 10 | 21 | 12 | 18 | 2 | 5 | 0 | 23 | 19 | 19 | 22 | 1 | 4 | 29 | 20 | 49 | 171 | 186 | 141 |
| | 11 | 24 | 18 | 19 | 11 | 14 | 2 | 47 | 31 | 27 | 10 | 2 | 5 | 47 | 41 | 36 | 242 | 184 | 125 |
| Workshop 2 | 12 | 24 | 13 | 18 | 4 | 4 | 2 | 42 | 26 | 25 | 8 | 1 | 6 | 102 | 298 | 295 | 1200 | 235 | 290 |
| | 13 | 26 | 18 | 25 | 15 | 32 | 6 | 45 | 41 | 34 | 18 | 6 | 10 | 180 | 60 | 60 | 1380 | 900 | 540 |
| | 14 | 28 | 12 | 21 | 4 | 7 | 1 | 23 | 16 | 18 | 9 | 1 | 9 | 109 | 80 | 107 | 705 | 451 | 538 |
| | 15 | 25 | 16 | 22 | 2 | 5 | 1 | 31 | 32 | 34 | 16 | 5 | 8 | 129 | 64 | 80 | 864 | 737 | 528 |
| | 16 | 24 | 13 | 19 | 5 | 13 | 1 | 14 | 14 | 20 | 12 | 0 | 4 | 105 | 89 | 230 | 660 | 600 | 259 |
| | 17 | 19 | 11 | 20 | 3 | 6 | 2 | 20 | 15 | 13 | 13 | 7 | 14 | 94 | 58 | 48 | 622 | 310 | 509 |
| | 18 | 26 | 12 | 19 | 3 | 2 | 0 | 19 | 13 | 10 | 15 | 2 | 7 | 134 | 71 | 84 | 504 | 235 | 186 |
| | 19 | 22 | 9 | 20 | 4 | 4 | 3 | 26 | 14 | 32 | 8 | 1 | 4 | 210 | 112 | 176 | 598 | 350 | 430 |
| | 20 | 16 | 12 | 18 | 1 | 2 | 1 | 5 | 23 | 18 | 5 | 2 | 6 | 45 | 240 | 254 | 125 | 325 | 217 |
| | 21 | 25 | 14 | 22 | 4 | 7 | 4 | 6 | 10 | 12 | 18 | 11 | 8 | 55 | 33 | 45 | 295 | 237 | 289 |
| | 22 | 18 | 11 | 13 | 5 | 3 | 0 | 7 | 11 | 4 | 7 | 1 | 10 | 98 | 131 | 74 | 730 | 517 | 550 |
| | 23 | 25 | 11 | 15 | 16 | 11 | 6 | 8 | 9 | 10 | 6 | 2 | 7 | 120 | 60 | 120 | 300 | 120 | 540 |
| | 24 | 22 | 12 | 11 | 4 | 3 | 0 | 7 | 7 | 9 | 12 | 2 | 6 | 120 | 120 | 120 | 600 | 180 | 240 |
| | 25 | 23 | 12 | 16 | 2 | 2 | 0 | 18 | 11 | 21 | 6 | 0 | 6 | 70 | 20 | 40 | 240 | 60 | 180 |
| | 26 | 32 | 12 | 17 | 8 | 11 | 2 | 29 | 14 | 25 | 12 | 3 | 8 | 121 | 34 | 46 | 458 | 165 | 138 |
| | 27 | 20 | 12 | 15 | 4 | 2 | 0 | 21 | 18 | 12 | 9 | 0 | 7 | 52 | 25 | 32 | 527 | 252 | 213 |
| | 28 | 16 | 12 | 13 | 1 | 0 | 0 | 7 | 9 | 5 | 9 | 1 | 10 | 46 | 21 | 15 | 30 | 60 | 82 |
| | 29 | 27 | 14 | 21 | 8 | 9 | 1 | 22 | 18 | 25 | 13 | 1 | 10 | 240 | 90 | 180 | 1440 | 1050 | 1140 |

| (b) | G | Count n-point C4 | C3 | i-node C4 | C3 | y-node C4 | C3 | x-node C4 | C3 | Time (s) n-point C4 | C3 | Node C4 | C3 |
|---|---|---|---|---|---|---|---|---|---|---|---|---|---|
| Workshop 1 | 1 | 14 | 22 | 7 | 20 | 17 | 24 | 3 | 11 | 330 | 90 | 521 | 521 |
| | 2 | 13 | 18 | 11 | 4 | 11 | 19 | 1 | 6 | 62 | 82 | 324 | 208 |
| | 3 | 18 | - | 19 | - | 27 | - | 4 | - | 97 | - | 405 | - |
| | 4 | 11 | 18 | 9 | 5 | 9 | 23 | 2 | 5 | 60 | 73 | 357 | 332 |
| | 5 | 18 | 21 | 9 | 6 | 6 | 23 | 4 | 5 | 110 | 55 | 420 | 312 |
| Workshop 2 | 6 | 18 | 23 | 14 | 11 | 13 | 17 | 1 | 6 | 120 | 60 | 600 | 360 |
| | 7 | 18 | 18 | 5 | 3 | 27 | 22 | 3 | 3 | 129 | 129 | 720 | 600 |
| | 8 | 14 | 14 | 5 | 7 | 14 | 18 | 1 | 4 | 323 | 713 | 115 | 143 |
| | 9 | 12 | 16 | 5 | 16 | 11 | 22 | 1 | 3 | 184 | 389 | 445 | 168 |
| | 10 | 10 | 16 | 5 | 4 | 5 | 13 | 2 | 4 | 116 | 113 | 290 | 465 |
| | 11 | 12 | 18 | 5 | 2 | 8 | 11 | 0 | 10 | 300 | 240 | 120 | 360 |
| | 12 | 17 | 16 | 23 | 54 | 8 | 22 | 3 | 4 | 64 | 52 | 140 | 205 |

**Key**

| Rank for n-point and node counts | Rank for n-point and node time |
|---|---|
| Lowest | Fastest |
| Medium | Medium |
| Highest | Slowest |

**Figure 4: Recorded fracture data (Nen, and node counts) and the time taken to undertake Nen and node counts for workshop (WS) participants (P) and groups (G). The data for each attribute has been colour-coded according to where the reported value for the parameter ranked for that circle. Data are presented in the order that they were completed in the workshop.**

[Figure]

**Figure 5: Node triangles for workshop participants and groups.** For individual circles (a), Participants 5, 21, and 11 were highlighted to show the consistency the way participants classified nodes. Participants were selected according the whether they reported a low (P5), medium (P21) or high (P11) node count. Similarly, for group circles (b) Groups 7 and 12 were highlighted as groups who recorded a high and low node count.

[Figure]

**Figure 6: Fracture trace length distributions for (a) individual and (b) group window sampling data. The results are presented as both histograms and normalised cumulative frequency curves of fracture trace length with bin widths of 0.05 m for individual and 0.1 m for group window sampling data. The range in the relative percentage of small fractures observed in the data is highlighted using Participants and groups who consistently observed a high and low percentage of small fractures (Participant's 3 and 24 and Groups 12 and 11 respectively).**

d

[Figure]

**Figure 7: A detailed study of the areas which cause increased uncertainty in Circle 8. The figure comprises of clean field photographs of Circle 8 with the (a) heat map of y-node point density, (b) heat map of fracture trace density and (c) areas identified as problem areas. In panel (d) the close up of areas 1, 2 and 3 along with the features recorded by Participants 11, 18 and 21 are shown. See text for full description.**

[Figure]

[Figure]

**Figure 8: The impact of participant experience on the collection of fracture data. (a)** The time taken in seconds to record fracture data ( and node counts) from circular scanlines both in the field and workshops. **(b)** The impact of experience on the recorded y-count and number of fractures in individual scanlines and the time taken to complete the workshop tasks.

[Figure]

[Figure]

**Figure 9: Topological sampling results for individuals and groups for circular scanlines 1, 3, 4, 5 and 8. Each histogram reports the results for all workshop participants. The statistics have been derived from the data for each participant. Data is presented as both bar charts and shaded histograms with the bin width, b, indicated on the chart. (please note the bin width varies between circles as a function of the range in reported or calculated values). In all cases the y-axis represents frequency and is scaled so the shape of the distributions can be assessed.**

[Figure]

**Figure 10: The effect of subject bias on the validity of circular scanlines.** The number of terminations recorded by individuals or groups is displayed for each circle and colour coded depending on where a valid (>30, green), possibly valid (20-30, yellow) or invalid (<20, red) number of terminations were recorded.

[Figure]

| (c) | Fracture statistic | A | B | Node triangle | Key |
|---|---|---|---|---|---|
| | scanline radius | 3.75 | 1.00 | | |
| Raw data | Nc | 9 | 12 | | |
| | i-node | 17 | 12 | | |
| | y-node | 14 | 28 | | |
| | x-node | 4 | 2 | | |
| Statistics | Intensity ($fm^{-1}$) | 0.60 | 3.00 | | |
| | Density ($FA^{-1}$) | 0.35 | 6.40 | | |
| | Tl (m) | 1.71 | 0.47 | | |
| | Pc | 0.77 | 0.88 | | |

[Figure]

| (c) | Fracture statistic | A | B | Node triangle | Key |
|---|---|---|---|---|---|
| | scanline radius | 3.75 | 1.00 | | |
| Raw data | n-point | 9 | 12 | | |
| | i-node | 17 | 12 | | |
| | y-node | 14 | 28 | | |
| | x-node | 4 | 2 | | |
| Statistics | Intensity ($fm^{-1}$) | 0.60 | 3.00 | | |
| | Density ($FA^{-1}$) | 0.35 | 6.40 | | |
| | TI (m) | 1.71 | 0.47 | | |
| | Pc | 0.77 | 0.88 | | |

**Figure 11: The impact of interpreter style on fracture statistics of a synthetic fracture network. (a) statistically valid topological sampling within a circular scanline for a fracture network which only considers the large scale fracture network. (b) statistically valid topological sampling within a circular scanline for the same large scale fracture network as (a), however, also capturing small scale fractures at fracture intersections. (c) The topology attributes (, i-, y- and x-nodes), derived fracture statistics and node triangle of the different interpretations of the fracture network.**

---

## Author Comment (AC5) · 25 Mar 2019

Dear Prof Laubach,

We are delighted for your supportive comment that you provided on our manuscript. We have included you in our response to reviewers document, as reviewer 3 (Please see RC reply's). Although many of the suggestions were clearly important, we often felt them beyond the scope of this contribution. Many of these points should be targets of research for the fracture community in the future. We included some of the key relevant points into our revised manuscript, and i would like to thank you again for the support shown to our work,

Kind regards, Billy J Andrews